# Geometric Optimal Transport for Unsupervised Domain Adaptation

**Gal Maman** *gmaman@campus.technion.ac.il*
*Viterbi Faculty of Electrical and Computer Engineering*
*Technion*

**Ronen Talmon** *ronen@ee.technion.ac.il*
*Viterbi Faculty of Electrical and Computer Engineering*
*Technion*

**Reviewed on OpenReview:** *https://openreview.net/forum?id=8Nef4vZUzU*

## Abstract

Optimal Transport (OT) is a widely used and powerful approach in domain adaptation. While effective, most existing methods rely on the pairwise squared Euclidean distances for the transportation cost, implicitly assuming a Euclidean space. In this paper, we challenge this assumption by introducing Geometric Optimal Transport (GOT), a new transport cost designed for domain adaptation under the manifold assumption. By utilizing concepts and tools from the field of manifold learning, specifically diffusion geometry, we derive an operator that accounts for the intra-domain geometries, extending beyond the conventional inter-domain distances. This operator, which quantifies the probability of transporting between source and target samples, forms the basis for our cost. We demonstrate how the proposed cost, defined by an anisotropic diffusion process, naturally aligns with the desired properties for domain adaptation. To further enhance performance, we integrate source labels into the operator, thereby guiding the anisotropic diffusion according to the classes. We showcase the effectiveness of GOT through comprehensive experiments, demonstrating its superior performance compared to recent methods across various benchmarks and datasets.

## 1 Introduction

Unsupervised Domain Adaptation (UDA) is an extensively studied field within machine learning. In UDA, given labeled data from a domain denoted as the source domain, we aim to train a model that will generalize well to unlabeled data from a different domain denoted as the target domain. Optimal Transport (OT) stands out as a key method for addressing the UDA problem. The essence of OT lies in finding the most efficient way to transport from one distribution to another while minimizing a transportation cost. It offers a robust approach to measuring the similarity between probability density functions and facilitating an optimal mapping between them. This capability makes it well-suited for UDA tasks, where there is a need for distribution alignment.

An established approach to utilizing OT for UDA is the Optimal Transport for Domain Adaptation (OTDA) framework (Courty et al., 2016), which directly employs the transport plan to map source samples to the target domain via barycentric mapping. While OTDA has proven effective in addressing the UDA task, the OT solution may not always yield the optimal mapping for maximizing target accuracy. To improve accuracy, a common practice is to add a regularization term into the OT problem formulation (Courty et al., 2014; 2016; Flamary et al., 2014). Alternatively, in recent years, the OT problem has been incorporated into deep learning models. One notable method, termed DeepJDOT (Damodaran et al., 2018), uses the OT problem to address the discrepancy between source and target distributions, subsequently leveraging the obtained transportation plan to train a feature extractor implemented through a deep model. Following DeepJDOT,

numerous studies have proposed OT-based deep models, aimed at improving target accuracy in the UDA task. Some have suggested different OT formulations to be used within the DeepJDOT framework (Fatras et al., 2021; Nguyen et al., 2022a), while others have introduced entirely new architectures and objective functions (Chen et al., 2018; Lee et al., 2019; Balaji et al., 2020). While research on utilizing OT for DA tasks is extensive and constantly evolving, one aspect that remains relatively unexplored is the selection of the transportation cost function. The transportation cost can take the form of any dissimilarity measure between samples in the source and target domains. Nevertheless, the predominant choice tends to be a distance function, with the most common choice being the squared Euclidean distance, computed between source and target samples. Only a small number of papers have suggested using a cost function diverging from the typical Euclidean distance measure (Tai et al., 2021; Nguyen et al., 2022b; Duque et al., 2023). Notably, this exploration of the transportation cost function is even rarer in the DA context, with only a few studies proposing alternative approaches, such as employing metric learning with a weighted Euclidean distance (Li et al., 2020; Xu et al., 2020) or Mahalanobis distance (Kerdoncuff et al., 2020; Jawanpuria et al., 2025) as the transport cost.

In this paper, we depart from reliance on distance functions as the transportation cost, and instead, focus on the probability of transportation from source samples to target samples. To achieve this, we introduce Geometric Optimal Transport (GOT), a new transportation cost based on an operator composed of three diffusion operators. The construction of the operator uses concepts and tools from the field of manifold learning, specifically, diffusion geometry (Coifman & Lafon, 2006). Initially, we construct a graph solely from the source samples, where each node represents a sample in the source data. From this graph, we build the first diffusion operator, which aims at preserving the local geometry of the source domain. When source labels are available, they are utilized to further guide the diffusion process, ensuring that each source sample can only diffuse (transport) to other samples within the same label category. Subsequently, we build a bipartite graph incorporating both source and target samples to capture the inter-domain relationships. We construct the second operator from this graph, which facilitates cross-domain diffusion. For the third diffusion operator, we construct a graph exclusively from target samples, aimed at capturing intra-domain similarities within the target domain. Taking the product of the three aforementioned operators forms the composite operator underlying GOT. We analyze the asymptotic behavior of this operator, demonstrating that the cost function is defined by an anisotropic diffusion process between the source and target domains. Our analysis underscores that this process is primarily influenced by the intrinsic geometry of the domains and their discrepancy.

**Our main contribution.** We introduce a new transportation cost, termed GOT, which is based on a composite diffusion operator consisting of three diffusion steps: (i) in the source domain, (ii) across domains, and (iii) in the target domain. This framework enables the learning of the geometries and relationships both between and within the two domains, distinguishing it from conventional approaches by considering both inter-domain distances and intra-domain structures. By incorporating source label information into our cost, we eliminate the need for regularization terms. Furthermore, the proposed cost is straightforward to compute and is derived directly from the data, in contrast to competing cost functions that necessitate learning the cost and are thus limited to specific frameworks. These features make GOT applicable to any OT problem formulation and ensure compatibility with most OT-based methods, with the experiments in the paper illustrating only a fraction of its broader potential. Our experiments demonstrate GOT's effectiveness compared to competing approaches. Specifically, when integrated into a deep domain adaptation framework, GOT achieves superior performance on benchmark datasets over the baseline and other OT-based methods. Additionally, we evaluate our cost on Motor Imagery (MI) benchmarks, showcasing its applicability to non-Euclidean data.

## 2 Related work

UDA is a subfield of machine learning that addresses the challenges of domain shifts and generalization, where a model trained on data from a source domain performs poorly when applied to a target domain with a different data distribution. Various methods and metrics have been proposed to align the distributions of the domains, including Maximum Mean Discrepancy (MMD) (Gretton et al., 2012; Pan et al., 2010),

Correlation Alignment (CORAL) (Sun et al., 2017), and Domain Adversarial Neural Networks (DANN) (Ganin et al., 2016), to mention just a few.

The Optimal Transport for Domain Adaptation (OTDA) framework was introduced by Courty et al. (2016). In addition to the general framework, the authors propose the integration of several regularization terms to improve performance, including a class-based regularization term (Courty et al., 2014) that utilizes source label information. By employing specific parameter configurations, the optimization problem can be effectively addressed using the Sinkhorn algorithm (Cuturi, 2013). The authors also proposed applying graph regularization on the transported samples (Flamary et al., 2014). While this regularization, termed Laplace regularization, considers neighborhood information, it is limited to Euclidean geometry. Furthermore, their optimization problem is quadratic, rendering it unsolvable by efficient algorithms and resulting in high computational complexity. A notable work is the JDOT framework introduced in Courty et al. (2017), which presents an optimization problem designed to simultaneously optimize both the transportation plan and the classifier. Building upon this concept, the DeepJDOT framework (Damodaran et al., 2018) further extends the idea by leveraging deep learning algorithms. This approach addresses two significant drawbacks in the OTDA framework. Firstly, handling large datasets becomes infeasible with OTDA due to the quadratic increase in complexity with the number of samples $N$, and the fact that it involves computations with $N \times N$ matrices. Conversely, DeepJDOT can train deep models using mini-batches, making it suitable for large datasets. Secondly, OTDA predominantly employs the squared Euclidean distance as the transportation cost, and, as a result, it often performs poorly on non-Euclidean datasets. In contrast, although the DeepJDOT framework also uses the squared Euclidean distance as the cost, it operates in a latent space learned by a deep model, potentially enabling superior performance on non-Euclidean data. In Fatras et al. (2021), the authors proposed enhancing the DeepJDOT framework by employing Unbalanced Optimal Transport (UOT), yielding improved results. A related work by Nguyen et al. (2022a) suggested solving it using Partial Optimal Transport (POT) and presented a two-stage implementation for the DeepJDOT framework, further enhancing target accuracy, particularly when coupled with the POT formulation. The OT problem was later incorporated into various deep learning and adversarial training frameworks (Chen et al., 2018; Lee et al., 2019; Xie et al., 2019; Balaji et al., 2020), all relying on the squared Euclidean distance as the ground cost.

Few papers in the literature have proposed alternative cost functions diverging from the standard distance metric. For instance, Tai et al. (2021) introduced an innovative approach termed SLA, which utilizes the OT formulation as a novel label assignment method. In this approach, each element $(i, k)$ in the transportation cost matrix is associated with the probability that sample $i$ belongs to class $k$. Gu et al. (2022) introduced a semi-supervised cost for Heterogeneous DA, which assumes access to a set of source-target pairs referred to as keypoints. Using these keypoints, the authors propose to compute a relation score for all source and target samples, which is subsequently used to construct the cost matrix. A related semi-supervised approach is presented in Duque et al. (2023) for manifold alignment. This method similarly relies on prior correspondence knowledge between some source and target samples to align the domains. It constructs intra-domain diffusion operators for each domain and combines them by multiplying the operators. The known correspondences are used to select the relevant rows, resulting in a joint diffusion operator. The final cost matrix is computed using the cosine distance between rows of the joint and intra-domain diffusion operators. While diffusion plays a central role in shaping the alignment, the approach is semi-supervised and does not incorporate inter-domain distances. Furthermore, the resulting cost lacks a probabilistic interpretation. Asadulaev et al. (2022) proposed a neural network-based framework for learning task-specific semi-supervised cost functions. Although the Gromov-Wasserstein distance typically leverages only intra-domain relationships, several works have explored approaches that integrate both intra- and inter-domain information, albeit not in the context of a transportation cost or specifically for DA. These methods are worth noting, as they share conceptual similarities with our approach in incorporating structural information from both domains. For example, Yan et al. (2018) introduced a semi-supervised DA method that adds a regularization term to the optimization problem. This term incorporates cross-domain distances, computed in the target domain after transportation, and relies on target labels. Alternatively, Xu et al. (2019) presented a graph-matching method that learns an embedding space for the nodes, enabling the computation of inter-domain relationships, which are included in the optimization via an additional term.

In the context of UDA, Kerdoncuff et al. (2020) proposed using Mahalanobis metric learning for the cost matrix. Specifically, they learn the metric and the transport plan simultaneously through alternating optimization, where source labels are incorporated via a regularization term to guide the metric learning. Similarly, Jawanpuria et al. (2025) also employed Mahalanobis distance with metric learning. Here, a regularization term is introduced, which not only prevents trivial solutions but also enables a closed-form solution for the metric given the transport plan. In both methods, the Mahalanobis distance accounts only for inter-domain relationships (similar to the Euclidean cost), and intra-domain structure is not incorporated. Dhouib et al. (2020) derived a bound on the target margin violation rate and proposed a DA algorithm based on this bound. By using the dual representation of the infinity norm in their discrete bound, they parameterize the cost with a learned vector (that acts as a weights vector – non-negative elements and sums to 1), and solve a min-max problem to optimize the plan and the learned vector. This method considers only inter-domain relationships and does not incorporate source labels.

In the context of deep UDA, Li et al. (2020) introduced ETD, which utilizes a weighted Euclidean distance for the cost function. They proposed leveraging an attention mechanism to learn the correlation between source and target samples, subsequently using this correlation as the weights in their proposed transportation cost. The OT distance is then employed in the training objective, serving as a quantification of the domain discrepancy. The work in Xu et al. (2020) also proposed using a weighted Euclidean distance as the transportation cost. In that method, termed RWOT, the element $(i,k)$ in the weight matrix quantifies the probability that the $i$-th sample belongs to class $k$. This probability is dynamically computed based on both spatial prototypical information and the pseudo-classification probability of target samples. Both methods have fundamental drawbacks compared to our proposed approach. First, ETD and RWOT rely on learned weights, limiting their applicability to specific deep learning frameworks, while GOT is derived directly from the data. Second, the costs in ETD and RWOT, including the learned weights, consider only the inter-domain relationships and source label information, whereas our cost also incorporates intra-domain relationships for both source and target domains.

By integrating principles from diffusion maps (Coifman & Lafon, 2006), our approach incorporates intra-domain relationships and aims to reflect the underlying geometric structure of the data in the proposed cost. Diffusion maps, originally a dimensionality reduction technique, provide a method to analyze and visualize high-dimensional data. The core concept involves constructing a transition kernel, where transition probabilities are defined by the local similarities between data samples. Diffusion geometry has been widely applied in various fields, including data analysis (Coifman & Lafon, 2006), computer vision (Bronstein et al., 2010; Liu et al., 2012), and medical imaging (Haghverdi et al., 2015; Zheludev et al., 2015), among others. For further background on diffusion geometry, see Appendix A.

## 3 Optimal Transport for Domain Adaptation

We formulate the problem we consider and briefly describe the OTDA framework proposed in Courty et al. (2016).

**Problem formulation.** Let $\mathcal{X}$ be an input space and $\mathcal{Y}$ be a label space. We define a domain as a distribution $\mathcal{D}$ over the input space $\mathcal{X}$. Consider $N_s$ samples $\{x_i^s \in \mathcal{X}\}_{i=1}^{N_s}$ drawn i.i.d from a source domain $\mathcal{D}_s$, associated with labels $\{y_i^s \in \mathcal{Y}\}_{i=1}^{N_s}$, and $N_t$ unlabeled samples $\{x_j^t \in \mathcal{X}\}_{j=1}^{N_t}$, drawn i.i.d from a target domain $\mathcal{D}_t$. In UDA, the goal is to enable a predictive model $f$, trained on labeled source samples, to generalize well to target samples. There are two dominant approaches for UDA. In the first approach, a mapping function $g : \mathcal{X} \to \mathcal{X}$ is learned to align the source domain to the target domain, e.g., by minimizing the divergence between them. A predictive model $f$ is then trained using the mapped source samples, denoted by $z_i^s = g(x_i^s)$. In the second approach, we learn an embedding function $g : \mathcal{X} \to \mathcal{Z}$ that maps both domains to a latent space $\mathcal{Z}$ of domain-invariant features. Thus, a predictive model $f : \mathcal{Z} \to \mathcal{Y}$, trained using the source embeddings $\{z_i^s = g(x_i^s)\}_{i=1}^{N_s}$ with labels $\{y_i^s\}_{i=1}^{N_s}$ will generalize well on the target embeddings $\{z_j^t = g(x_j^t)\}_{j=1}^{N_t}$.

**OTDA framework.** We start by outlining the discrete OT formulation, which is consistently employed throughout this paper. For any source sample $x_i^s$ and target sample $x_j^t$, let $c(x_i^s, x_j^t)$ denote the cost to

move a probability mass from $x_i^s$ to $x_j^t$, which is typically chosen to be the squared Euclidean distance. Let $\mathbf{C} \in \mathbb{R}^{N_s \times N_t}$ be the cost matrix, with elements $\mathbf{C}_{ij} = c(x_i^s, x_j^t)$. The discrete OT formulation is given by:

$$\gamma^* = \arg\min_{\gamma \in \Gamma} \langle \gamma, \mathbf{C} \rangle_F, \tag{1}$$

where $\Gamma = \left\{ \gamma \in \mathbb{R}^{N_s \times N_t} | \gamma \mathbb{1}_{N_t} = \mathbf{a}, \gamma^T \mathbb{1}_{N_s} = \mathbf{b} \right\}$ denotes the set of transport plans that satisfy conservation of mass, $\mathbf{a} \in \mathbb{R}^{N_s}$ and $\mathbf{b} \in \mathbb{R}^{N_t}$ are the empirical distributions of the source and target samples, respectively, and $\langle \cdot, \cdot \rangle_F$ denotes the standard Frobenius inner product.

This optimization problem has been studied for many years and addressed through various algorithms. To date, one of the most commonly employed algorithms is the efficient Sinkhorn algorithm (Cuturi, 2013), which incorporates entropy regularization into the objective function. This leads to the following modified problem that can be solved iteratively using linear projections, offering reduced complexity:

$$\gamma^* = \arg\min_{\gamma \in \Gamma} \langle \gamma, \mathbf{C} \rangle_F + \lambda \Omega_s(\gamma), \tag{2}$$

where $\Omega_s(\gamma) = \sum_{i,j} \gamma(i,j) \log \gamma(i,j)$ is the negative entropy of $\gamma$, and $\lambda$ is a hyperparameter.

The OT problem can be utilized for UDA in both of the aforementioned approaches. Recent deep learning-based methods, such as Damodaran et al. (2018); Chen et al. (2018); Lee et al. (2019), incorporate OT into the loss function to learn domain-invariant representations, thereby following the second approach for UDA. In contrast, the OTDA framework follows the first approach, where, after solving the OT problem, source samples are transported to the target domain using the obtained transport plan. The transportation of any source sample $x_i^s$ is given by the barycentric mapping (Courty et al., 2016; Perrot et al., 2016), defined by:

$$z_i^s = \arg\min_{x \in \mathcal{X}} \sum_{j=1}^{N_t} \gamma_{ij}^* c(x, x_j^t). \tag{3}$$

## 4 Geometric Optimal Transport

In this work, we present GOT, a new cost function for the OT problem, designed specifically for the UDA setting described in Section 3. Our premise is the manifold assumption, a widely accepted principle in modern high-dimensional data analysis that has given rise to an extensive body of work termed manifold learning (Roweis & Saul, 2000; Tenenbaum et al., 2000; Belkin & Niyogi, 2003; Coifman & Lafon, 2006). In the UDA setting, it implies that both domains are supported on the same hidden manifold $\mathcal{M}$ embedded in $\mathcal{X}$. When the squared Euclidean distance is used as the transport cost, whether explicitly stated or not, it is assumed that the source and target domains are embedded in $\mathbb{R}^d$. The manifold assumption extends this concept, allowing for the possibility that the samples may not lie within a Euclidean space. Building on this foundation, we propose defining the OT cost matrix $\mathbf{C}$ based on a diffusion operator, composed of three diffusion steps. Notably, the construction of this operator stems from diffusion geometry (Coifman & Lafon, 2006), which is a prevalent manifold learning approach (see Appendix A for details).

Initially, we build an operator solely from the source samples, aimed at capturing the local geometry within the source domain. We construct a graph $G_s$ with $N_s$ nodes, where each node represents a source sample. Under the manifold assumption, geodesic distances on $\mathcal{M}$ are locally Euclidean. We use the Euclidean distance to build the affinity matrix of the graph, $\mathbf{K}_s \in \mathbb{R}^{N_s \times N_s}$, with a Gaussian kernel:

$$\mathbf{K}_s(i,j) = \exp\left( -\frac{\|x_i^s - x_j^s\|_2^2}{\epsilon_s} \right), \tag{4}$$

where $\epsilon_s > 0$ is a scale hyperparameter. Importantly, although the kernel is based on Euclidean distances, this is fundamentally different from using the Euclidean distance directly as the OT cost: by employing this local kernel, which quantifies pairwise local similarities between data samples, the graph effectively captures the local neighborhood relationships.

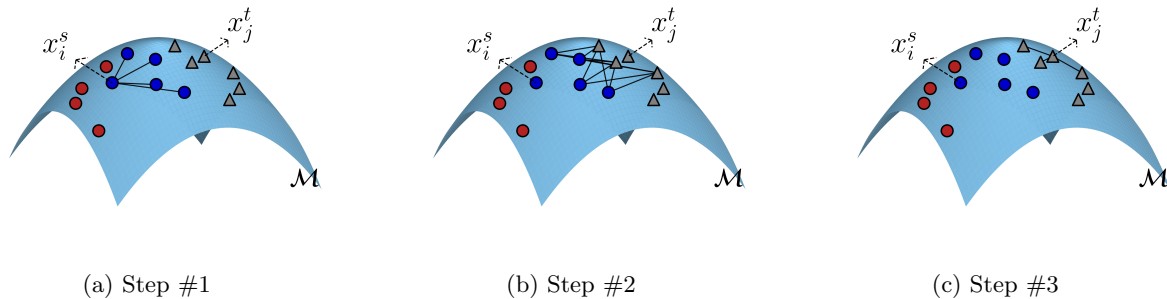

(a) Step #1        (b) Step #2        (c) Step #3

Figure 1: Illustration of the diffusion process for computing $\mathbf{S}_{ij}$, the transition probability from the source $i$-th sample to the target $j$-th sample. Source samples are represented by circles, target samples by triangles, and the source markers are color-coded by labels. (a) Transition from $x_i^s$ to nearby source samples of the same class. (b) Transition from those source samples to the 3 nearest neighbors of $x_j^t$. (c) Transition from the target neighbors to $x_j^t$. The final probability $\mathbf{S}_{ij}$ aggregates all paths from $x_i^s$ to $x_j^t$.

Subsequently, we obtain the source diffusion operator, denoted by $\mathbf{P}_s \in \mathbb{R}^{N_s \times N_s}$, by normalizing the affinity matrix $\mathbf{K}_s$ to be row-stochastic, i.e., ensuring that the sum of each row equals 1. Following this standard normalization (Coifman & Lafon, 2006), the rows of $\mathbf{P}_s$ represent the transition probabilities of a diffusion process on $\mathcal{M}$ defined at the source samples $\{x_i^s\}_{i=1}^{N_s}$.

In the second diffusion step, we apply a cross-domain diffusion, which captures the inter-domain relationships by incorporating both the source and target samples. To this end, we define a bipartite graph $G_c$ with a vertex set $V_c = \{v_1^s, \ldots, v_{N_s}^s, v_1^t, \ldots, v_{N_t}^t\}$ that represents the $N_s$ source samples and $N_t$ target samples. In this graph, each node $v_i^s$ has edges only to target nodes, and each node $v_j^t$ has edges only to source nodes. Consequently, the graph adjacency matrix consists of two off-diagonal blocks. For our purpose, we utilize only the $N_s \times N_t$ top-right block of the matrix. The weight of the edge between source node $v_i^s$ and target node $v_j^t$ is given by:

$$\mathbf{K}_c(i, j) = \exp\left(-\frac{\|x_i^s - x_j^t\|_2^2}{\epsilon_c}\right), \tag{5}$$

where $\epsilon_c > 0$ is a scale hyperparameter. We normalize this kernel matrix to be row-stochastic, and denote the resulting cross-domain diffusion operator by $\mathbf{Q} \in \mathbb{R}^{N_s \times N_t}$.

In the third and last diffusion step, designed to capture intra-domain similarities within the target domain, we construct the graph $G_t$ exclusively from target samples. This graph comprises $N_t$ nodes, representing the target samples, and an affinity matrix $\mathbf{K}_t \in \mathbb{R}^{N_t \times N_t}$, constructed using a Gaussian kernel with scale $\epsilon_t$ similarly to Equation 4. After normalizing $\mathbf{K}_t$ to be row-stochastic, we obtain the target diffusion operator, denoted by $\mathbf{P}_t \in \mathbb{R}^{N_t \times N_t}$.

The final operator $\mathbf{S}$, which serves as the basis for the transport cost, is defined by:

$$\mathbf{S} = \mathbf{P}_s \mathbf{Q} \mathbf{P}_t. \tag{6}$$

By construction, $\mathbf{S}(i, j)$ is the probability of transporting from source sample $x_i^s$ to target sample $x_j^t$ through local neighborhoods. Specifically, $\mathbf{S}(i, j)$ encapsulates the probability of (i) diffusing from source sample $x_i^s$ to its source neighbors, (ii) diffusing from these source neighbors to target samples that are neighbors of $x_j^t$, and finally (iii) diffusing from these target neighbors to $x_j^t$. An illustration of this triple diffusion process is presented in Figure 1. The intra-domain graphs use local neighborhoods, defined based on class labels for $G_s$ and nearest neighbors for $G_t$ (details in Section 4.2).

As a final step, we apply a negative element-wise logarithm to the diffusion probabilities to translate the notion of similarity into a notion of cost, aligning with the requirements of the OT problem. Thus, the proposed transportation cost matrix, referred to as GOT, is defined by: $\mathbf{C} = -\log(\mathbf{S})$.

The computational complexity of GOT is $\mathcal{O}(n^3)$; see Appendix B.1 for details, and Figure 10 for an empirical run-time comparison with competing methods.

## 4.1   Theoretical results

While competing methods typically do not include theoretical insights, our approach is driven by the following theoretical basis. Specifically, we prove that the proposed operator $\mathbf{S}$ is a diffusion operator and analyze its asymptotic behavior, showing how it aligns with the desired properties. This analysis provides a deeper understanding of the motivation behind our method. Building on this, we now transition to examining the diffusion operator in a continuous setting.

Consider infinite source and target datasets, sampled i.i.d from the probability distributions $\mu$ and $\nu$, respectively, defined over the continuous manifold $\mathcal{M}$. Consider a Gaussian kernel $k_\epsilon(x, y)$ with a scale parameter $\epsilon$. In analogy to the row-stochastic normalization in the discrete case, we define the continuous source diffusion operator $P_{s,\epsilon}$ by:

$$P_{s,\epsilon}f(y) = \int p_{s,\epsilon}(x, y)f(x)\mu(x)dx = \int \frac{k_\epsilon(x, y)}{d_{s,\epsilon}(x)}f(x)\mu(x)dx, \tag{7}$$

where $p_{s,\epsilon}(x, y)$ is the normalized kernel, and $d_{s,\epsilon}(x) = \int k_\epsilon(x, y)\mu(y)dy$.

Similarly, we define the continuous target diffusion operator $P_{t,\epsilon}$, which operates on the target domain, by employing Equation 7 with respect to $\nu$ and with the normalized kernel $p_{t,\epsilon}(x, y) = \frac{k_\epsilon(x, y)}{d_{t,\epsilon}(x)}$, where $d_{t,\epsilon}(x) = \int k_\epsilon(x, y)\nu(y)dy$.

The cross-domain diffusion operator is defined by:

$$Q_\epsilon f(y) = \int \frac{k_\epsilon(x, y)}{d_{t,\epsilon}(x)}f(x)\mu(x)dx. \tag{8}$$

Note that in this analysis, without loss of generality, we use the same kernel $k_\epsilon(x, y)$ for all three operators.

In the discrete case, applying the diffusion operator defined in Equation 6 to the source samples necessitates using the transpose of $\mathbf{S}$. Thus, in the continuous case, the diffusion operator $S_\epsilon$, defined as the composition of the three operators $P_{s,\epsilon}$, $Q_\epsilon$, and $P_{t,\epsilon}$, can be expressed as $S_\epsilon f(x) = P_{t,\epsilon}Q_\epsilon P_{s,\epsilon}f(x)$.

**Proposition 4.1.** *Suppose $f \in \mathcal{C}^4(\mathcal{M})$, and suppose $\mu, \nu \in \mathcal{C}^4(\mathcal{M})$ denote the probability measures of the source and target domains, respectively, where $\mu$ is dominated by $\nu$. For sufficiently small $\epsilon$, the asymptotic expansion of operator $S_\epsilon$ is given by:*

$$S_\epsilon f(x) = \frac{\mu}{\nu}(x)\left[f - \frac{m_2}{m_0}\epsilon\left[3\Delta f + 2\left(f\frac{\Delta\left(\frac{\mu}{\nu}\right)}{\frac{\mu}{\nu}} + 2\nabla f\nabla \log\left(\frac{\mu}{\nu}\right)\right)\right. \right. \tag{9}$$
$$\left. \left. - f\left(\frac{\Delta\mu}{\mu} + 2\frac{\Delta\nu}{\nu}\right)\right]\right](x) + O(\epsilon^2),$$

*where $\Delta, \nabla$ are the Laplace–Beltrami operator and the covariant derivative on $\mathcal{M}$, respectively, and $m_0, m_2$ are two constants defined by the Gaussian kernel and by the manifold $\mathcal{M}$.*

The proof appears in Appendix D.

We remark that if $\mu(x) = \nu(x)$ for all $x \in \mathcal{M}$, indicating no domain shift, the asymptotic expansion is given by:

$$S_\epsilon f(x) = f(x) - \frac{m_2}{m_0}\epsilon\left(3\Delta f - 3f\frac{\Delta\mu}{\mu}\right)(x) + O(\epsilon^2), \tag{10}$$

coinciding with the asymptotic expansion of a diffusion operator on $\mathcal{M}$ with distribution $\mu$ (Coifman & Lafon, 2006), applied three times.

The expression derived in Proposition 4.1 suggests that GOT inherently considers both the domain shift and the intra-domain relationships of the source and target domains. In particular, the result implies that

the proposed diffusion operator leads to an anisotropic diffusion process on the manifold $\mathcal{M}$, influenced by the manifold geometry (conveyed by the differential operators $\Delta$ and $\nabla$ w.r.t. the manifold), the source and target domains (captured by the terms within the second set of round parentheses), and the discrepancy between them (captured by the term within the first set of round parentheses, involving both the first- and second-order derivatives of $\frac{\mu}{\nu}$). We illustrate the induced anisotropic diffusion process with a 2D toy example outlined in Appendix B.2.

In OT, the goal is to find a transport plan that minimizes the total cost of moving mass between distributions. In our approach, this cost is obtained by applying a negative logarithm to the diffusion operator $S$. Viewing the expression derived in Proposition 4.1 from this perspective highlights the advantages of using the operator as the basis for our transport cost, particularly in the context of domain adaptation.

When solving the transportation problem, samples that minimize the cost are favored – that is, they are assigned more mass in the transport plan. More specifically, by maximizing the term $\left(\frac{\Delta\mu}{\mu} + 2\frac{\Delta\nu}{\nu}\right)(x)$ (or assigning more weight to samples for which this term is large), we focus on regions with non-uniform density, where the source and target densities change rapidly. These regions tend to be sparser (where $\mu$ is small). More importantly, they often correspond to key geometric structures in the data (where $\Delta\mu$ is large). In the context of transport, this focus provides an additional advantage: samples in such regions carry richer structural information and are less ambiguous, whereas samples in uniform-density areas tend to lack distinctive local features, making them harder to match.

A natural question arises: what happens at locations where $\frac{\Delta\mu}{\mu}$ is high while $\frac{\Delta\nu}{\nu}$ is low, or vice versa? In other words, how do we avoid assigning high weight to regions where, for example, the source has low density while the target has high density? This is where the term $\frac{\Delta\left(\frac{\mu}{\nu}\right)}{\frac{\mu}{\nu}} + 2\nabla f \nabla \log\left(\frac{\mu}{\nu}\right)$ becomes important. Minimizing this expression encourages transport between regions where the ratio $\frac{\mu}{\nu}$ varies smoothly. In such cases, high-density regions in the source are aligned with high-density regions in the target, and likewise for low-density regions. This prevents density inversion, where a dense source region is mapped to a sparse target region – a mismatch that would distort the local geometry. Figure 5 in Appendix B.3 provides a detailed illustration of this analysis.

## 4.2 Enhanced intra-domain kernels

In UDA, we assume access to labeled source samples. To leverage this information, we incorporate source labels into the proposed cost, guiding the diffusion based on the class labels. Specifically, rather than using fully connected graphs for the intra-domain kernels, as in Equation 4, we adopt the widely used approach of integrating neighborhood information into the Gaussian kernel. The resulting intra-domain kernel:

$$\mathbf{K}(i,k) = \begin{cases} \exp\left(-\frac{\|x_i - x_k\|_2^2}{\epsilon}\right) & \text{if } k \in \mathcal{N}_i \\ 0 & \text{otherwise} \end{cases}, \tag{11}$$

where $\mathcal{N}_i$ is defined as the set of all indices $\{k\}$ such that the sample $x_k$ is within the neighborhood of $x_i$.

For the source operator $\mathbf{P}_s$, where labels are available, we define the neighborhood as $\mathcal{N}_i = \{k \mid y_k^s = y_i^s\}$, ensuring that each source sample $x_i^s$ diffuses only to other source samples within the same class. This approach stands in contrast to other methods that propose incorporating source labels by adding a regularization term to the problem formulation. Such regularization terms often impose constraints on the OT solver or the learned geometry, potentially restricting the method's adaptability and performance.

For the target operator $\mathbf{P}_t$, or in scenarios where labels are unavailable in a specific DA setup, we define the neighborhood $\mathcal{N}_i$ as the $K$ nearest neighbors of $x_i$. This allows leveraging local structure effectively, even in the absence of label information. In this work, we set $K = 3$ for the target neighbors across all toy and real experiments. Appendix C.1.2 provides a discussion of this choice and demonstrates its robustness.

The proposed method is summarized in Algorithm 1.

---

**Algorithm 1** GOT

---

**Input**: $\{((x_i^s, y_i^s)\}_{i=1}^{N_s}, \{x_j^t\}_{j=1}^{N_t}, \epsilon_s, \epsilon_c, \epsilon_t.$

1: Compute the intra-domain kernel of the source, $\mathbf{K}_s$, using Equation 11 and supervised neighborhood information.
2: Compute the intra-domain kernel of the target, $\mathbf{K}_t$, using Equation 11 with a three nearest neighbors neighborhood.
3: Compute the cross-domain kernel $\mathbf{K}_c$ according to equation 5.
4: Compute the row-stochastic probability matrices, $\mathbf{P}_s, \mathbf{P}_t$, and $\mathbf{Q}$.
5: Compute the diffusion matrix $\mathbf{S}$ according to Equation 6.
6: Apply element-wise negative logarithm $\mathbf{C} = -\log(\mathbf{S})$.
7: Return the cost matrix $\mathbf{C}$.

---

## 5 Experiments

In this section, we provide a visual comparison of GOT with competing methods using a toy example (Section 5.1). Additionally, we present experiments on deep domain adaptation tasks (Section 5.2) and non-Euclidean dataset adaptation (Section 5.3), comparing the performance of GOT against the leading methods. For all experiments, we employed the label-enhanced kernel for the source diffusion operator (Section 4.2). For all three diffusion operators $\mathbf{P}_s, \mathbf{Q}, \mathbf{P}_t$, before computing the Gaussian kernels, we conducted median normalization on the distance matrix. After computing the kernels, we performed doubly-stochastic normalization on the square matrices $\mathbf{P}_s$ and $\mathbf{P}_t$ using the Sinkhorn algorithm, as suggested in Landa et al. (2021). The matrix $\mathbf{Q}$, which may not necessarily be square, was normalized to be row-stochastic. Unless specified otherwise, we employed the Sinkhorn algorithm (Cuturi, 2013) to solve the OT problem, utilizing the publicly available POT package (Flamary et al., 2021). While target labels were not used during training, we did rely on them as a validation set for hyperparameter tuning, consistent with competing methods. Parameters were optimized for each dataset as a whole, rather than for each individual source–target pair. Full details of the GOT parameters are provided in Appendix C.1, and the parameters used in each experiment are reported in Appendix C. In Section 5.3, we explore an unsupervised tuning strategy on a multiclass BCI task. All the experiments were conducted on Nvidia DGX A100. The source code is available here[1].

### 5.1 Two Moons: Visual Illustration

We utilize the toy example presented in Courty et al. (2016), which involves two entangled moons as the source domain, each representing a different class. The target domain is independently sampled from a rotated version of the source distribution. As the rotation angle increases, the problem becomes more challenging. Further details are provided in Appendix C.2.

We employ GOT with the entropy-regularized OT problem (Cuturi, 2013), as presented in Equation 2, to derive the transport plan. The baseline method, OT, uses this formulation with the standard cost. Additionally, due to the limited number of methods proposing alternative transport costs – particularly those that integrate source label information and can be applied in non-deep frameworks – we include two regularization approaches: OT-reg (Courty et al., 2016; 2014), which incorporates source labels, and OT-Laplace (Courty et al., 2016; Flamary et al., 2014), which uses a Laplace regularization term to preserve data structure. We emphasize that all three methods use the squared Euclidean distance as the transportation cost. Table 6 in Appendix C.2 summarizes the key differences between the methods.

In Figure 2, we visualize the obtained plans for a 40° rotation angle, and focus on one target sample, marked by a bold black triangle. For clarity, we plot lines from this sample to the ten source samples with the highest values in the corresponding column of the plan. In Figure 2a, we observe that GOT successfully achieves the optimal transportation plan, with all source samples contributing mass to the red-labeled target sample associated with the red class. In OT (Figure 2b), since the cost is based solely on the pairwise Euclidean distances, the mass is coming from the closest source samples, most of which belong to the wrong class. In

---

[1]https://github.com/GalMaman/GOT

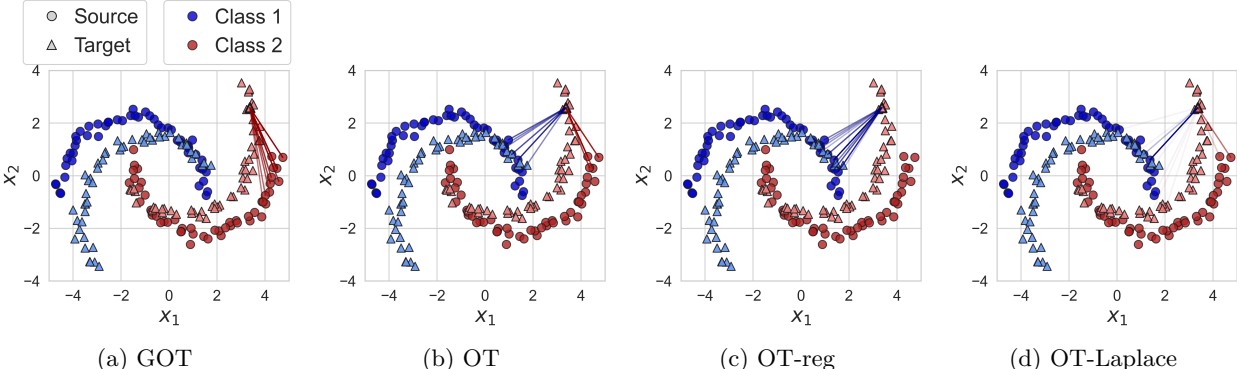

Figure 2: Two moons illustration for a 40° rotation angle. Source samples are marked with circles, target samples with triangles, and the colors of the markers indicate the labels. Each line represents an entry $\gamma(i,j)$ in the transportation plan, where the $j$-th sample is marked by a bold black triangle. For clarity, we plot lines from this sample to the ten source samples with the highest values in the $j$-th column of the plan. The line color corresponds to the label of the source sample $x_i^s$, and the line intensity reflects the values in the plan. (Best viewed in colors).

OT-reg (Figure 2c), the regularization term encourages a solution where each target sample receives mass only from source samples of the same class. In this case, it results in the most undesirable plan, where all source samples are from the opposite class. In OT-Laplace (Figure 2d), the regularization aims to keep nearby samples close after transportation. However, the encoded neighborhood information is insufficient to correct the Euclidean cost, and the plan mirrors the behavior of the standard OT. We note that the plan is sparser as the method does not include entropy regularization.

This scenario highlights GOT's effectiveness in handling challenging mappings. GOT leverages source labels not as strict constraints, as OT-reg, but to guide the diffusion process in the source domain, while still accounting for the local geometry of the data. Additionally, it shows that by taking into account intra-domain relationships, rather than relying on pairwise Euclidean distances, the chosen sample can receive mass from source samples that are remote in Euclidean terms, as the diffusion distance is effectively incorporated. Indeed, we see that GOT successfully connects the chosen target sample to the correct, more distant source samples, which no other method considered. We provide further visual analysis and numerical comparison in Appendix C.2.

## 5.2 Deep Domain Adaptation

In the following experiments, we incorporate GOT into the DeepJDOT framework (Damodaran et al., 2018), which comprises two components: an embedding function $g : \mathcal{X} \to \mathcal{Z}$, designed to learn features that are both discriminative and domain-invariant, and a classification function $f : \mathcal{Z} \to \mathcal{Y}$. The DeepJDOT cost matrix for estimating the transport plan $\gamma$, given the fixed-parameter models $\hat{f}$ and $\hat{g}$, is given by:

$$\mathbf{C}_{ij} = \eta_1 \|\hat{g}(x_i^s) - \hat{g}(x_j^t)\|_2^2 + \eta_2 \mathcal{L}\left(y_i^s, \hat{f}(\hat{g}(x_j^t))\right), \tag{12}$$

where $\mathcal{L}$ is the cross-entropy loss, and $\eta_1, \eta_2$ are hyperparameters.

The cost combines (i) feature alignment via Euclidean distance in the latent space, and (ii) label alignment via cross-entropy between source labels and target pseudo-labels. In our work, we replace the Euclidean term in $\mathbf{C}_{ij}$ with the proposed GOT cost. For additional information about the framework and the optimization process, see Appendix C.3.

We apply our method to three benchmark domain adaptation datasets: Digits, Office-Home, and VisDA. One of the key advantages of our method is its adaptability to any OT formulation. To demonstrate this flexibility, we employ different formulations for each dataset, as detailed below, resulting in a different baseline method across datasets. We compare our method to the DeepJDOT framework with the standard

Table 1: Target domain accuracy. Results on (a) the Digits dataset, and (b) the VisDA dataset.

(a)

| Method | SVHN-MNIST | USPS-MNIST | MNIST-USPS | Avg |
|---|---|---|---|---|
| DANN | 95.80 ± 0.29 | 94.71 ± 0.12 | 91.63 ± 0.53 | 94.05 |
| ALDA | 98.81 ± 0.08 | 98.29 ± 0.07 | 95.29 ± 0.16 ' | 97.46 |
| ETD | 97.9 ± 0.4 | 96.3 ± 0.1 | 96.4 ± 0.3 | 96.9 |
| RWOT | 98.8 ± 0.1 | 97.5± 0.2 | **98.5 ± 0.2** | 98.3 |
| DeepJDOT | 96.04 ± 0.66 | 97.22 ± 0.23 | 86.12 ± 0.60 | 93.13 |
| JUMBOT | 98.99 ± 0.06 | 98.68 ± 0.08 | 96.93 ± 0.42 | 98.20 |
| m-POT | 98.98 ± 0.08 | 98.63 ± 0.13 | 96.04 ± 0.02 | 97.88 |
| Ours | **99.19 ± 0.04** | **98.87 ± 0.03** | 97.81 ± 0.30 | **98.62** |

(b)

| Method | Accuracy |
|---|---|
| DANN | 67.63 ± 0.34 |
| ALDA | 71.22 ± 0.12 |
| DeepJDOT | 69.58 ± 0.34 |
| JUMBOT | 72.97 ± 0.26 |
| TS-POT | 75.65 ± 0.78 |
| Ours | **78.56 ± 0.15** |

Table 2: Target domain accuracy on the Office-Home dataset (ResNet-50).

| Method | A-C | A-P | A-R | C-A | C-P | C-R | P-A | P-C | P-R | R-A | R-C | R-P | Avg |
|---|---|---|---|---|---|---|---|---|---|---|---|---|---|
| ResNet-50 | 34.9 | 50.0 | 58.0 | 37.4 | 41.9 | 46.2 | 38.5 | 31.2 | 60.4 | 53.9 | 41.2 | 59.9 | 46.13 |
| DANN | 47.9 | 67.1 | 74.9 | 53.8 | 63.5 | 66.4 | 53.0 | 44.4 | 74.4 | 65.5 | 53.0 | 79.4 | 61.93 |
| ALDA | 54.0 | 74.9 | 77.1 | 61.4 | 70.6 | 72.8 | 60.3 | 51.0 | 76.7 | 67.9 | 55.9 | 81.9 | 67.04 |
| ETD | 51.3 | 71.9 | **85.7** | 57.6 | 69.2 | 73.7 | 57.8 | 51.2 | 79.3 | 70.2 | 57.5 | 82.1 | 67.29 |
| RWOT | 55.2 | 72.5 | 78.0 | 63.5 | 72.5 | 75.1 | 60.2 | 48.5 | 78.9 | 69.8 | 54.8 | 82.5 | 67.63 |
| DeepJDOT | 52.0 | 70.9 | 76.1 | 60.5 | 66.6 | 69.2 | 58.4 | 48.7 | 75.3 | 68.9 | 54.9 | 79.9 | 65.12 |
| JUMBOT | 55.7 | 75.0 | 80.7 | 65.1 | 74.5 | 75.1 | 65.3 | 53.3 | 79.6 | 74.5 | 59.3 | 83.9 | 70.17 |
| TS-POT | **57.4** | 77.1 | 81.6 | 68.3 | 72.8 | 76.5 | 67.4 | 55.1 | 80.6 | 75.4 | 59.9 | 84.0 | 71.36 |
| Ours | 57.2 | **78.0** | 82.1 | **70.2** | **74.9** | **78.8** | **68.1** | **56.5** | **82.0** | **75.6** | **60.9** | **84.8** | **72.43** |

cost (DeepJDOT, JUMBOT, and m-POT), OT-based methods with alternative costs (ETD and RWOT), and DA baselines (DANN and ALDA). Further details are provided in Appendix C.3.

**Results.** Table 1a presents the results for the Digits datasets. Details regarding the model architecture and parameters for each method are available in Appendix C.3.1. For the digits dataset, we follow Fatras et al. (2021) and use the UOT formulation to obtain the transportation plan, making JUMBOT the baseline method. The table displays the mean and standard deviation of the target test set over three runs. Results for each individual run can be found in Table 8 in Appendix C.3.1. The highest accuracy is indicated in bold, while the second highest is underlined. Our method achieves the highest accuracy for SVHN→MNIST and USPS→MNIST. Additionally, it attains the second highest accuracy for MNIST→USPS, followed by RWOT. We acknowledge that this dataset is considered an easy DA task, and it appears that the generator learns features that are well separated by the classes, even when using the standard cost. Therefore, the obtained minor improvement is expected.

Table 2 shows the results for the Office-Home dataset. In this experiment, we again incorporate the proposed cost into the UOT problem formulation to derive the optimal plan. Here, DeepJDOT and JUMBOT were reproduced using the original DeepJDOT framework, while TS-POT was reproduced using a two-stage (TS) implementation, as proposed in the original paper (Nguyen et al., 2022a). The results are averaged over three runs, with full details, including the parameters used for each method, provided in Appendix C.3.2. Table 2 summarizes the target test accuracy for each scenario of the Office-Home dataset. Remarkably, the proposed method achieves the highest average accuracy across the 12 scenarios and outperforms the tested existing methods in 10 out of the 12 scenarios. Specifically, it surpasses the baseline JUMBOT, which was implemented using the UOT formulation with the standard cost, by more than 2%.

Table 1b presents the mean and standard deviation over three runs for the VisDA dataset. In this experiment, we integrate GOT into the POT problem formulation, using the TS implementation proposed by Nguyen et al. (2022a). Consequently, TS-POT serves as the baseline. DeepJDOT and JUMBOT were reproduced using

Table 3: Binary classification accuracy on two BCI datasets. Results for (a) the cross-subject task, and (b) the leave-one-subject-out task.

(a)

| Method | MI1 | MI2 | Avg |
|---|---|---|---|
| CSP-LDA | 57.23 | 58.70 | 57.97 |
| RA-MDRM | 64.98 | 66.60 | 65.79 |
| EA-CSP-LDA | 66.85 | 65.00 | 65.93 |
| MEKT-R | 70.99 | 68.73 | 69.86 |
| METL | _71.81_ | _69.06_ | _70.44_ |
| OT | 63.83 | 64.59 | 64.21 |
| OT-reg | 67.17 | 67.26 | 67.22 |
| GOT (ours) | **72.24** | **70.41** | **71.33** |

(b)

| Method | MI1 | MI2 | Avg |
|---|---|---|---|
| CSP-LDA | 59.71 | 67.75 | 63.73 |
| RA-MDRM | 69.21 | 70.91 | 70.06 |
| EA-CSP-LDA | 79.79 | 73.53 | 76.66 |
| MEKT-R | 83.42 | _76.31_ | _79.87_ |
| METL | 83.14 | 76.00 | 79.57 |
| WBT | _85.21_ | 74.31 | 79.76 |
| WBT + GOT (ours) | **87.29** | **77.24** | **82.26** |

the original DeepJDOT framework, while TS-POT was reproduced using the TS implementation. Additional details regarding the TS implementation and the parameters used for each method are available in Appendix C.3.3. We observe that employing the TS implementation with the POT formulation is highly effective for this dataset, as TS-POT obtained significantly higher accuracy than the other competing methods. Notably, incorporating GOT into this framework further enhances performance, increasing accuracy by nearly 3% on average. In Figure 11 in Appendix C.3.3, we analyze the t-SNE representation of the VisDA target features obtained from the deep model trained with GOT, comparing them to those learned using the baseline method, TS-POT. In Table 11 of Appendix C.3.3, we present the accuracy achieved by integrating GOT into a framework with a vision transformer (ViT) backbone, in comparison to state-of-the-art (SOTA) methods utilizing similar architectures.

## 5.3 Domain Adaptation for non-Euclidean Data

In this section, we depart from employing OT to learn domain-invariant features and instead directly utilize the optimal plan to transport source features into the target domain. We evaluate the proposed method on electroencephalogram (EEG) data, addressing both binary and multi-class classification tasks. Specifically, we apply our approach to two Motor Imagery (MI) benchmarks from the BCI Competition IV. The first dataset, dataset I from Blankertz et al. (2007), is referred to as MI1, and the second dataset, IIa from Tangermann et al. (2012), is referred to as MI2. We follow the methodologies of Barachant et al. (2011; 2013); Zanini et al. (2017); Rodrigues et al. (2018); Yair et al. (2019), and solve the DA problem on the (non-Euclidean) Riemannian manifold of Symmetric Positive Definite (SPD) matrices. For all experiments in this section, we utilize the entropy-regularized problem formulation (Cuturi, 2013), referring to our methods as GOT. The baseline method, which uses this formulation with the standard transport cost, is labeled as OT in the tables. Additionally, we include a comparison with OT-reg (Courty et al., 2016; 2014), a regularization approach that incorporates source labels into the problem formulation, which we consider as a secondary baseline. For detailed information on the framework, datasets, pre-processing, and parameters, please refer to Appendix C.4.

**Results.** To facilitate a meaningful comparison with leading methods, which are limited to two-class scenarios, we initially evaluate our proposed method on a **binary classification** task. For the evaluation, we utilize both BCI datasets; specifically, for the MI2 dataset, we select the left hand (class 1) and the right hand (class 2) as the two classes of interest. Our method is benchmarked against baseline algorithms commonly employed in BCI classification, namely CSP-LDA (Grosse-Wentrup & Buss, 2008), RA-MDRM (Zanini et al., 2017), and EA-CSP-LDA (He & Wu, 2019). Additionally, we compare our approach to two leading methods: MEKT (Zhang & Wu, 2020) and METL (Cai et al., 2022). Results for CSP-LDA, RA-MDRM, EA-CSP-LDA, and METL are taken from the tables in Cai et al. (2022), while results for MEKT are sourced from the original paper (Zhang & Wu, 2020).

Table 4: Multi-class classification accuracy on dataset MI2. (a) Results for the cross-session task. (b) Results for the cross-subject task.

(a)

| Method | 1 | 3 | 7 | 8 | 9 | Avg |
|---|---|---|---|---|---|---|
| OT | 81.08 | 84.03 | 78.12 | 77.26 | 68.06 | 77.71 |
| OT-reg | 80.9 | 85.59 | 78.99 | 83.33 | 75.17 | 80.8 |
| MADAOT | 68.75 | 71.53 | 70.14 | 72.22 | 65.8 | 69.69 |
| MLOT | _84.9_ | 84.72 | _83.33_ | 83.85 | _77.26_ | _82.81_ |
| RMLOT | 83.33 | 78.3 | 67.36 | 81.08 | 68.23 | 75.66 |
| GOT | 83.51 | **86.98** | 74.31 | **88.19** | **80.73** | 82.74 |
| MLOT+GOT | **86.63** | _86.28_ | **84.72** | _83.85_ | 75.52 | **83.4** |

(b)

| Method | 1 | 3 | 7 | 8 | 9 | Avg |
|---|---|---|---|---|---|---|
| OT | 55.38 | 62.74 | 57.53 | 56.25 | 52.28 | 56.84 |
| OT-reg | 54.36 | 62.48 | 57.25 | 59.09 | 54.62 | 57.56 |
| MADAOT | 54.41 | 59.79 | 54.97 | 57.29 | 53.49 | 55.99 |
| MLOT | _61.48_ | _70.01_ | _64.04_ | _66.43_ | _59.7_ | _64.33_ |
| RMLOT | 43.4 | 54.51 | 43.12 | 46.2 | 47.35 | 46.92 |
| GOT | **63.24** | **75.13** | **67.3** | **71.53** | **64.45** | **68.33** |

Table 3a presents the results for the cross-subject task. In this experiment, each subject, in turn, serves as the target domain, with the remaining subjects alternately acting as the source domain. Accuracy is computed by training a linear SVM classifier on the transformed source mappings and evaluating it on the target mappings. The results show the average target accuracy across source subjects, averaged over all subjects. The proposed method achieved the highest accuracy for both datasets, improving the baselines OT and OT-reg by 7% and 4%, respectively. Detailed subject-wise results appear in Table 12 and Table 13 in Appendix C.4.1, along with the experimental parameters.

In Table 3b, we present the results for the leave-one-subject-out task, where each subject alternately acts as the target domain while all others are used as source domains. We employ the multi-source OT-based DA framework from Montesuma & Mboula (2021), referred to as Wasserstein Barycenter Transport (WBT). The table reports the average target accuracy, computed by training a linear SVM on all the transported source representations and evaluating on target samples. Interestingly, the baseline method (WBT with the standard cost) outperformed all competing methods on dataset MI1. Notably, incorporating GOT resulted in further accuracy improvements, achieving the highest performance across both datasets. For details on the framework and the experimental parameters, see Appendix C.4.1.

Finally, we assess the performance of GOT in a **multi-class classification** task on dataset MI2. In addition to the baseline methods OT and OT-reg, we compare GOT to the competing methods MADAOT (Dhouib et al., 2020), MLOT (Kerdoncuff et al., 2020), and RMLOT (Jawanpuria et al., 2025). In Kerdoncuff et al. (2020), the authors propose to rely solely on source labels for parameter tuning. We follow a similar approach, and implement a more robust selection procedure applied consistently across all competing methods. Further details on the tuning procedure are provided in Appendix C.4.2.

Table 4a shows the results for the cross-session task using source-label tuning. In this experiment, for each subject, the first session (day) serves as the source domain, while the second session serves as the target domain, and vice versa. The reported target accuracy represents the average accuracy across both sessions, obtained by training a linear SVM on the mapped source samples. Following Zanini et al. (2017); Yair et al. (2019), we present results for five subjects out of the available nine, as the remaining subjects exhibited poor performance and were considered invalid. We observe that GOT achieves the highest accuracy for three out of the five subjects, improving upon the baseline OT by 5%. Its overall average is comparable to MLOT, with a difference of less than 0.1%. Notably, when tuning with target labels, GOT achieves the best performance overall (Table 19b in the appendices). A practical advantage of GOT is that it can be integrated into almost any OT-based method. Exploiting this property, we also present results for MLOT combined with GOT (denoted by "MLOT+GOT") for the cross-session task, which further improves performance.

Table 4b presents the cross-subject results, analogous to the binary classification results in Table 3a. Same as in the cross-session task, only the five valid subjects are considered. Here, GOT consistently achieves the highest accuracy across all subjects and yields an average accuracy that surpasses OT and OT-reg by more than 10%. It also outperforms all competing methods, with MLOT – the second-best method – lower by 4%. Comprehensive results for all nine subjects and all methods, under both tuning strategies and both tasks, are provided in Appendix C.4.2. For completeness, Appendix C.4.2 also includes results for the multi-class

Table 5: Multi-class classification accuracy (%) on the MI2 BCI dataset, averaged over all source–target experiments. Results are reported for two parameter tuning strategies: tuning based on source labels and tuning based on target labels.

| Tuning strategy | OT | OT-reg | MADAOT | MLOT | RMLOT | GOT (ours) |
|---|---|---|---|---|---|---|
| Source-label | 40.99 | 41.97 | 39.07 | 44.73 | 35.90 | **47.18** |
| Target-label | 44.20 | 49.47 | 43.33 | 52.85 | 50.63 | **55.00** |

classification task obtained using a single parameter configuration tuned on target labels across the entire dataset. Interestingly, these results are closer to those obtained with unsupervised tuning than to those from target-tuning per source–target pair.

In Table 5, we report average results for multi-class classification on the MI2 dataset, which includes a total of 306 cross-session and cross-subject experiments. The results show that GOT achieves the highest accuracy under both parameter tuning strategies: source-label tuning (unsupervised) and target-label tuning. Across all methods, we observe a clear gap between the two tuning strategies, indicating that selecting parameters solely based on source accuracy is suboptimal. We note that for methods with limited hyperparameters, such as OT and RMLOT, the selected parameters often coincide with those that maximize source accuracy (as originally done in Kerdoncuff et al. (2020)). This effect is particularly evident for RMLOT, which exhibits a gap of nearly 15% between source- and target-based tuning. In Appendix C.4.2, we provide detailed results for both tuning strategies and all methods.

# 6  Conclusions

In this work, we introduce a novel transportation cost for OT in DA that leverages diffusion geometry to capture both intra-domain structures and inter-domain relationships. By analyzing the asymptotic behavior of the proposed diffusion operator $\mathbf{S}$, we identified the key factors influencing the anisotropic diffusion process and demonstrated how it aligns with the goals of DA. Motivated by the manifold hypothesis – that high-dimensional data often lie on lower-dimensional manifolds – our approach exploits local neighborhood information to define a geometry-aware cost. Intuitively, such costs are better suited to capture meaningful relationships between samples than purely Euclidean distances, particularly in real-world domains. While there is no theoretical guarantee that GOT will always outperform traditional OT costs, our experimental results show that GOT consistently achieves superior performance across various UDA tasks, datasets, and frameworks. That said, GOT introduces three parameters related to the Gaussian kernel scale, and poor choices can lead to degraded performance. We therefore view GOT as a strong, robust, and broadly applicable choice for OTDA when accompanied by careful parameter tuning.

**Limitations and future work.** We acknowledge a fundamental limitation of our method: since it incorporates inter-domain relationships, it operates under the assumption that both source and target domains reside in the same space. While this assumption is common to all DA methods, we believe that GOT has the potential to be extended to other applications, which we view as directions for future work. In addition, the GOT cost is not a distance metric, as it lacks symmetry and does not satisfy the triangle inequality. Consequently, the OT solution does not define a valid distance (unlike the Wasserstein distance computed using the Euclidean cost). While this limitation did not affect the DA tasks we considered – where only the transport plan is needed – adapting the cost to satisfy metric properties could greatly expand its utility. Finally, we note that recent SOTA methods for DA, which we did not include in our comparisons, are based on transformer architectures. While OT-based methods, including ours, may not naturally fit into the transformer framework, we believe they remain valuable – particularly in scenarios where deep learning is impractical, or when explicit modeling of geometric relationships is important.

### Acknowledgments

We thank the anonymous reviewers for their insightful feedback. This work was supported by the European Union's Horizon 2020 research and innovation programme under grant agreement No. 802735-ERC-DIFFOP.

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

## Supplementary Materials

## A  Background on Diffusion Geometry

Let $\{x_i\}_{i=1}^N$ be a given set of data samples on a hidden manifold $\mathcal{M}$ with a metric $g$. Constructing a graph is a common strategy for approximating the manifold's structure (Roweis & Saul, 2000; Tenenbaum et al., 2000; Belkin & Niyogi, 2003; Coifman & Lafon, 2006). By leveraging a local kernel, which quantifies pairwise local similarities between data samples, this graph captures local neighborhood relationships, facilitating effective representation of the underlying manifold geometry. Consider a weighted graph $G = (V, E)$ with a vertex set $V = \{v_1, \ldots, v_N\}$, where each node $v_i \in V$ corresponds to the sample $x_i$. Let $\mathbf{K} \in \mathbb{R}^{N \times N}$ be an affinity matrix, whose $(i, j)$-th element encodes the weight of the edge between nodes $v_i$ and $v_j$ and is given by:

$$\mathbf{K}(i, j) = h\left(\frac{d_g^2(x_i, x_j)}{\epsilon}\right), \tag{13}$$

where $d_g(\cdot, \cdot)$ is a distance function induced by the metric $g$, $\epsilon > 0$ is a scale hyperparameter, and $h$ is a positive function with exponential decay.

Next, the affinity matrix $\mathbf{K}$ is normalized to be row-stochastic:

$$\mathbf{P}(i, j) = \frac{\mathbf{K}(i, j)}{d(i)}, \tag{14}$$

where $d(i) = \sum_j \mathbf{K}(i, j)$. The matrix $\mathbf{P}$ is often viewed as a diffusion operator (Coifman & Lafon, 2006), as $\mathbf{P}(i, j)$ defines a transition probability from node $i$ to node $j$ of a Markov chain on the graph. Analogously, $\mathbf{P}$ can be viewed as the transition probabilities of a diffusion process defined at the data samples $\{x_i\}_{i=1}^N$ on the manifold $\mathcal{M}$ (Coifman & Lafon, 2006).

## B  Geometric Optimal Transport – Additional Information

### B.1  Computational Complexity of GOT

The most commonly used transport cost, the pairwise squared Euclidean distances, has a computational complexity of $\mathcal{O}(n^2)$, assuming balanced datasets. In contrast, our proposed cost involves the following steps:

- Computing pairwise distances for three matrices: $\mathcal{O}(n^2)$.

- Applying the exponential function to the matrices: $\mathcal{O}(n^2)$.

- Normalizing the matrices to be row-stochastic: $\mathcal{O}(n^2)$.

- Multiplying the three probability matrices: $\mathcal{O}(n^3)$.

- Applying the logarithm to the final diffusion operator: $\mathcal{O}(n^2)$.

As noted in Section 5, we often use doubly-stochastic normalization instead of row-stochastic normalization, employing the Sinkhorn algorithm. Since the complexity of Sinkhorn is also $\mathcal{O}(n^2)$, this does not increase the overall complexity. In total, the computational complexity of the proposed cost is $\mathcal{O}(n^3)$. While higher than the traditional cost, our method avoids the need for a regularization term, which can often add to the optimization complexity.

Additionally, we note that up to the fourth step, the computation involves three $n \times n$ matrices rather than one, resulting in higher memory requirements.

**Complexity comparison.** While the standard Euclidean cost, which involves computing pairwise distances between source and target samples, has complexity $\mathcal{O}(n^2)$, obtaining the transport plan depends on the specific optimal transport formulation. Solving the original OT problem requires a computational cost of $\mathcal{O}(n^3)$, while using the Sinkhorn algorithm reduces the complexity to $\mathcal{O}(n^2)$. OT-reg, used in Sections 5.1 and 5.3, relies on Sinkhorn iterations for the optimization and, therefore, has complexity $\mathcal{O}(n^2)$, whereas OT-Laplace requires $\mathcal{O}(n^3)$. In Figure 10a, we report the empirical run-time for a toy example, comparing GOT with OT, OT-reg, and OT-Laplace. The results illustrate the distinction between complexity arising from computing the cost matrix and complexity arising from the optimization process (for example, due to added regularization terms). Notably, although both GOT and OT-Laplace have a theoretical complexity of $\mathcal{O}(n^3)$, OT-Laplace is considerably slower in practice. GOT, which leverages the Sinkhorn algorithm, not only outperforms OT-Laplace in speed but also converges in far fewer iterations than Sinkhorn with the standard Euclidean cost. As a result, the total run-time of GOT can be faster than OT-reg, despite OT-reg having a lower theoretical complexity of $\mathcal{O}(n^2)$.

The competing methods ETD and RWOT introduce a weighted Euclidean distance as the transport cost and differ in how they learn the weights. Both methods are deep-learning-based, and even when disregarding the complexity introduced by the neural networks, their computational cost remains at least $\mathcal{O}(n^3)$ due to the matrix multiplications between the learned weight matrix and the pairwise distance matrix. Our method, GOT, also has complexity $\mathcal{O}(n^3)$ for the same reason – matrix multiplication. However, unlike ETD and RWOT, GOT is not restricted to deep-learning frameworks, making it more memory-efficient. Moreover, as shown in Section 5.2, GOT achieves superior performance while maintaining a comparable computational cost.

## B.2 The Diffusion Process – Simulation

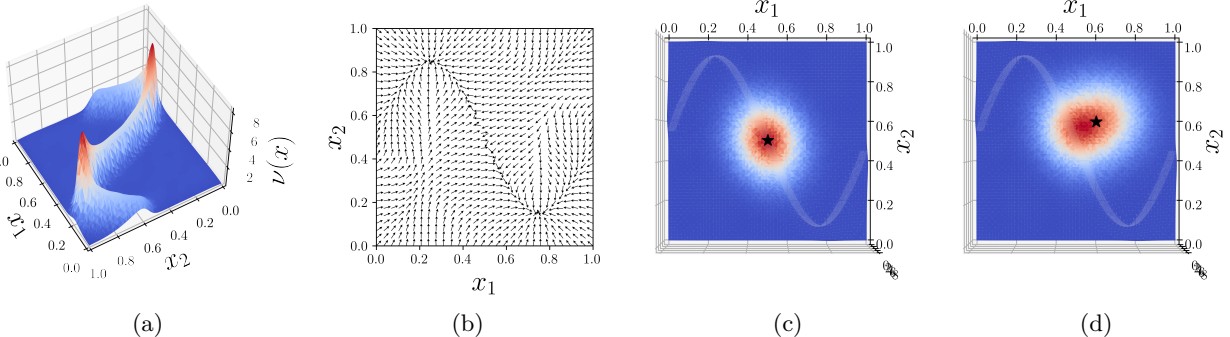

|  (a)  |  (b)  |  (c)  |  (d)  |

Figure 3: (a) The PDF of the target domain, colored by PDF values. (b) Visualization of the vector field induced by the proposed diffusion operator. (c) and (d) show the PDF of the target domain projected to 2D, colored by the proposed diffusion process initiated at high-density and low-density regions, respectively. (Best viewed in colors).

In this section, we illustrate the behavior of the proposed diffusion operator, and more specifically, the expression outlined in Proposition 4.1, through a toy example. Consider the 2D space $[0,1]^2$, where the source domain $\mu$ is represented by a uniform distribution over the entire space, and the target domain $\nu$ is concentrated along a 1D contour. Specifically, 20% of the target data are sampled from a uniform distribution over $[0.1]^2$. For the remaining 80% of the target samples, the coordinates are generated as follows. The first dimension $x_1$ is computed as the sum of two variables sampled from uniform distributions: $u_1 \in U[0.02, 0.98]$ and $u_2 \in U[-0.02, 0.02]$, giving $x_1 = u_1 + u_2$. The second dimension $x_2$ is generated with $x_2 = 0.4 \sin(2\pi u_1) + 0.5$, where $u_1$ is the same variable used for $x_1$. Figure 3a presents the target domain, estimated using Kernel Density Estimation (KDE).

To illustrate the behavior of the proposed operator, we compute the diffusion operator **S**, which contains the probabilities of transporting from the uniform source samples to the target samples concentrated on the contour, as presented in Equation 6. Figure 3b visualizes the resulting vector field on the unit square $[0,1]^2$,

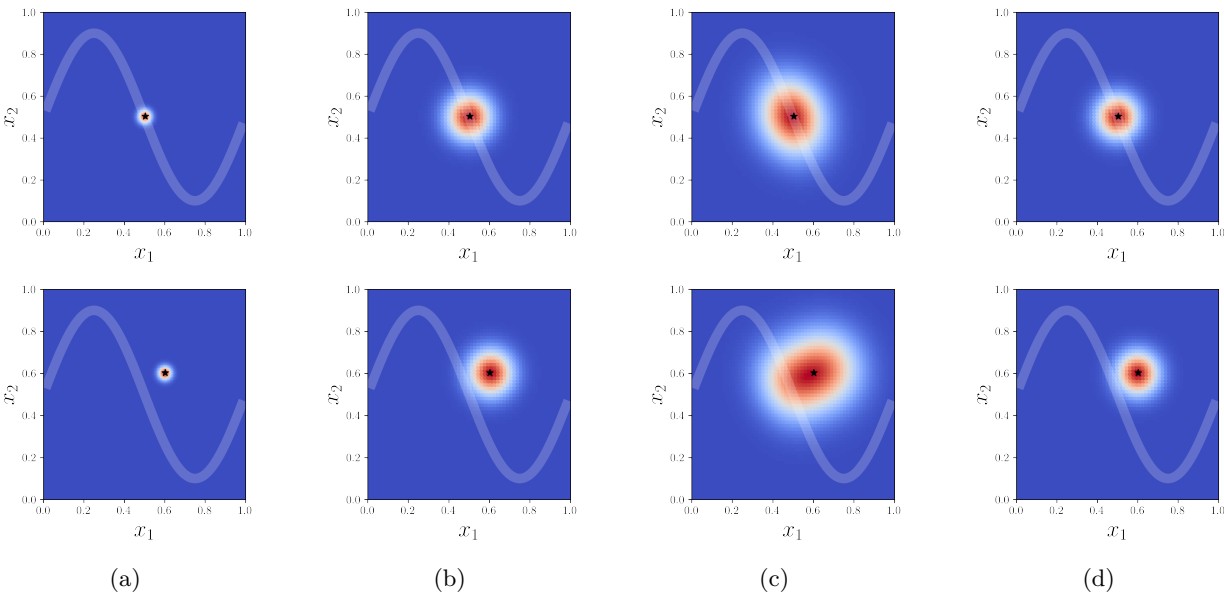

Figure 4: The 2D space colored by the diffusion induced by (a) the source operator $P_s$, (b) the composed operator $QP_s$, and (c) the final operator $P_sQP_t$. (d) shows the diffusion induced by applying only the cross operator $Q$, analogous to using the Euclidean cost. In the first row, diffusion starts from a source sample located in a high-density target region, and in the second row, it starts from a low-density target region.

as induced by **S**. As expected, we observe that for all source samples, the average direction induced by the diffusion operator is toward the 1D contour. Figures 3c and 3d present two specific cases that illustrate how the anisotropic diffusion balances inter-domain and intra-domain relationships. In Figure 3c, the diffusion process is initiated at a source sample located on the contour, a region of high target probability density, marked by the black star. As the source distribution is uniform, by definition, the value of $\frac{\mu}{\nu}(x)$ at this initial sample is small, indicating a high probability that this source sample belongs to the target domain. Here, the diffusion assigns more weight to the **geometry of the target domain**, spreading rapidly along the contour where the target density is highest. In contrast, Figure 3d shows the diffusion process starting from a source sample located off the contour in a region with a sparser target distribution. Similar to the previous case, we observe an anisotropic diffusion directed toward regions with high probability density. However, due to the location of the starting point, the diffusion first draws it toward the denser region, utilizing **inter-domain relations**, and resulting in a noticeable asymmetry in the diffusion process.

For further intuition, Figure 4a-c illustrates the space after each diffusion step applied to the two points. Figure 4d shows the diffusion obtained by applying only the cross-domain operator $Q$, which is equivalent to using the traditional cost that considers only inter-domain Euclidean distances. Since the source domain is uniform and source labels are not incorporated into the operator, the effect of $P_s$ is not apparent in this case. Importantly, it is only after applying $P_t$ that the diffusion spreads into the dense regions of the target, thereby accounting for the geometry of the target domain.

### B.3 Illustration of Proposition 4.1

In Figure 5, we illustrate the analysis of Proposition 4.1 using a simple example. The source distribution is a Gaussian centered at $(2, 2)$, while the target is a mixture of two Gaussians centered at $(-2, -1)$ and $(2, -1)$. Figures 5a-5d present the source $\mu$ and target $\nu$, along with their normalized Laplacians, $\Delta\mu/\mu$ and $\Delta\nu/\nu$, respectively.

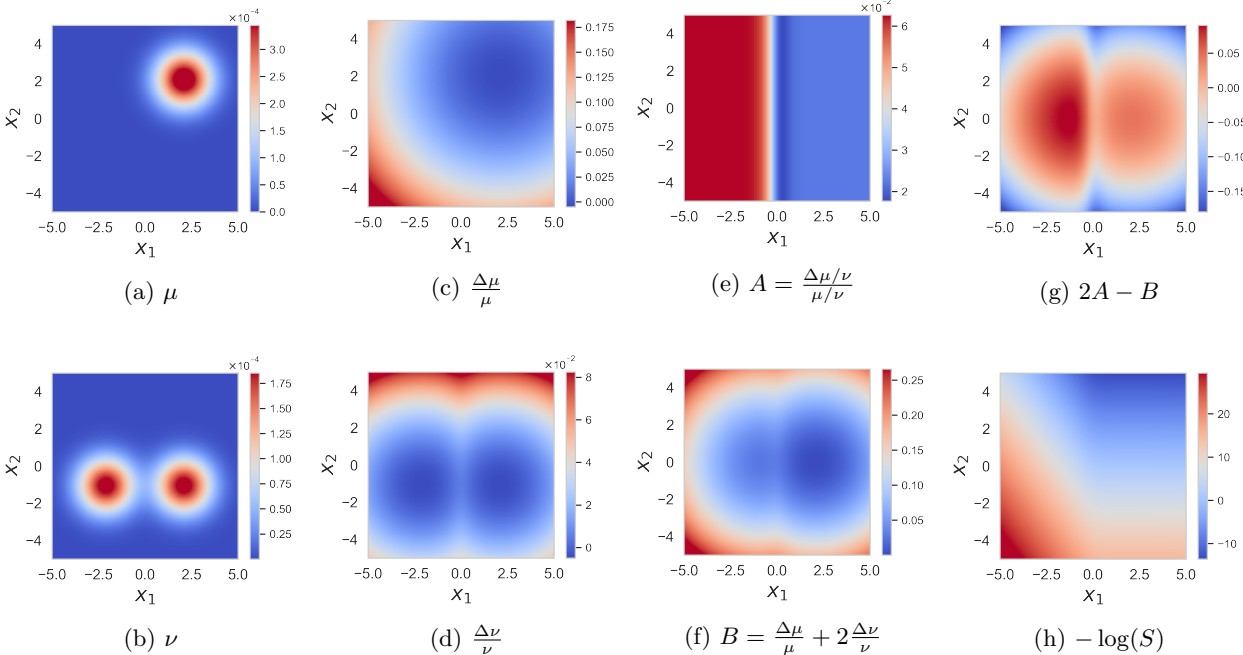

Figure 5: Illustration of Proposition 4.1. The source $\mu$ is a Gaussian and the target $\nu$ is a mixture of two Gaussians, shown in (a) and (b). Panels (c) and (d) display the normalized Laplacians, (e) and (f) show the terms $A$ and $B$ as defined in Equation 15, (g) shows the combined effect $2A - B$, and (h) shows the final diffusion cost. In (e), (g), and (h), the quantities enter the transport objective with a positive sign and are therefore minimized (blue regions are favored for transport), while in (f), the term enters with a negative sign and is effectively maximized (red regions are favored).

For simplicity, we consider the constant function $f = 1$ and assume $m_2/m_0 = 1$, resulting in the simplified asymptotic expression:

$$S_\epsilon 1 = \frac{\mu}{\nu}(x)\Big[1 - \epsilon(2A - B)\Big](x) + O(\epsilon^2), \tag{15}$$

where $A = \frac{\Delta(\mu/\nu)}{\mu/\nu}$ and $B = \frac{\Delta\mu}{\mu} + 2\frac{\Delta\nu}{\nu}$.

Figure 5e shows $A$, revealing a clear split along the $y$-axis: values are high on the left and low on the right. This asymmetry occurs because the source distribution is concentrated on the right, while the target is dense on both sides due to the symmetric Gaussians. Since $A$ favors matching dense-to-dense and sparse-to-sparse regions, this implies that source samples from the right side, where $\mu$ is dense, are transported toward the left, where $\nu$ is also dense. Note that the source and target geometry is not visible when considering $A$ alone. Figure 5f shows $B$, highlighting two Gaussian components: values are low at the centers and increase toward the boundaries. The right Gaussian has slightly lower values than the left, reflecting the overlap between the source and target distributions in that region. At the boundaries, where densities are sparse and vary rapidly, $B$ reaches its maximum. Due to the negative sign of $B$ in the cost, samples with higher values – mostly outside the Gaussians – receive greater weight in the transport. Figure 5g shows $2A - B$, illustrating the combined effect. The two Gaussians reappear with higher overall values, with the left Gaussian being larger and stronger. Moving outward from the Gaussians, the values decrease, indicating that the transport maps source samples from the surrounding regions into the two target clusters, with a stronger emphasis on the left Gaussian. Finally, Figure 5h shows the final diffusion cost $C = -\log(S)$ for $S$ in Equation 15 with $\epsilon = 1$. Minimizing this cost transports samples primarily from the top-right region – the dense source area – to the bottom region containing the two target Gaussians, focusing mostly on the left cluster.

## C   Experimental Details

### C.1   Hyperparameter selection in GOT

#### C.1.1   The Gaussian scale

The Gaussian kernel is known to be highly sensitive to its scale parameter. When the scale is too large, the kernel fails to distinguish between samples, leading to a loss of important information. Conversely, when the scale is too small, the kernel becomes overly sensitive to noise, which may result in overfitting. Moreover, the choice of scale is inherently dependent on the data scale. There is no single optimal methodology for selecting the Gaussian kernel parameter. Moreover, determining the appropriate kernel scale is an open problem that remains an active area of research across various fields, beyond OT and domain adaptation. A common practice is to set it as the median of the pairwise distances in the dataset. We followed this approach consistently in all our experiments, as stated at the beginning of Section 5. In the deep experiments, we fixed the Gaussian scale to $\epsilon_s = 1, \epsilon_c = 1, \epsilon_t = 1$ and applied only the median normalization. This choice proved robust in practice, though further tuning could potentially yield even better performance. For the remaining experiments, in addition to the median normalization, we tuned the parameters $\epsilon_s, \epsilon_c, \epsilon_t$.

To further evaluate the effect of the Gaussian scale, we conducted a sensitivity analysis on the MI2 multi-class classification task. We fixed $\lambda = 0.01$ and varied one of $\{\epsilon_s, \epsilon_c, \epsilon_t\}$ at a time while keeping the others fixed to 1, after applying median normalization. Figure 6 reports results for both cross-subject and cross-session adaptation, for all subjects and for the five valid subjects. The analysis confirms that $\epsilon_s$ and $\epsilon_c$ are indeed sensitive to the choice of scale: both very small and very large values lead to performance degradation. By contrast, $\epsilon_t$ yields relatively stable results, which is expected given that the neighborhood structure is already constrained by the choice of three neighbors. Importantly, setting all scales to 1 (the choice we adopted in the deep experiments) provides a robust middle ground, consistently yielding reasonable accuracy even if not always optimal.

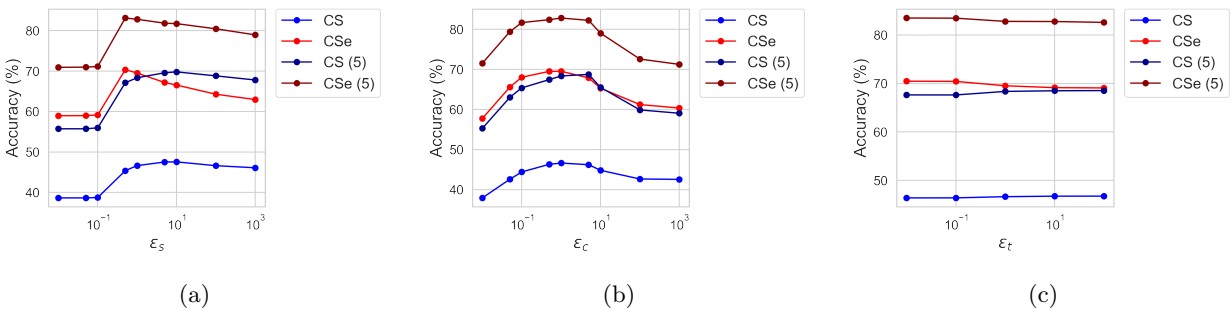

Figure 6: Ablation study on the scale parameters (a) $\epsilon_s$, (b) $\epsilon_c$, and (c) $\epsilon_t$ (log scale) in the BCI multi-class classification task. Results are shown for Cross-subject (CS), Cross-session (CSe), Cross-subject with 5 valid subjects (CS-5), and Cross-session with 5 valid subjects (CSe-5).

#### C.1.2   The number of neighbors

The neighborhood of the kernel is controlled by the scale parameter $\epsilon$. For the source and target operators, we further refine the neighborhoods by either incorporating label information (when available) or selecting the $K$ nearest neighbors. In all our experiments, we fixed $K = 3$, and found this choice to be empirically robust. While tuning $K$ could potentially improve performance, its influence was less pronounced compared to other parameters, so we opted to keep it fixed.

Figure 7 illustrates the impact of $K$ on target accuracy in a set of BCI multi-class experiments. We fixed the scale parameters ($\epsilon_s = 3, \epsilon_c = 0.1, \epsilon_t = 0.1$) and varied only the number of target neighbors. With joint parameter tuning, different values of $K$ could potentially yield better results. Dataset details appear in Appendix C.4, and the multi-class classification setup is described in Appendix C.4.2.

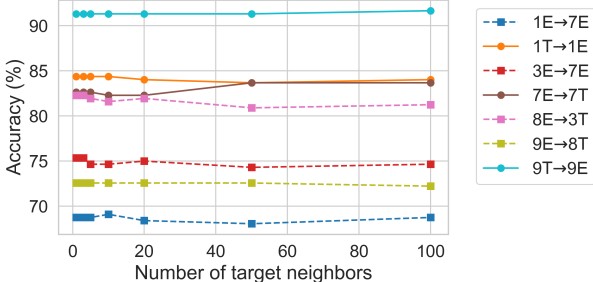

Figure 7: Effect of the number of nearest neighbors on target accuracy for multi-class experiments on the MI2 BCI dataset.

### C.1.3   Hyperparameter tuning

For the reproduced results of the competing methods, we followed the hyperparameter settings reported in their respective papers, with one exception – DeepJDOT. The reported parameters for DeepJDOT led to poor results in our setting, so we re-tuned them ourselves.

Our method, GOT, requires tuning four parameters: the diffusion operator parameters $\epsilon_s, \epsilon_c, \epsilon_t$ and the entropy regularization weight $\lambda$ (due to the use of the Sinkhorn algorithm). While target labels were not used during training, we did rely on them as a validation set for hyperparameter tuning, consistent with competing methods. Importantly, we optimized parameters for each dataset as a whole, rather than for each individual source-target pair. For instance, the same set of parameters was used across all multi-class experiments on the BCI dataset (MI2), as presented in Appendix C.4.2. For reproducibility, the parameters used for each dataset and each experiment are detailed in this appendix in the respective sections. Additionally, in Section 5.3 and Appendix C.4.2, we present results obtained using an unsupervised tuning approach. Notably, these results are very similar to those obtained using a single set of parameters for the entire dataset, tuned with target labels.

### C.2   Two Moons: Visual Illustration

In this experiment, we analyze the resulting transportation plan and evaluate the performance of GOT compared to competing methods using the toy example from Courty et al. (2016). For the source domain, we generate two entangled moons, each containing 50 samples, with Gaussian noise added (standard deviation 0.05) to represent two distinct classes. The target domain is independently sampled from a rotated version of this distribution. As the rotation angle increases, the adaptation problem becomes more challenging.

We compare GOT with the entropy-regularized OT problem (Courty et al., 2016; Cuturi, 2013), referred to as OT, which serves as the baseline method. Additionally, due to the limited number of methods proposing alternative transport costs – particularly those that integrate source label information and can be applied in non-deep frameworks – we include two regularization approaches: OT-reg (Courty et al., 2016; 2014), which incorporates source label information, and OT-Laplace (Courty et al., 2016; Flamary et al., 2014), which uses a Laplace regularization term to preserve data structure. Notably, all three methods use the squared Euclidean distance as the transportation cost. Table 6 summarizes the key differences between GOT and these three competing methods. While the first two characteristics in the table are straightforward, the last two may require further clarification. OT-Laplace's optimization relies on simplifying the regularization term under the assumption of Euclidean geometry. For non-Euclidean geometries, the intra-domain distances, inter-domain distances, and barycentric mapping become too complex to optimize, limiting it to Euclidean settings. While the squared Euclidean distance is commonly used as the transport cost in OT and OT-reg, both methods can handle non-Euclidean geometries, as demonstrated in Section 5.3. The specific optimization processes proposed for OT-reg and OT-Laplace restrict their applicability to other formulations or solvers. In contrast, our GOT, which is a new transport cost rather than a regularization term, is versatile and applicable to any problem formulation or solver, as shown in Section 5.2.

Table 6: Summary of key differences between GOT and competing methods.

|  | OT | OT-reg | OT-Laplace | GOT (ours) |
|---|---|---|---|---|
| Intra-domain relationship integration |  |  | ✓ | ✓ |
| Label information utilization |  | ✓ |  | ✓ |
| Supports non-Euclidean geometry | ✓ | ✓ |  | ✓ |
| Versatile OT compatibility | ✓ |  |  | ✓ |

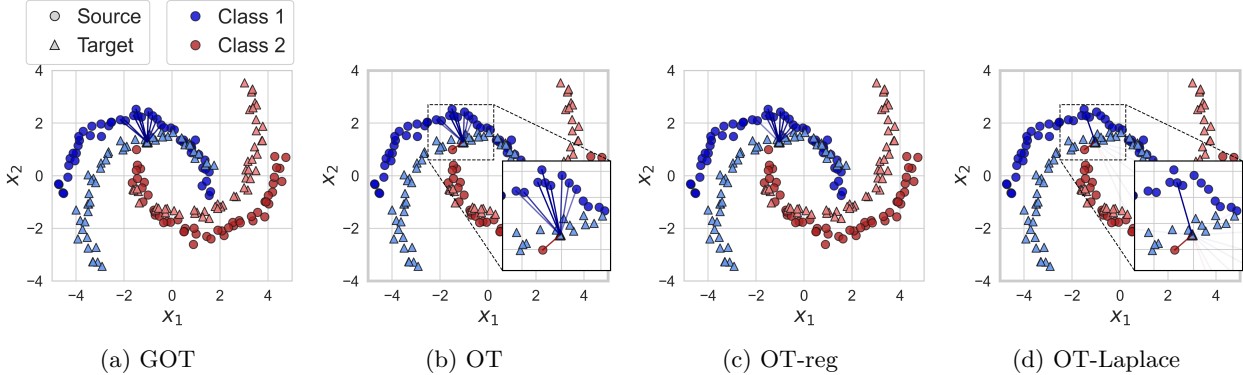

Figure 8: Two moons illustration for a 40° rotation angle. Source samples are marked with circles, target samples with triangles, and the colors of the markers indicate the labels. Each line represents an entry $\gamma(i,j)$ in the transportation plan, where the $j$-th sample is marked by a bold black triangle. For clarity, we plot lines from this sample to the ten source samples with the highest values in the $j$-th column of the plan. The line color corresponds to the label of the source sample $x_i^s$, and the line intensity reflects the values in the plan. (Best viewed in colors).

To analyze the differences between the methods, we visualize the obtained plans for a 40° rotation angle, focusing on two arbitrary target samples that demonstrate the typical advantages of GOT over the other methods. We obtain the optimal plan for GOT and the other three methods as follows. For GOT, we first compute the proposed cost and then use the entropy-regularized OT problem (Cuturi, 2013) to derive the transportation plan $\gamma$. For the other methods, we calculate the standard transportation cost – the squared Euclidean pairwise distances – and then solve the OT problem using entropy regularization, label regularization, and Laplace regularization to obtain the transportation plans for OT, OT-reg, and OT-Laplace, respectively.

Figure 8 presents the plan values for a target sample located in the middle of the blue moon, marked by a bold black triangle. For clarity, we plot lines from this target sample only to the ten source samples with the highest values in the corresponding column of the plan. In GOT (Figure 8a), we observe the desired optimal plan, where all lines connected to the chosen target sample originate from the same blue class, even though a red-labeled source sample is closer. This outcome is due to incorporating label information into the diffusion process. In OT (Figure 8b), the transportation cost is based solely on the pairwise Euclidean distances between source samples (circles) and target samples (squares). As a result, the blue-labeled target sample receives mass from the closest source samples, including those associated with the other red class. In OT-reg (Figure 8c), the regularization term promotes group sparsity, encouraging a solution where each target sample receives mass only from source samples within the same class. As a result, this method also provides the desired plan in this case. In OT-Laplace (Figure 8d), the core idea is that samples with small Euclidean distances before transportation should remain close after transportation, and vice versa. As a result, the blue-labeled target sample receives mass not only from the nearby red-labeled source sample, but also from a more distant blue-labeled source sample. We note that there are fewer than ten lines because this method does not include entropy regularization, resulting in a sparser transportation plan.

**Cost analysis.** To illustrate the connection between the diffusion process and the resulting transportation plan, we present in Figure 9a the average direction of transport from each target point to the source domain, as induced by the diffusion operator **S**, for a 70° rotation angle. Although the standard OT cost lacks a probability notion, we present a comparable visualization in Figure 9b, by applying the exponential function to the negative cost matrix and normalizing the resulting matrix to be column-stochastic. The diffusion induced by the cost aligns with the resulting plan values depicted in Figures 2 and 8. For example, all target points within the blue dashed ellipse in Figure 9a are directed toward the correct red moon of the source. Conversely, in Figure 9b, most target points are directed toward the closer but incorrect blue moon of the source. Similarly, all target points within the red dashed ellipse in Figure 9a, including the chosen point shown in Figure 8, are directed toward the correct blue moon of the source, even though closer source samples, labeled red, are present. It is worth noting that, while the presented directions correspond to the target samples, the GOT cost relies solely on source label information. In contrast, in Figure 9b, the same target points are directed incorrectly, with many directed toward the wrong red source samples. For clarity, only these two areas are highlighted in the figures; however, additional regions, such as the bottom-left edge of the blue target moon and the center of the red target moon, could also be analyzed similarly. This example demonstrates how the proposed method effectively controls the resulting transportation plan by guiding the diffusion process toward the desired directions.

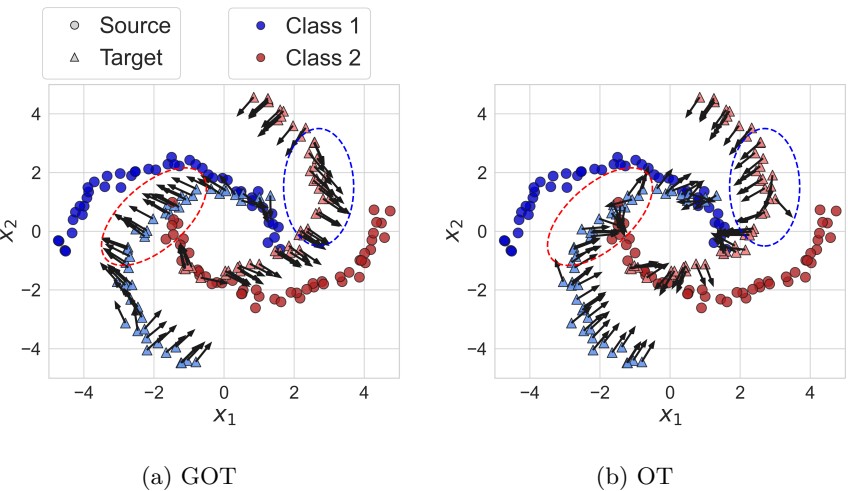

(a) GOT                        (b) OT

Figure 9: Visualization of the vector field induced by the GOT cost and the standard OT cost, for a 70° rotation angle. (Best viewed in colors).

**Classification results.** In addition to the visual illustration, we evaluate the performance of the methods across various rotation angles. Table 7 presents the target classification error rates for these different angles. For the evaluation, we follow the OTDA framework (Courty et al., 2016). This involves computing the optimal plan using GOT or one of the competing methods, transporting the source samples to the target domain via barycentric mapping, and then training an SVM classifier with a Gaussian kernel on the source data. The classifier is subsequently tested on the target samples. The classification results are consistent with the previous analysis. We observe that GOT is the only method that achieves 100% accuracy up to a 60° rotation angle. Furthermore, it is evident that GOT outperforms all other methods with significantly higher accuracy across all tested angles.

**Computational time comparison.** We compare the run-time of the methods, measuring both the time to compute the cost matrix $C$ (Euclidean or GOT) and the time to obtain the transport plan $\gamma$ under each optimization approach. Figure 10a reports the average results over 100 runs for varying number of samples. As described in Section B.1, computing the GOT cost has complexity $\mathcal{O}(n^3)$, whereas computing the standard Euclidean cost has complexity $\mathcal{O}(n^2)$. Regarding the optimization process, OT and GOT are solved using the efficient Sinkhorn algorithm with complexity $\mathcal{O}(n^2)$. OT-reg is solved via an iterative procedure that calls Sinkhorn, with complexity $\mathcal{O}(kn^2)$, where $k$ is the number of outer iterations. OT-Laplace is solved

Table 7: Two moons data. Target error rates for different rotation angles, obtained by training an SVM classifier with a Gaussian kernel on the transported source data.

| rotation angle | GOT (ours) | OT | OT-reg | OT-Laplace |
|:---:|:---:|:---:|:---:|:---:|
| 10° | 0 | 0 | 0 | 0 |
| 20° | 0 | 0 | 0 | 0 |
| 30° | 0 | 0 | 0 | 0.01 |
| 40° | 0 | 0.06 | 0.09 | 0.14 |
| 50° | 0 | 0.14 | 0.14 | 0.16 |
| 60° | 0 | 0.18 | 0.2 | 0.22 |
| 70° | 0.07 | 0.23 | 0.3 | 0.31 |
| 80° | 0.23 | 0.34 | 0.34 | 0.34 |
| 90° | 0.33 | 0.37 | 0.39 | 0.42 |

using a gradient-based algorithm, with complexity $\mathcal{O}(n^3)$. As expected, OT is the fastest method in practice. OT-Laplace, due to its expensive optimization, is substantially slower than the other methods. Interestingly, OT-reg is slightly slower than GOT, despite using the Euclidean cost. This can be explained by the number of iterations: as shown in Figure 10b, OT-reg always runs the maximum number of iterations (200 inner Sinkhorn iterations and 10 outer iterations – the default in the POT package (Flamary et al., 2021)), whereas Sinkhorn with the GOT cost converges after a small number of iterations (far fewer iterations than OT – which runs Sinkhorn with the Euclidean cost). OT-Laplace also typically reaches its default maximum of 101 iterations. These results suggest that although computing the GOT cost is more expensive than the Euclidean cost, its compatibility with a wide range of OT solvers and its ability to accelerate convergence could make it empirically faster in practice, even compared to methods with lower theoretical complexity.

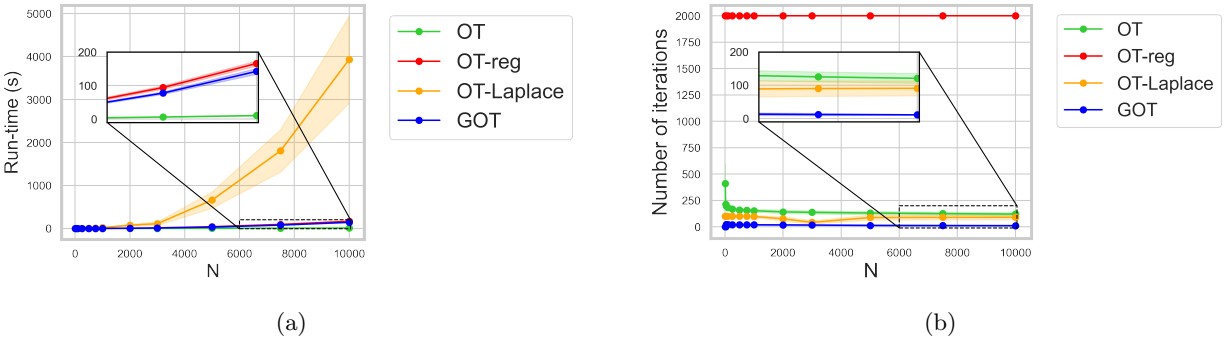

(a)                                           (b)

Figure 10: Comparison of (a) run-time and (b) iteration count over 100 runs on the two moons dataset with a 40° rotation, evaluated over varying numbers of samples.

### C.3 Deep Domain Adaptation

In the following section, we outline the framework utilized in the experiments presented in Section 5.2. For these experiments, we adopt the framework introduced by Damodaran et al. (2018). This framework, termed DeepJDOT, comprises two components: an embedding function, denoted by $g : \mathcal{X} \to \mathcal{Z}$, which aims to learn features optimizing both classification accuracy and domain invariance, and a classification function, denoted by $f : \mathcal{Z} \to \mathcal{Y}$, which predicts the labels based on the learned features.

The DeepJDOT objective, which operates on a minibatch of size $m$, is given by:

$$\min_{\gamma, f, g} \sum_{i}^{m} \mathcal{L}(y_i^s, f(g(x_i^s))) + \sum_{i,j}^{m} \gamma_{i,j} \left( \eta_1 \| g(x_i^s) - g(x_j^t) \|_2^2 + \eta_2 \mathcal{L}(y_i^s, f(g(x_j^t))) \right), \quad (16)$$

where $\mathcal{L}$ is the cross-entropy loss, $\eta_1$ and $\eta_2$ are hyperparameters, and $\gamma$ is the optimal plan.

The training process follows an alternating optimization scheme. Initially, the parameters of both the feature extractor $g$ and the classifier $f$ are fixed. The fixed-parameter models are denoted as $\hat{f}$ and $\hat{g}$. Subsequently, the OT problem is solved with the associated cost matrix $\mathbf{C}_{i,j} = \eta_1 \|\hat{g}(x_i^s) - \hat{g}(x_j^t)\|_2^2 + \eta_2 \mathcal{L}(y_i^s, \hat{f}(\hat{g}(x_j^t)))$. Finally, the optimization proceeds by fixing the obtained plan, $\gamma$, and optimizing the model on the mini-batch using Stochastic Gradient Descent (SGD). The objective function for this training phase is defined by Equation 16.

In our experiments, we propose to derive the plan using the GOT cost. We solve the OT problem with the associated cost $\mathbf{C}_{i,j} = -\eta_1 \log(\mathbf{S}_{i,j}) + \eta_2 \mathcal{L}(y_i^s, \hat{f}(\hat{g}(x_j^t)))$, where:

$$\mathbf{S}_{i,j} = \sum_{k|y_k^s = y_i^s} \sum_{l \in \mathcal{N}_j^t} \mathbf{P}_s(i,k)\mathbf{Q}(k,l)\mathbf{P}_t(l,j). \tag{17}$$

The probability matrices $\mathbf{P}_s, \mathbf{Q}$ and $\mathbf{P}_t$, are obtained by applying a row-stochastic normalization to the kernels defined in Equations 5 and 11. However, in this case, the kernels are applied in the latent space, using the source and target representations $\{\hat{g}(x_i^s)\}_{i=1}^m$ and $\{\hat{g}(x_j^t)\}_{j=1}^m$, where $m$ denotes the batch size. For the source and target kernels, we compute the kernels by employing the Cosine distance, while for the cross-domain we use the Euclidean distance. $\mathcal{N}_j^t$ denotes the neighborhood of $x_j^t$, determined by the three nearest neighbors in the features space. $\epsilon_s, \epsilon_c$ and $\epsilon_t$ are scale hyper-parameters. In all deep experiments, we set $\epsilon_s = \epsilon_c = \epsilon_t = 1$.

Once the plan is obtained, we fix $\hat{\gamma}$ and train $f$ and $g$ using Equation 16. We note that for simplicity in handling derivatives, we omit the use of GOT in the second phase of training, and utilize Equation 16 as is, although integrating the GOT cost remains a viable option.

**Baselines and competing methods.** In addition to DeepJDOT (Damodaran et al., 2018), which utilizes the framework with the standard transport cost, we present comparisons to more recent methods that leverage the DeepJDOT framework. However, unlike DeepJDOT, which employs the network simplex flow algorithm, these approaches utilize different OT problem formulations: Unbalanced OT (UOT), referred to as JUMBOT (Fatras et al., 2021), and Partial OT (POT), termed as m-POT (Nguyen et al., 2022a), to derive the optimal plan. Notably, all three methods use the squared Euclidean distance as the transportation cost and differ solely in their approach to obtaining $\gamma$. We also compare our approach to two other OT-based deep models, which propose to employ a different transportation cost function from the conventional squared Euclidean distance, namely ETD (Li et al., 2020) and RWOT (Xu et al., 2020). In addition to OT-based methods, we benchmark our method against established DA baselines, including DANN (Ganin et al., 2016) and ALDA (Chen et al., 2020). For DeepJDOT (Damodaran et al., 2018), JUMBOT (Fatras et al., 2021), and m-POT (Nguyen et al., 2022a), all employing the DeepJDOT framework with different OT problem formulations, we reproduce the results ourselves using the POT (Python Optimal Transport) package (Flamary et al., 2021), an open-source Python library. Results for DANN (Ganin et al., 2016) and ALDA (Chen et al., 2020) are sourced from Nguyen et al. (2022a). For the remaining methods, we rely on the reported results from their respective papers.

**Datasets.** The **Digits** dataset comprises three widely-used digit datasets, serving as the different domains: (i) MNIST (28x28 grayscale images of handwritten digits), (ii) USPS (16x16 grayscale images of digits derived from scanned handwritten letters), and (iii) SVHN (32x32 colored images of digits with diverse backgrounds and fonts collected from Google Street View). All datasets consist of 10 classes, corresponding to the digits 0 through 9. The **Office-Home** dataset is composed of images from four distinct domains, each representing a specific visual style and content distribution. The four domains are Art (A), Clipart (C), Product (P), and Real-World (R), all with 65 classes. The **VisDA-2017** dataset comprises synthetic and real-world images across 12 diverse classes and often serves as a benchmark for domain adaptation in computer vision. In our experimental setup, we designate the training set containing synthetic images as the source domain, while the validation set, consisting of real images, serves as the target domain.

### C.3.1 Digits

For the digits experiments, we adopt the architecture proposed by Damodaran et al. (2018). The generator $g$ is a CNN model trained from scratch, with convolutional layers containing 32, 32, 64, 64, 128, and 128

filters, followed by a fully connected (FC) layer of 128 hidden units and a Sigmoid activation function. Layer normalization is applied after each convolution layer. The classifier $f$ is an FC layer with 10 units, corresponding to the number of classes. We maintain a batch size of $m = 500$ for both the source and target datasets. We construct balanced batches for the source training set by utilizing the labels, ensuring an equal number of samples for each class. The models are optimized using the Adam optimizer. Initially, the training is performed on the source data using cross-entropy loss for 10 epochs. Subsequently, the models are trained for 100 epochs using both the source and target data, with the objective as described above.

For the proposed method, we use a learning rate $lr = 0.001$. We set the hyperparameters $\eta_1 = 1$, $\eta_2 = 1$ for the objective function, $\epsilon = 1$ for all the Gaussian kernels constructing our diffusion operator, and $\lambda = 0.02$, $\tau = 1$ for the unbalanced Sinkhorn problem. For the reproduced methods we use the parameters reported in the original papers. For all reproduced methods, we use a learning rate $lr = 0.0002$. For DeepJDOT, we use $\eta_1 = 0.001$, $\eta_2 = 0.0001$. For JUMBOT, we use $\eta_1 = 0.1$, $\eta_2 = 0.1$ for the objective function, and $\lambda = 0.1$, $\tau = 1$ for the unbalanced Sinkhorn problem. We note that since we struggled to reproduce the m-POT results for this specific dataset, Table 1a presents the m-POT results reported in the original paper (Nguyen et al., 2022a). We ran each method 3 times, Table 8 presents the results for each run.

Table 8: Accuracy for Digits dataset over three seeds for (a) DeepJDOT, (b) JUMBOT, and (c) JUMBOT + GOT (ours).

|  | (a) | | | | (b) | | | | (c) | | |
|---|---|---|---|---|---|---|---|---|---|---|---|
| Run | S-M | U-M | M-U | Run | S-M | U-M | M-U | Run | S-M | U-M | M-U |
| 1 | 95.81 | 96.96 | 86.05 | 1 | 99.05 | 98.64 | 96.66 | 1 | 99.15 | 98.90 | 97.81 |
| 2 | 95.53 | 97.34 | 86.75 | 2 | 98.99 | 98.62 | 97.41 | 2 | 99.23 | 98.87 | 98.11 |
| 3 | 96.78 | 97.36 | 85.55 | 3 | 98.93 | 98.77 | 96.71 | 3 | 99.19 | 98.84 | 97.51 |

### C.3.2 Office-Home

For this dataset, the generator $g$ is a pre-trained ResNet50 with the final FC layer removed, and the classifier $f$ consists of an FC layer. We construct balanced batches for the source training set by utilizing the labels, ensuring an equal number of samples for each class. The models are optimized using the SGD optimizer with a learning rate of 0.001. Given that our cost incorporates source neighborhood information based on source labels, it is required for each mini-batch to contain at least two samples from every class. Notably, this dataset comprises 65 classes. Thus, we employ a batch size of $m = 195$, and train the model for 4000 iterations. For all reproduced methods, we follow the original papers by using a batch size of $m = 65$ and training the model for 10000 iterations.

For the proposed method, we set the hyperparameters $\eta_1 = 0.01$, $\eta_2 = 2$. for the objective function, $\epsilon = 1$ for all the Gaussian kernels constructing our diffusion operator, and $\lambda = 0.02$, $\tau = 0.5$ for the unbalanced Sinkhorn problem. For the reproduced methods, we use the parameters reported in the original papers. For DeepJDOT, we set $\eta_1 = 0.01$ and $\eta_2 = 0.05$ for the objective function. For JUMBOT and TS-POT, we use $\eta_1 = 0.01$, $\eta_2 = 0.5$ for the objective function. For JUMBOT, we use $\lambda = 0.01$, $\tau = 0.5$ for the unbalanced Sinkhorn problem. For TS-POT, we set the fraction of mass to 0.6, and the number of mini-batches $k$ to 2.

### C.3.3 VisDA

Similar to the approach taken with the Office-Home dataset, we utilize a pre-trained ResNet50 as the generator $g$, with the classifier $f$ implemented as an FC layer. For all methods, we use a batch size $m = 72$. We construct balanced batches for the source training set by utilizing the labels, ensuring an equal number of samples for each class. The models are optimized using the SGD optimizer with a learning rate of 0.0005, and trained for 10000 iterations. We employ the two-stage (TS) implementation proposed by Nguyen et al. (2022a) for our method and for TS-POT. In this implementation, the OT problem is initially solved using a batch size of $k \times m$ on the CPU, leveraging its ability to handle larger matrices. After obtaining the optimal plan, the gradient step is executed on the GPU with a batch size of $m$. To utilize the large plan in loss

Table 9: Accuracy for Office-Home dataset over three seeds for (a) DeepJDOT (b) JUMBOT (c) TS-POT and (d) JUMBOT + GOT (ours).

(a)

| Run | A-C | A-P | A-R | C-A | C-P | C-R | P-A | P-C | P-R | R-A | R-C | R-P | Avg |
|---|---|---|---|---|---|---|---|---|---|---|---|---|---|
| 1 | 51.8 | 70.7 | 76.4 | 59.7 | 66.9 | 69.1 | 57.6 | 49.1 | 75.3 | 69.4 | 54.6 | 80.1 | 65.07 |
| 2 | 52.2 | 71.1 | 76.2 | 60.8 | 67.2 | 69.5 | 58.8 | 48.4 | 75.4 | 68.7 | 54.8 | 79.8 | 65.22 |
| 3 | 51.9 | 70.9 | 75.8 | 60.9 | 65.9 | 68.9 | 58.8 | 48.7 | 75.2 | 68.6 | 55.3 | 79.8 | 65.06 |

(b)

| Run | A-C | A-P | A-R | C-A | C-P | C-R | P-A | P-C | P-R | R-A | R-C | R-P | Avg |
|---|---|---|---|---|---|---|---|---|---|---|---|---|---|
| 1 | 55.7 | 75.0 | 80.7 | 65.3 | 74.3 | 75.1 | 65.3 | 53.1 | 79.5 | 74.6 | 59.3 | 83.8 | 70.16 |
| 2 | 56.0 | 74.6 | 80.7 | 64.4 | 74.3 | 74.9 | 65.5 | 53.4 | 79.6 | 74.5 | 59.4 | 83.9 | 70.11 |
| 3 | 55.5 | 75.4 | 80.6 | 65.5 | 74.9 | 75.1 | 65.2 | 53.3 | 79.7 | 74.4 | 59.2 | 84.0 | 70.23 |

(c)

| Run | A-C | A-P | A-R | C-A | C-P | C-R | P-A | P-C | P-R | R-A | R-C | R-P | Avg |
|---|---|---|---|---|---|---|---|---|---|---|---|---|---|
| 1 | 57.7 | 76.3 | 81.7 | 68.4 | 73.7 | 76.9 | 67.3 | 55.2 | 80.6 | 75.4 | 60.2 | 83.7 | 71.43 |
| 2 | 58.4 | 77.4 | 81.4 | 67.9 | 72.5 | 76.2 | 67.4 | 54.8 | 80.7 | 75.4 | 59.1 | 84.2 | 71.29 |
| 3 | 56.3 | 77.5 | 81.7 | 68.7 | 72.2 | 76.5 | 67.7 | 55.3 | 80.4 | 75.5 | 60.4 | 84.1 | 71.36 |

(d)

| Run | A-C | A-P | A-R | C-A | C-P | C-R | P-A | P-C | P-R | R-A | R-C | R-P | Avg |
|---|---|---|---|---|---|---|---|---|---|---|---|---|---|
| 1 | 57.0 | 77.2 | 82.4 | 69.8 | 74.8 | 79.4 | 68.0 | 56.7 | 82.2 | 75.3 | 60.5 | 84.6 | 72.33 |
| 2 | 57.0 | 78.3 | 82.0 | 70.3 | 74.9 | 78.4 | 68.2 | 56.4 | 82.0 | 75.7 | 61.2 | 84.5 | 72.40 |
| 3 | 57.6 | 78.4 | 81.8 | 70.6 | 74.9 | 78.7 | 68.2 | 56.3 | 82.0 | 75.8 | 61.0 | 85.2 | 72.55 |

functions computed on smaller batch sizes, an adaptation to the original loss function is proposed. For more details, see Nguyen et al. (2022a).

For the proposed method, we set the hyperparameters $\eta_1 = 0.01$, $\eta_2 = 1$ for the objective function, $\epsilon = 1$ for all the Gaussian kernels constructing our diffusion operator. For the TS implementation, we set $s = 0.5$ as the fraction of mass for the partial OT problem, and $k = 2$ as the number of mini-batches. For the reproduced methods, we use the parameters reported in the original papers. For DeepJDOT, we use $\eta_1 = 0.005$ and $\eta_2 = 0.1$ for the objective function. For JUMBOT and TS-POT, we use $\eta_1 = 0.005$ and $\eta_2 = 1$ for the objective function. For JUMBOT, we use $\lambda = 0.01$, $\tau = 0.3$ for the unbalanced Sinkhorn problem. For TS-POT, we set the fraction of mass to 0.75, and $k = 1$ for the TS implementation. In Tables 10, we present the results per class for GOT and the competing methods.

**Features Analysis.** Additionally, we analyze the t-SNE representation of the VisDA target features obtained from the deep model trained with GOT, compared to those learned using TS-POT (Nguyen et al., 2022a), the baseline for this dataset. For better visualization, we randomly sampled 10% of the samples from each class before applying t-SNE.

While the deep model is designed to learn representations that are both domain-invariant and optimized for source classification accuracy, resulting in well-separated classes for both methods (as shown in the figure), the visualization still highlights differences between the methods. These differences help explain why GOT achieves better target accuracy compared to the standard transportation cost. For example, in Figure 11a, which shows the TS-POT features, the "knife" class (colored magenta and circled by a red dashed line) completely overlaps with the "skateboard" class (colored black). In contrast, Figure 11b, which presents features obtained using GOT, shows that these classes are well-separated. Additionally, the TS-POT model appears to have learned a representation with more than the expected 12 clusters. For instance, the cluster

Table 10: Accuracy for VisDA dataset per class over three seeds for (a) DeepJDOT, (b) JUMBOT, (c) TS-POT, and (d) TS-POT + GOT (ours).

(a)

| Run | Avg | plane | bicycle | bus | car | horse | knife | mcycle | person | plant | sktbrd | train | truck | Avg$_{class}$ |
|---|---|---|---|---|---|---|---|---|---|---|---|---|---|---|
| 1 | 69.66 | 88.6 | 59.2 | 70.1 | 67.2 | 87.6 | 4.0 | 90.4 | 68.3 | 93.5 | 49.6 | 88.4 | 31.2 | 66.5 |
| 2 | 69.88 | 91.3 | 57.8 | 69.0 | 72.5 | 84.5 | 1.9 | 89.9 | 61.1 | 91.9 | 64.2 | 84.9 | 30.5 | 66.6 |
| 3 | 69.21 | 87.8 | 57.1 | 70.9 | 69.0 | 87.2 | 3.2 | 91.7 | 70.2 | 92.3 | 46.8 | 87.5 | 25.2 | 65.7 |

(b)

| Run | Avg | plane | bicycle | bus | car | horse | knife | mcycle | person | plant | sktbrd | train | truck | Avg$_{class}$ |
|---|---|---|---|---|---|---|---|---|---|---|---|---|---|---|
| 1 | 72.84 | 93.0 | 53.4 | 76.9 | 76.6 | 89.8 | 2.1 | 94.5 | 76.2 | 94.4 | 71.2 | 89.7 | 18.4 | 69.7 |
| 2 | 72.81 | 93.7 | 56.9 | 75.5 | 71.8 | 90.5 | 3.5 | 94.2 | 77.6 | 96.1 | 60.6 | 89.0 | 27.4 | 69.7 |
| 3 | 73.27 | 89.7 | 57.0 | 74.8 | 77.3 | 90.6 | 1.5 | 93.9 | 74.6 | 94.4 | 69.9 | 90.3 | 24.6 | 69.9 |

(c)

| Run | Avg | plane | bicycle | bus | car | horse | knife | mcycle | person | plant | sktbrd | train | truck | Avg$_{class}$ |
|---|---|---|---|---|---|---|---|---|---|---|---|---|---|---|
| 1 | 75.08 | 94.4 | 66.6 | 78.8 | 73.2 | 93.4 | 1.9 | 95.2 | 73.9 | 94.8 | 79.8 | 88.4 | 31.6 | 72.7 |
| 2 | 76.54 | 95.1 | 61.4 | 82.1 | 75.2 | 91.9 | 67.1 | 94.8 | 79.2 | 96.2 | 80.0 | 89.8 | 13.7 | 77.2 |
| 3 | 75.32 | 93.1 | 68.8 | 81.6 | 74.7 | 93.8 | 2.1 | 94.4 | 75.1 | 95.9 | 76.7 | 90.8 | 26.4 | 72.8 |

(d)

| Run | Avg | plane | bicycle | bus | car | horse | knife | mcycle | person | plant | sktbrd | train | truck | Avg$_{class}$ |
|---|---|---|---|---|---|---|---|---|---|---|---|---|---|---|
| 1 | 78.48 | 95.0 | 58.6 | 82.3 | 76.9 | 95.5 | 67.6 | 95.1 | 79.2 | 95.3 | 84.2 | 89.7 | 26.9 | 78.9 |
| 2 | 78.74 | 95.0 | 64.3 | 85.2 | 77.2 | 95.5 | 68.7 | 94.0 | 78.6 | 94.7 | 82.8 | 88.9 | 25.7 | 79.2 |
| 3 | 78.47 | 94.8 | 61.7 | 85.5 | 75.9 | 95.7 | 70.8 | 94.1 | 79.3 | 95.2 | 82.8 | 87.8 | 25.9 | 79.1 |

circled by a gold dashed line in Figure 11a does not correspond to any specific class. In contrast, while GOT does not achieve perfect separation between classes (as expected, given the approximately 78.5% target accuracy), the label associated with each cluster is easily identifiable.

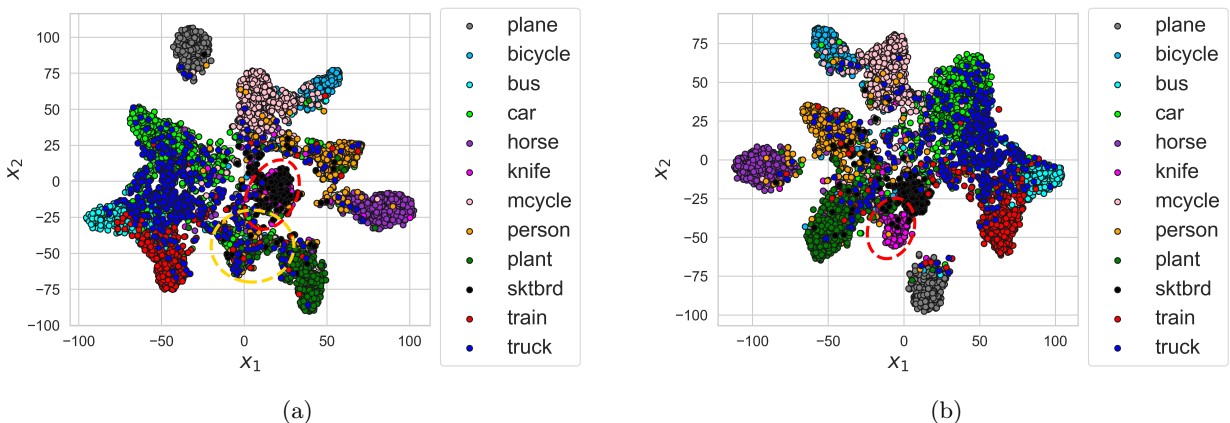

Figure 11: t-SNE visualization of VisDA features, learned from the TS-POT model using (a) the Euclidean cost and (b) GOT (ours).

**ViT-Based Domain Adaptation.** Vision Transformers (ViTs) have gained significant attention in recent years due to their ability to process visual data in a manner similar to natural language processing tasks. Unlike traditional convolutional neural networks (CNNs), which use localized convolutions to capture spatial

hierarchies, ViTs leverage self-attention mechanisms to capture long-range dependencies between pixels across the entire image. This characteristic enables ViTs to effectively model global features, making them highly suitable for tasks such as domain adaptation.

In this experiment, we integrate the GOT cost into the state-of-the-art ViT-based framework, FFTAT (Yu et al., 2024), to enhance the model's domain adaptation capabilities. The GOT cost is incorporated by adding it as an additional term to the FFTAT loss function, as described by the second term in Equation 16. The cost is applied to the global representations learned by the ViT backbone, which are then used for classification.

Following FFTAT, the backbone is pre-trained on ImageNet. For the reproduced FFTAT results, we use the parameters reported in the original paper. In our experiment (FFTAT + GOT), we use balanced source batches with a batch size of $m = 36$ and a learning rate of 0.07. The parameters for the OT-based loss are set to $\eta_1 = 0.01$ and $\eta_2 = 1$, while the FFTAT loss uses the default parameters specified in the paper. We apply the UOT formulation to derive the transportation plan, with $\lambda = 0.005$ and $\tau = 0.5$. As in the DeepJDOT framework, we set $\epsilon = 1$ for all Gaussian kernels used to construct our diffusion operator. Additional details about the FFTAT framework can be found in the original paper (Yu et al., 2024).

For comparison, we include the latest ViT-based methods: TVT (Yang et al., 2023), PMTrans (Zhu et al., 2023), CDTrans (Xu et al., 2021), and MIC (Hoyer et al., 2023). Table 11 presents the average target accuracy per class for the VisDA dataset. We observe that incorporating the GOT cost leads to a performance boost, increasing the accuracy by more than 1.% compared to FFTAT. It also achieves a 0.36% increase over the baseline method, FFTAT + UOT.

Table 11: Per-class accuracy comparison of ViT-based methods on the VisDA dataset.

| Method | plane | bicycle | bus | car | horse | knife | mcycle | person | plant | sktbrd | train | truck | Avg |
|---|---|---|---|---|---|---|---|---|---|---|---|---|---|
| TVT | 97.1 | 88.8 | 86.4 | 64.4 | 96.4 | 97.4 | 90.6 | 64.1 | 92.0 | 90.3 | 93.7 | 59.6 | 85.1 |
| PMTrans | 99.4 | 88.3 | 88.1 | 78.9 | 98.8 | 98.3 | 95.8 | 70.3 | 94.6 | 98.3 | 96.3 | 48.5 | 88.0 |
| CDTrans | 97.1 | 90.5 | 82.4 | 77.5 | 96.6 | 96.1 | 93.6 | 88.6 | 97.9 | 86.9 | 90.3 | 62.8 | 88.4 |
| MIC | 99.0 | 93.3 | 86.5 | 87.6 | 98.9 | 99.0 | 97.2 | **89.8** | **98.9** | 98.9 | 96.5 | 68.0 | 92.8 |
| FFTAT | 99.78 | 97.41 | 93.73 | 80.4 | 99.59 | 99.13 | 98.05 | 87.68 | 98.07 | 99.69 | 99.32 | **75.74** | 94.05 |
| FFTAT + UOT | 99.95 | 98.99 | 93.69 | 87.29 | **100.0** | 99.52 | **99.17** | 89.4 | 98.42 | 99.82 | 99.6 | 71.45 | 94.77 |
| FFTAT + GOT | **100.0** | **99.14** | **94.78** | **87.67** | **100.0** | **99.66** | 99.15 | 89.35 | 98.44 | **99.87** | 99.67 | 73.88 | **95.13** |

## C.4 Domain Adaptation for non-Euclidean Data

In the following section, we outline the framework utilized in the experiments presented in Section 5.3.

Previous studies (Barachant et al., 2011; 2013; Zanini et al., 2017; Rodrigues et al., 2018) have demonstrated the efficacy of using the empirical covariance matrices of the EEG recordings as an informative feature representation for this type of data. Following this established approach, we adhere to the framework introduced by Yair et al. (2019), which solves the DA problem using OT on the (non-Euclidean) Riemannian manifold of Symmetric Positive Definite (SPD) matrices.

Consider the source data samples and labels as $\{(\mathbf{X}_i^s, y_i^s)\}_{i=1}^{N_s}$ and the target data samples as $\{\mathbf{X}_j^t\}_{j=1}^{N_t}$. Each data sample $\mathbf{X}_i$ is a covariance matrix, which is an SPD matrix that lies on the Riemannian manifold of SPD matrices $\mathcal{M} \subset \mathbb{R}^{d \times d}$, where $d$ is the number of EEG channels. Applying Algorithm 1 with the Log-Euclidean metric (Arsigny et al., 2006), we derive the GOT cost. Next, we solve the OT problem using the Sinkhorn algorithm (Cuturi, 2013), as described in Section 3. Subsequently, leveraging the obtained optimal plan $\gamma$, we map the source features into the target domain. This mapping can be computed using the barycentric mapping (Courty et al., 2016; Perrot et al., 2016), as described in Section 3.

When utilizing the Log-Euclidean metric to compute the cost function, the barycenter can be represented as the weighted Riemannian mean, expressed by the following closed-form:

$$\mathbf{Z}_i^s = \exp\left(\sum_{j=1}^{N_t} \gamma_{i,j} \log\left(\mathbf{X}_j^t\right)\right), \tag{18}$$

where $\exp(\cdot)$ and $\log(\cdot)$ are the matrix exponential and logarithm. The framework is summarized in Alg. 2.

---

**Algorithm 2** GOT on the Riemannian manifold of SPD matrices

---

**Input**: $\{(\mathbf{X}_i^s, y_i^s)\}_{i=1}^{N_s}, \{\mathbf{X}_j^t\}_{j=1}^{N_t}, \epsilon_s, \epsilon_c, \epsilon_t.$
1: Compute the GOT cost $\mathbf{C}$ by applying Algorithm 1 to the source and target matrices.
2: Obtain the optimal plan $\gamma$ by applying the Sinkhorn algorithm.
3: Compute the barycentric mappings of the source samples using equation 18.
4: Return the source mappings.

---

Since the representations, $\{\mathbf{Z}_i^s\}_{i=1}^{N_s}$ and $\{\mathbf{X}_j^t\}_{j=1}^{N_t}$, are covariance matrices that lie on the Riemannian manifold of SPD matrices $\mathcal{M}$, prior to conducting any linear computation, such as training a linear classifier, we project both the source and target covariance matrices onto the tangent space $\mathcal{T}_\mathbf{M}\mathcal{M}$, using the logarithmic map:

$$\mathrm{Log}_\mathbf{M}(\mathbf{Z}_i) = \log(\mathbf{Z}_i) - \log(\mathbf{M}), \tag{19}$$

where $\mathbf{M}$ represents the Riemannian mean of all the covariance matrices.

**Datasets.** The MI1 dataset (Blankertz et al., 2007) contains EEG recordings from 7 subjects utilizing 59 electrodes. Subjects were instructed to perform two out of three MI tasks: imagining the movement of the left hand, the right hand, or the foot. Our evaluation focuses exclusively on the calibration data, comprising 100 trials for each subject. EEG signals were recorded at 1000 Hz, bandpass-filtered between 0.05 Hz and 200 Hz, and down-sampled to 100 Hz. The MI2 dataset (Tangermann et al., 2012) consists of EEG recordings from 9 subjects using 22 electrodes. The experiments include four MI tasks: the imagination of moving the left hand (class 1), right hand (class 2), both feet (class 3), and tongue (class 4). The recordings in MI2 of each subject contain two sessions, recorded on separate days, with 72 trials conducted for each MI task in each session, resulting in 288 trials per session. EEG signals were sampled at 250 Hz and bandpass-filtered between 0.5 Hz and 100 Hz. In both datasets, subjects were presented with a cue at each trial, instructing them to imagine the corresponding MI task.

**Pre-processing.** We apply the same pre-processing pipeline proposed in Zhang & Wu (2020) to both datasets. Initially, we extract a 3-second time window, spanning from $t = 0.5s$ to $t = 3.5s$ following the cue. Next, a bandpass (BP) filter ranging from 8 Hz to 30 Hz is applied to the data. Following this, we compute the empirical covariance matrix of each trial. Finally, centroid alignment (CA) (Zanini et al., 2017; Zhang & Wu, 2020) is applied to each set of covariance matrices, representing a domain, according to:

$$\mathbf{X}_i^{(k)} = \left(\mathbf{M}^{(k)}\right)^{-\frac{1}{2}} \widetilde{\mathbf{X}}_i^{(k)} \left(\mathbf{M}^{(k)}\right)^{-\frac{1}{2}}, \tag{20}$$

where $\widetilde{\mathbf{X}}_i^{(k)}$ denotes the empirical covariance of trial $i$ of the $k$-th subject, and $\mathbf{M}^{(k)}$ represents the Riemannian mean of the collection of covariance matrices of the $k$-th subject. The pre-processed set of each subject is denoted by $\{\mathbf{X}_i^{(k)}\}_{i=1}^N$, where $N$ is the total number of trials, possibly with the set of corresponding class labels $\{y_i\}_{i=1}^N$.

### C.4.1 Binary classification

**Single-source domain adaptation.** In the cross-subject experiment, we apply Algorithm 2, where each subject, in turn, serves as the target domain, with the remaining subjects alternately acting as the source domain. Accuracy is computed by initially mapping the obtained source and target representations onto a tangent space, following Equation 19, and subsequently training a linear SVM classifier on the transformed

source mappings and evaluating it on the target mappings. The presented results showcase the average target accuracy across source subjects.

For the cross-subject task, we employed the Sinkhorn algorithm with entropy regularization $\lambda = 0.02$. For the first dataset, MI1, we used the following scale parameters for the probability kernels to produce the results shown in Table 3a: $\epsilon_s = 0.4$, $\epsilon_c = 1$, and $\epsilon_t = 1$. For the second dataset, MI2, the scale parameters for the probability kernels were set as follows: $\epsilon_s = 0.9$, $\epsilon_c = 2$, and $\epsilon_t = 2$.

In Tables 12 and 13, we present the accuracy per subject on this task, including a comparison with MEKT (Zhang & Wu, 2020), utilizing their provided source code for our analysis.

Table 12: Binary classification accuracy for the cross-subject task on MI1, presented per subject.

|  | 1 | 2 | 3 | 4 | 5 | 6 | 7 | Avg | STD |
|---|---|---|---|---|---|---|---|---|---|
| MEKT-R | **71.58** | **65.58** | **64.25** | 65.58 | 83.33 | 64.67 | **81.92** | 70.99 | 8.33 |
| GOT | 70.92 | 63.5 | 63.25 | **69.75** | **86.33** | **72.67** | 79.25 | **72.24** | 8.3 |
| Difference | -0.66 | -2.08 | -1.0 | 4.17 | 3.0 | 8.0 | -2.67 | 1.25 | 3.92 |

Table 13: Binary classification accuracy for the cross-subject task on MI2, presented per subject.

|  | 1 | 2 | 3 | 4 | 5 | 6 | 7 | 8 | 9 | Avg | STD |
|---|---|---|---|---|---|---|---|---|---|---|---|
| MEKT-R | 74.57 | 50.69 | **84.55** | 64.06 | 53.39 | 64.06 | 59.38 | 86.55 | 74.31 | 67.95 | 12.84 |
| GOT | **75.52** | **53.39** | 83.94 | **67.27** | **57.55** | **65.02** | **63.8** | **91.06** | **76.13** | **70.41** | 12.28 |
| Difference | 0.95 | 2.7 | -0.61 | 3.21 | 4.16 | 0.96 | 4.42 | 4.51 | 1.82 | 2.46 | 1.8 |

**Multi-source domain adaptation.** For the leave-one-out task, we utilize the Wasserstein Barycenter Transport (WBT) framework, proposed in Montesuma & Mboula (2021). This method involves transporting all source samples to an intermediate domain, termed Wasserstein Barycenter Transport (WBT), before subsequently transferring the source samples from WBT to the target domain using standard OTDA. For the transportation of the source samples to the WBT only, we utilized the label-enhanced kernel for both the source domain and the WBT, leveraging the available labels. For simplicity, we initially map both the source and target covariance matrices onto a tangent space, using Equation 19, and then apply the WBT algorithm with the Euclidean metric. Notably, we utilize GOT instead of the standard cost for all OT applications within the WBT framework, including transportation from each source domain to WBT and from the Wasserstein barycenter of sources to the target domain. The results for these experiments appear in Table 3b.

For both datasets, we set $\lambda^b = 0.02$ for transporting all the source domains to the WBT and $\lambda = 0.1$ for transporting samples from the WBT to the target domain. In the latter case, we apply max normalization to the cost matrix before using the Sinkhorn algorithm. For the first dataset, MI1, the scale parameters for the probability kernels used to transport all source domains to the WBT are $\epsilon_s^b = 0.05$, $\epsilon_c^b = 0.3$, and $\epsilon_t^b = 0.05$. For the transportation from the WBT to the target domain, the parameters are $\epsilon_s = 0.05$, $\epsilon_c = 0.02$, and $\epsilon_t = 0.03$. For the second dataset, MI2, the scale parameters for the probability kernels used to transport all source domains to the WBT are $\epsilon_s^b = 0.01$, $\epsilon_c^b = 0.3$, and $\epsilon_t^b = 0.01$. For the transportation from the WBT to the target domain, the parameters are $\epsilon_s = 0.3$, $\epsilon_c = 0.02$, and $\epsilon_t = 0.02$.

In Tables 14 and 15, we present the accuracy per subject on this task, including a comparison with MEKT (Zhang & Wu, 2020).

### C.4.2 Multi-class classification

In this section, we assess the performance of GOT in a multi-class classification task using the MI2 dataset, taking into account all four classes available in the dataset. In addition to the baseline methods, OT and

Table 14: Binary classification accuracy for the leave-one-out task on MI1, presented per subject.

|  | 1 | 2 | 3 | 4 | 5 | 6 | 7 | Avg | STD |
|---|---|---|---|---|---|---|---|---|---|
| MEKT-R | 86.5 | 69.5 | 73.5 | 89.0 | 94.0 | **89.5** | 91.5 | 84.79 | 9.43 |
| GOT | **90.0** | **78.0** | **75.0** | **92.0** | **96.0** | 87.0 | **93.0** | **87.29** | 7.91 |
| Difference | 3.5 | 8.5 | 1.5 | 3.0 | 2.0 | -2.5 | 1.5 | 2.5 | 3.28 |

Table 15: Binary classification accuracy for the leave-one-out task on MI2, presented per subject.

|  | 1 | 2 | 3 | 4 | 5 | 6 | 7 | 8 | 9 | Avg | STD |
|---|---|---|---|---|---|---|---|---|---|---|---|
| MEKT-R | **93.06** | 49.31 | 95.83 | 73.61 | 56.25 | **70.83** | **70.14** | 94.44 | **83.33** | 76.31 | 16.76 |
| GOT | 91.67 | **51.39** | **97.92** | **75.0** | **61.81** | 70.14 | 68.06 | **96.53** | 82.64 | **77.24** | 16.14 |
| Difference | -1.39 | 2.08 | 2.09 | 1.39 | 5.56 | -0.69 | -2.08 | 2.09 | -0.69 | 0.93 | 2.38 |

OT-reg (Courty et al., 2016; 2014; Yair et al., 2019), we compare GOT to standard DA methods, including CORAL (Sun et al., 2017), SA (Fernando et al., 2013), and TCA (Pan et al., 2010).

For all the multi-class classification experiments we employ the Sinkhorn algorithm with entropy regularization parameter $\lambda = 0.1$ and apply Algorithm 1 with $\epsilon_s = 3, \epsilon_c = 0.1, \epsilon_t = 0.1$. For the reproduced methods, OT and OT-reg, we apply median normalization to the cost matrix before applying the Sinkhorn algorithm. We set $\lambda = 0.005$ for the entropy parameter. For OT-reg, we use $\eta = 10$ for the label regularization, as suggested in the original paper (Yair et al., 2019). For CORAL, SA, and TCA, we utilize the ADAPT package (de Mathelin et al., 2021) with the default parameters.

Table 16a shows the results for the cross-session task. In this experiment, for each subject, the first session (day) serves as the source domain, while the second session serves as the target domain, and vice versa. The reported target accuracy represents the average accuracy across both sessions, obtained as follows: First, we apply Algorithm 2 to obtain the transported source samples. Next, we map both the target and the transported source covariance matrices to a tangent plane, as described in Equation 19. We then train a linear SVM on the mapped source samples and test it on the target samples. Following Zanini et al. (2017); Yair et al. (2019), we present results for five subjects out of the available nine, as the remaining subjects exhibited poor results and were considered invalid in those works. We observe that the proposed method achieves the highest accuracy for three out of the five subjects. For subject 1, however, it shows lower accuracy compared to the baseline OT-reg. This is the only instance across all experiments where we do not achieve an improvement over the baseline. Notably, GOT yields the highest average accuracy, surpassing OT and OT-reg by over 6% and 2%, respectively. Detailed results for all subjects appear in Table 17a.

Table 16: Multi-class classification accuracy on dataset MI2 (five valid subjects). (a) Results for the cross-session task. (b) Results for the cross-subject task.

| (a) | | | | | | | (b) | | | | | |
|---|---|---|---|---|---|---|---|---|---|---|---|---|
| Method | 1 | 3 | 7 | 8 | 9 | Avg | Method | 1 | 3 | 7 | 8 | 9 | Avg |
| CORAL | 82.3 | 87.7 | 80.9 | 83.0 | 72.9 | 81.35 | CORAL | 53.1 | 58.2 | 52.0 | 52.5 | 53.3 | 53.82 |
| SA | 83.7 | **87.8** | 81.4 | 83.9 | 74.8 | 82.33 | SA | 58.6 | 63.2 | 53.8 | 59.3 | 58.2 | 58.63 |
| TCA | 71.0 | 78.1 | 64.6 | 77.8 | 70.1 | 72.33 | TCA | 59.1 | 61.2 | 49.4 | 60.7 | 52.7 | 56.64 |
| OT | 84.0 | 85.9 | 81.1 | 76.4 | 68.2 | 79.13 | OT | 57.2 | 65.4 | 59.7 | 57.5 | 52.7 | 58.51 |
| OT-reg | **85.2** | 84.5 | 81.1 | 85.4 | 78.1 | 82.88 | OT-reg | 63.2 | 69.7 | 63.8 | 66.1 | 62.1 | 64.97 |
| Ours | 84.7 | 86.5 | **84.5** | **88.9** | **81.4** | **85.21** | Ours | **66.7** | **75.1** | **69.4** | **73.4** | **65.5** | **70.03** |

In Table 16b, we present the results for the cross-subject task, analogous to the results shown in Table 3a for the binary classification case. Same as in the cross-session task, only the five valid subjects are considered.

Our method achieves the highest accuracy across all subjects and demonstrates an average performance that significantly exceeds both baseline methods. Detailed results for all nine subjects can be found in Table 17b.

Table 17: Comparison of (a) cross-session and (b) cross-subject multi-class classification on the MI2 dataset.

| | (a) | | | | | | | (b) | | | | | |
|---|---|---|---|---|---|---|---|---|---|---|---|---|---|
| Subject | CORAL | SA | TCA | OT | OT-reg | GOT | Subject | CORAL | SA | TCA | OT | OT-reg | GOT |
| 1 | 82.29 | 83.68 | 71.01 | 84.03 | **85.24** | 84.72 | 1 | 43.66 | 49.57 | 51.91 | 49.1 | 54.24 | **57.99** |
| 2 | 50.35 | 51.91 | 44.27 | 56.08 | 57.99 | **58.85** | 2 | 27.19 | 27.82 | 27.58 | 27.79 | **28.23** | 27.44 |
| 3 | 87.67 | **87.85** | 78.13 | 85.94 | 84.55 | 86.46 | 3 | 47.34 | 53.34 | 50.59 | 52.89 | 57.7 | **62.08** |
| 4 | 57.47 | 59.9 | 37.5 | 58.51 | 57.81 | **60.24** | 4 | 35.31 | 34.71 | 32.82 | 37.7 | 38.85 | **39.29** |
| 5 | 41.32 | 41.67 | 28.47 | **46.7** | 45.14 | 44.27 | 5 | 30.96 | 30.57 | 29.42 | 32.79 | 33.26 | **33.51** |
| 6 | 47.4 | 50.87 | 36.81 | **52.95** | 50.69 | 48.61 | 6 | 33.18 | 32.54 | 32.08 | 34.13 | 34.09 | **34.42** |
| 7 | 80.9 | 81.42 | 64.58 | 81.08 | 81.08 | **84.55** | 7 | 39.69 | 41.41 | 40.82 | 44.95 | 48.11 | **52.21** |
| 8 | 82.99 | 83.85 | 77.78 | 76.39 | 85.42 | **88.89** | 8 | 44.67 | 52.07 | 51.81 | 47.73 | 55.53 | **62.04** |
| 9 | 72.92 | 74.83 | 70.14 | 68.23 | 78.13 | **81.42** | 9 | 42.76 | 48.67 | 45.5 | 42.45 | 51.68 | **54.94** |
| Avg | 67.03 | 68.44 | 56.52 | 67.77 | 69.56 | **70.89** | Avg | 38.31 | 41.19 | 40.28 | 41.06 | 44.63 | **47.1** |

**Parameter tuning strategies.** For the multi-class classification task on the MI2 dataset, we additionally explored different hyperparameter tuning strategies. Following Kerdoncuff et al. (2020), we applied an unsupervised tuning approach based solely on source labels. To make this procedure more robust, we extend the selection rule originally used in MLOT, which simply chooses the parameter set yielding the highest source accuracy. Specifically, we first collect the top 5% of parameter configurations with the highest source accuracy, or alternatively the configurations within 5 points of the maximum source accuracy (whichever results in a larger set). For each parameter, we then select the most frequent value in this subset. In case of ties, we resolve by choosing the value with the highest source accuracy. For the parameters $\epsilon_s$, $\epsilon_c$, and $\epsilon_t$, which are naturally more sensitive to the data, we instead take the average value across the subset. This procedure is applied consistently across all competing methods.

We compare GOT to the baselines OT and OT-reg in the Riemannian framework, where the cost function is defined as the Log-Euclidean distance between covariance matrices rather than the standard Euclidean cost. In addition, we evaluate against three competing methods: MLOT (Kerdoncuff et al., 2020) and RM-LOT (Jawanpuria et al., 2025), which incorporate Mahalanobis metric learning into the cost function, and MADAOT (Dhouib et al., 2020), which augments the cost with a weight vector and solves a minmax problem to jointly obtain the transport plan and the classifier. For MLOT and MADAOT we used the authors' publicly available code, while RMLOT had no released implementation and was therefore re-implemented by us. Since MADAOT assumes Euclidean geometry and both MLOT and RMLOT are limited to Mahalanobis distances, we projected the covariance matrices from the SPD manifold to a tangent plane (Eq. 19) before applying these methods. To ensure fairness, we trained a linear SVM on top of the representations obtained by MADAOT, instead of using its built-in predictions, after verifying that this yields higher accuracy. Furthermore, as suggested in MLOT, we applied PCA preprocessing to the data and treated the number of components as a tunable parameter (with the option of disabling it). This preprocessing step was also included for RMLOT and MADAOT, so that the parameter search was performed consistently across methods. We did not apply PCA preprocessing for OT, OT-reg, or GOT, as these methods directly operate on covariance matrices within the SPD manifold.

**Results.** We report results for both cross-session and cross-subject adaptation. Tables 18a and 18b present the cross-session results, obtained using parameters tuned by source labels (with the procedure described above) and by target labels, per source–target pair. Tables 19a and 19b report the cross-session results restricted to the five valid subjects. In this setting, GOT consistently improves upon the baseline OT with Log-Euclidean cost under both tuning strategies, for both the valid five subjects and the entire dataset. When compared to MLOT, GOT achieves similar performance with optimally tuned parameters (Table 18b), but lower accuracy under source-tuned parameters (Table 18a). Interestingly, for the five valid subjects,

GOT is comparable to MLOT in both tuning strategies. A key advantage of GOT is its flexibility: it can be seamlessly integrated into other OT-based methods. We incorporated the GOT cost into the MLOT framework, reported under the "MLOT+GOT" column. Unlike GOT, which requires tuning only four parameters $(\lambda, \epsilon_s, \epsilon_c, \epsilon_t)$, MLOT+GOT involves seven parameters in total. This allowed us to adopt a more robust selection strategy for the three GOT-specific parameters, which are interdependent and directly affect the cost: instead of averaging (as in GOT), we selected the most frequent combination of $(\epsilon_s, \epsilon_c, \epsilon_t)$ values within the top-performing subset. We observe that MLOT+GOT improves the cross-session performance of MLOT by more than 1% with optimally tuned parameters. Under unsupervised tuning, the results are comparable when considering all subjects, and improved when focusing on the five valid subjects.

It is worth noting that the cross-session task includes only 18 experiments out of the total 306 source–target pairs in the dataset. To provide a more comprehensive comparison, we now turn to the cross-subject task, presented in Tables 20a and 20b for source- and target-tuned results. In this setting, GOT alone outperforms all competing methods—surpassing the baseline OT by more than 10% with optimally tuned parameters and by more than 5% under unsupervised tuning. The advantage is even more evident for the five valid subjects (Tables 21a and 21b). In both tuning strategies, and for both the valid five and all subjects, GOT consistently outperforms all competing methods, with MLOT–the second-best–lower by at least 2%.

Table 18: Cross-session Multi-class classification accuracy on dataset MI2. (a) Tuning by source. (b) Tuning by target.

(a)

| Subject | OT | OT-reg | MADAOT | MLOT | RMLOT | GOT | MLOT+GOT |
|---|---|---|---|---|---|---|---|
| 1 | 81.08 | 80.9 | 68.75 | 84.9 | 83.33 | 83.51 | **86.63** |
| 2 | 55.38 | 52.26 | 43.4 | 54.17 | 48.09 | 55.21 | **55.56** |
| 3 | 84.03 | 85.59 | 71.53 | 84.72 | 78.3 | **86.98** | 86.28 |
| 4 | 59.2 | 53.12 | 39.58 | **61.11** | 46.35 | 57.29 | 57.81 |
| 5 | 47.22 | 45.83 | 36.46 | **50.69** | 35.24 | 42.88 | 45.83 |
| 6 | 40.1 | 48.09 | 38.89 | **53.12** | 31.6 | 42.36 | 51.74 |
| 7 | 78.12 | 78.99 | 70.14 | 83.33 | 67.36 | 74.31 | **84.72** |
| 8 | 77.26 | 83.33 | 72.22 | 83.85 | 81.08 | **88.19** | 83.85 |
| 9 | 68.06 | 75.17 | 65.8 | 77.26 | 68.23 | **80.73** | 75.52 |
| Avg | 65.61 | 67.03 | 56.31 | **70.35** | 59.95 | 67.94 | 69.77 |

(b)

| Subject | OT | OT-reg | MADAOT | MLOT | RMLOT | GOT | MLOT+GOT |
|---|---|---|---|---|---|---|---|
| 1 | 83.16 | 86.46 | 74.48 | 88.19 | 84.9 | 86.98 | **89.58** |
| 2 | 57.29 | 59.2 | 51.56 | 61.46 | 61.11 | 62.67 | **63.54** |
| 3 | 86.63 | 86.98 | 72.05 | 89.06 | 88.02 | 88.72 | **89.76** |
| 4 | 60.76 | 65.1 | 43.23 | 64.24 | 64.93 | **66.67** | 65.45 |
| 5 | 48.09 | 49.48 | 37.15 | 51.22 | 44.44 | 50.35 | **51.74** |
| 6 | 55.21 | 55.03 | 39.24 | 56.94 | 52.6 | 56.25 | **57.29** |
| 7 | 81.77 | 83.51 | 76.56 | 87.67 | 86.46 | 87.15 | **89.06** |
| 8 | 78.3 | 85.94 | 74.48 | 89.41 | 86.11 | **90.45** | 90.28 |
| 9 | 68.06 | 81.25 | 69.97 | 81.08 | 79.51 | 83.16 | **83.51** |
| Avg | 68.81 | 72.55 | 59.86 | 74.36 | 72.01 | 74.71 | **75.58** |

Table 19: Cross-session Multi-class classification accuracy on dataset MI2 (five valid subjects). (a) Tuning by source. (b) Tuning by target.

(a)

| Method | 1 | 3 | 7 | 8 | 9 | Avg |
|---|---|---|---|---|---|---|
| OT | 81.08 | 84.03 | 78.12 | 77.26 | 68.06 | 77.71 |
| OT-reg | 80.9 | 85.59 | 78.99 | 83.33 | 75.17 | 80.8 |
| MADAOT | 68.75 | 71.53 | 70.14 | 72.22 | 65.8 | 69.69 |
| MLOT | 84.9 | 84.72 | 83.33 | 83.85 | 77.26 | 82.81 |
| RMLOT | 83.33 | 78.3 | 67.36 | 81.08 | 68.23 | 75.66 |
| GOT | 83.51 | **86.98** | 74.31 | **88.19** | **80.73** | 82.74 |
| MLOT+GOT | **86.63** | 86.28 | **84.72** | 83.85 | 75.52 | **83.4** |

(b)

| Method | 1 | 3 | 7 | 8 | 9 | Avg |
|---|---|---|---|---|---|---|
| OT | 83.16 | 86.63 | 81.77 | 78.3 | 68.06 | 79.58 |
| OT-reg | 86.46 | 86.98 | 83.51 | 85.94 | 81.25 | 84.83 |
| MADAOT | 74.48 | 72.05 | 76.56 | 74.48 | 69.97 | 73.51 |
| MLOT | 88.19 | 89.06 | 87.67 | 89.41 | 81.08 | 87.08 |
| RMLOT | 84.9 | 88.02 | 86.46 | 86.11 | 79.51 | 85.0 |
| GOT | 86.98 | 88.72 | 87.15 | **90.45** | 83.16 | 87.29 |
| MLOT+GOT | **89.58** | **89.76** | **89.06** | 90.28 | **83.51** | **88.44** |

Table 20: Cross-subject Multi-class classification accuracy on dataset MI2. (a) Tuning by source. (b) Tuning by target.

(a)

| Subject | OT | OT-reg | MADAOT | MLOT | RMLOT | GOT |
|---|---|---|---|---|---|---|
| 1 | 46.56 | 46.07 | 43.84 | $\underline{50.75}$ | 37.39 | **55.53** |
| 2 | 26.18 | $\underline{26.92}$ | 24.78 | 26.54 | 25.82 | **28.55** |
| 3 | 49.98 | 51.35 | 46.27 | $\underline{55.99}$ | 43.53 | **61.65** |
| 4 | 36.06 | 36.78 | 35.35 | $\underline{38.35}$ | 31.1 | **38.99** |
| 5 | 32.18 | 31.85 | 31.76 | $\underline{32.47}$ | 29.21 | **32.62** |
| 6 | 32.88 | $\underline{33.33}$ | 30.97 | **34.22** | 29.86 | 32.71 |
| 7 | 42.84 | 43.87 | 41.1 | $\underline{47.17}$ | 34.56 | **49.89** |
| 8 | 45.89 | 48.99 | 45.64 | $\underline{54.5}$ | 39.12 | **60.36** |
| 9 | 42.55 | 44.43 | 42.2 | $\underline{48.17}$ | 39.02 | **52.65** |
| Avg | 39.46 | 40.4 | 37.99 | $\underline{43.13}$ | 34.4 | **45.88** |

(b)

| Subject | OT | OT-reg | MADAOT | MLOT | RMLOT | GOT |
|---|---|---|---|---|---|---|
| 1 | 50.22 | 57.86 | 48.73 | $\underline{62.67}$ | 60.77 | **64.32** |
| 2 | 29.63 | 30.96 | 29.67 | $\underline{32.14}$ | 31.24 | **34.44** |
| 3 | 54.47 | 61.79 | 50.95 | $\underline{65.61}$ | 63.28 | **69.37** |
| 4 | 38.66 | 41.79 | 38.39 | $\underline{44.22}$ | 43.13 | **45.94** |
| 5 | 34.35 | 35.56 | 33.76 | $\underline{36.02}$ | 34.51 | **37.87** |
| 6 | 35.33 | 36.7 | 33.47 | $\underline{37.86}$ | 37.14 | **39.72** |
| 7 | 46.47 | 53.48 | 45.71 | $\underline{59.6}$ | 54.46 | **60.25** |
| 8 | 49.88 | 59.19 | 50.18 | $\underline{65.99}$ | 62.84 | **69.52** |
| 9 | 44.9 | 54.96 | 49.76 | $\underline{59.44}$ | 56.27 | **62.47** |
| Avg | 42.66 | 48.03 | 42.29 | $\underline{51.51}$ | 49.29 | **53.77** |

Table 21: Cross-subject Multi-class classification accuracy on dataset MI2 (five valid subjects). (a) Tuning by source. (b) Tuning by target.

(a)

| Method | 1 | 3 | 7 | 8 | 9 | Avg |
|---|---|---|---|---|---|---|
| OT | 55.38 | 62.74 | 57.53 | 56.25 | 52.28 | 56.84 |
| OT-reg | 54.36 | 62.48 | 57.25 | 59.09 | 54.62 | 57.56 |
| MADAOT | 54.41 | 59.79 | 54.97 | 57.29 | 53.49 | 55.99 |
| MLOT | $\underline{61.48}$ | $\underline{70.01}$ | $\underline{64.04}$ | $\underline{66.43}$ | $\underline{59.7}$ | $\underline{64.33}$ |
| RMLOT | 43.4 | 54.51 | 43.12 | 46.2 | 47.35 | 46.92 |
| GOT | **63.24** | **75.13** | **67.3** | **71.53** | **64.45** | **68.33** |

(b)

| Method | 1 | 3 | 7 | 8 | 9 | avg |
|---|---|---|---|---|---|---|
| OT | 58.53 | 66.67 | 60.61 | 58.59 | 54.73 | 59.83 |
| OT-reg | 65.6 | 72.16 | 66.08 | 68.75 | 63.91 | 67.3 |
| MADAOT | 57.86 | 63.24 | 57.2 | 61.02 | 58.25 | 59.51 |
| MLOT | $\underline{69.16}$ | $\underline{76.39}$ | $\underline{72.98}$ | $\underline{74.98}$ | $\underline{68.36}$ | $\underline{72.37}$ |
| RMLOT | 68.47 | 74.37 | 69.25 | 72.01 | 65.23 | 69.87 |
| GOT | **71.18** | **79.82** | **73.42** | **78.17** | **72.18** | **74.95** |

# D   Proofs

**Lemma D.1.** *The asymptotic expansion of a single-domain diffusion operator, defined in Equation 7, is expressed as follows:*

$$P_\epsilon f(x) = f(x) - \frac{m_2}{m_0}\epsilon \left( \Delta f - f\frac{\Delta\mu}{\mu} \right)(x) + O(\epsilon^2). \tag{21}$$

*Proof of Lemma D.1.* According to Lemma 8 from Coifman & Lafon (2006), given an isotropic kernel $k_\epsilon(x,y) = h\left(\frac{\|x-y\|^2}{\epsilon}\right)$ with an exponential decay, and an operator $T_\epsilon$ defined by $T_\epsilon g(x) = \int_{\mathcal{M}} k_\epsilon(x,y)g(y)dy$, the asymptotic expansion of $T_\epsilon$ for all $g \in \mathcal{C}^3(\mathcal{M})$ and for all $x \in \mathcal{M}$ is given by:

$$T_\epsilon g(x) = m_0 g(x) - m_2\epsilon \left( \Delta g(x) - w(x)g(x) \right) + O(\epsilon^2), \tag{22}$$

where $\Delta$ is the Laplace–Beltrami operator on $\mathcal{M}$, $m_0, m_2$ are two constants depending on the kernel $h$, and $w(x)$ is a potential function.

For the diffusion operator $P_\epsilon$ defined in Equation 7, we use $g(x) = \frac{f(x)\mu(x)}{d_\epsilon(x)}$, and from Equation 22 we get:

$$P_\epsilon f(x) = m_0 \frac{f(x)\mu(x)}{d_\epsilon(x)} - m_2\epsilon \left( \Delta \left( \frac{f(x)\mu(x)}{d_\epsilon(x)} \right) - w(x)\frac{f(x)\mu(x)}{d_\epsilon(x)} \right) + O(\epsilon^2), \tag{23}$$

where $d_\epsilon(x) = \int K_\epsilon(x,y)\mu(y)dy$.

Similarly, we use $g(x) = \mu(x)$ in Equation 22, and obtain that the asymptotic expansion of $d_\epsilon$ is given by:

$$d_\epsilon(x) = m_0\mu(x) - m_2\epsilon\left(\Delta\mu(x) - w(x)\mu(x)\right) + O(\epsilon^2)$$
$$= m_0\mu(x)\left(1 - \frac{m_2}{m_0}\epsilon\left(\frac{\Delta\mu}{\mu} - w\right)(x)\right) + O(\epsilon^2) \tag{24}$$

By applying the Taylor expansion of $\frac{1}{1-x}$ and neglecting terms of order $O(\epsilon^2)$ and higher, we obtain:

$$(d_\epsilon)^{-1}(x) = (m_0\mu(x))^{-1}\left(1 + \frac{m_2}{m_0}\epsilon\left(\frac{\Delta\mu}{\mu} - w\right)(x)\right) + O(\epsilon^2) \tag{25}$$

Multiplying Equation 25 by $f\mu$ and subsequently applying the Laplace–Beltrami operator yields:

$$\frac{f\mu}{d_\epsilon}(x) = \frac{f}{m_0}\left(1 + \frac{m_2}{m_0}\epsilon\left(\frac{\Delta\mu}{\mu} - w\right)(x)\right) + O(\epsilon^2) \tag{26}$$

$$\Delta\left(\frac{f\mu}{d_\epsilon}\right)(x) = \frac{\Delta f}{m_0} + \frac{m_2}{m_0^2}\epsilon\Delta\left(f\left(\frac{\Delta\mu}{\mu} - w\right)\right)(x) + O(\epsilon^2) \tag{27}$$

Finally, by substituting Equations 26 and 27 into Equation 23, and neglecting terms of order $O(\epsilon^2)$, we derive the asymptotic expansion of the operator $P_\epsilon$:

$$P_\epsilon f(x) = f(x) - \frac{m_2}{m_0}\epsilon\left(-f\frac{\Delta\mu}{\mu} + wf + \Delta f - wf\right)(x) + O(\epsilon^2)$$
$$= f(x) - \frac{m_2}{m_0}\epsilon\left(\Delta f - f\frac{\Delta\mu}{\mu}\right)(x) + O(\epsilon^2)$$

$\square$

**Lemma D.2.** *The asymptotic expansion of the cross-domain diffusion operator, defined by:*

$$Q_\epsilon f(y) = \int \frac{k_\epsilon(x,y)}{d_{t,\epsilon}(x)}f(x)\mu(x)dx, \tag{28}$$

*is expressed as follows:*

$$Q_\epsilon f(x) = f(x)\frac{\mu}{\nu}(x) - \frac{m_2}{m_0}\epsilon\left(\Delta\left(f\frac{\mu}{\nu}\right) - f\frac{\mu}{\nu}\frac{\Delta\nu}{\nu}\right)(x) + O(\epsilon^2). \tag{29}$$

*Proof of Lemma D.2.* For the cross-domain diffusion operator defined in Equation 28, with $\epsilon_c$ as the scale hyperparameter of the cross-domain Gaussian kernel, we substitute $g(x) = \frac{f\mu}{d_{t,\epsilon_c}}(x)$ into Equation 22, and obtain:

$$Q_{\epsilon_c}f(x) = m_0\frac{f\mu}{d_{t,\epsilon_c}}(x) - m_2\epsilon_c\left(\Delta\left(\frac{f\mu}{d_{t,\epsilon_c}}\right)(x) - w(x)\frac{f\mu}{d_{t,\epsilon_c}}(x)\right) + O(\epsilon_c^2), \tag{30}$$

where $d_{t,\epsilon_c}(x) = \int k_{\epsilon_c}(x,y)\nu(y)dy$.

Similarly, we substitute $g(x) = \nu(x)$ to Equation 22, and obtain the asymptotic expansion of $d_{t,\epsilon_c}$:

$$d_{t,\epsilon_c}(x) = m_0\nu(x) - m_2\epsilon_c\left(\Delta\nu(x) - w(x)\nu(x)\right) + O(\epsilon_c^2). \tag{31}$$

We utilize the Taylor expansion as in Equation 25, leading to the derivation:

$$(d_{\epsilon_c})^{-1}(x) = (m_0\nu(x))^{-1}\left(1 + \frac{m_2}{m_0}\epsilon_c\left(\frac{\Delta\nu}{\nu} - w\right)(x)\right) + O(\epsilon_c^2). \tag{32}$$

Multiplying Equation 32 by $f$ and subsequently applying the Laplace–Beltrami operator yields:

$$\frac{f\mu}{d_{\epsilon_c}}(x) = \frac{f\mu}{m_0\nu}\left(1 + \frac{m_2}{m_0}\epsilon_c\left(\frac{\Delta\nu}{\nu} - w\right)(x)\right) + O(\epsilon_c^2). \tag{33}$$

$$\Delta\left(\frac{f\mu}{d_{\epsilon_c}}\right)(x) = \frac{1}{m_0}\Delta\left(f\frac{\mu}{\nu}\right) + \frac{m_2}{m_0^2}\epsilon_c\Delta\left(f\frac{\mu}{\nu}\left(\frac{\Delta\nu}{\nu} - w\right)\right)(x). \tag{34}$$

Finally, by substituting Equations 33 and 34 into Equation 30 and neglecting terms of order $O(\epsilon^2)$, we derive the asymptotic expansion of the operator $Q_{\epsilon_c}$:

$$Q_{\epsilon_c}f(x) = f\frac{\mu}{\nu}(x) - \frac{m_2}{m_0}\epsilon_c\left(-f\frac{\mu}{\nu}\frac{\Delta\nu}{\nu} + f\frac{\mu}{\nu}w + \Delta\left(f\frac{\mu}{\nu}\right) - f\frac{\mu}{\nu}w\right)(x) + O(\epsilon_c^2)$$

$$= f\frac{\mu}{\nu}(x) - \frac{m_2}{m_0}\epsilon_c\left(\Delta\left(f\frac{\mu}{\nu}\right) - f\frac{\mu}{\nu}\frac{\Delta\nu}{\nu}\right)(x) + O(\epsilon_c^2).$$

$\square$

**Proposition 4.1.** *Suppose $f \in \mathcal{C}^4(\mathcal{M})$, and suppose $\mu, \nu \in \mathcal{C}^4(\mathcal{M})$ denote the probability measures of the source and target domains, respectively, where $\mu$ is dominated by $\nu$. For sufficiently small $\epsilon$, the asymptotic expansion of operator $S_\epsilon$ is given by:*

$$S_\epsilon f(x) = \frac{\mu}{\nu}(x)\left[f - \frac{m_2}{m_0}\epsilon\left[3\Delta f + 2\left(f\frac{\Delta\left(\frac{\mu}{\nu}\right)}{\frac{\mu}{\nu}} + 2\nabla f\nabla\log\left(\frac{\mu}{\nu}\right)\right)\right.\right. \tag{9}$$

$$\left.\left. - f\left(\frac{\Delta\mu}{\mu} + 2\frac{\Delta\nu}{\nu}\right)\right]\right](x) + O(\epsilon^2),$$

*where $\Delta, \nabla$ are the Laplace–Beltrami operator and the covariant derivative on $\mathcal{M}$, respectively, and $m_0, m_2$ are two constants defined by the Gaussian kernel and by the manifold $\mathcal{M}$.*

*Proof of Proposition 4.1.* The proposed diffusion operator is defined by the composition $S_{\underline{\epsilon}}f(x) = P_{t,\epsilon_t}Q_{\epsilon_c}P_{s,\epsilon_s}f(x)$, where $\underline{\epsilon} = (\epsilon_s, \epsilon_c, \epsilon_t)$.

We start by defining the function $g(x) = P_{s,\epsilon_s}f(x)$. Substituting the asymptotic expansion of the source diffusion operator, as derived in Equation 21, into Equation 29, we obtain:

$$Q_{\epsilon_c}P_{s,\epsilon_s}f(x) = \frac{\mu}{\nu}\left(f - \frac{m_2}{m_0}\epsilon_s\left(\Delta f - f\frac{\Delta\mu}{\mu}\right)\right)(x) - \frac{m_2}{m_0}\epsilon_c\left(\Delta\left(f\frac{\mu}{\nu}\right) - f\frac{\mu}{\nu}\frac{\Delta\nu}{\nu}\right)(x) + O(\epsilon^2)$$

$$= f\frac{\mu}{\nu}(x) - \frac{m_2}{m_0}\left(\epsilon_s\frac{\mu}{\nu}\Delta f + \epsilon_c\Delta\left(f\frac{\mu}{\nu}\right) - f\frac{\mu}{\nu}\left(\epsilon_s\frac{\Delta\mu}{\mu} + \epsilon_c\frac{\Delta\nu}{\nu}\right)\right) + O(\epsilon^2), \tag{35}$$

where all terms with order $O(\epsilon^2)$ were neglected.

Remark that assuming $\epsilon_s = \epsilon_c = \epsilon$ yields:

$$Q_\epsilon P_{s,\epsilon}f(x) = f\frac{\mu}{\nu}(x) - \frac{m_2}{m_0}\epsilon\left(\frac{\mu}{\nu}\Delta f + \Delta\left(f\frac{\mu}{\nu}\right) - f\frac{\mu}{\nu}\left(\frac{\Delta\mu}{\mu} + \frac{\Delta\nu}{\nu}\right)\right)(x) + O(\epsilon^2). \tag{36}$$

Next, we define the function $g(x) = Q_{\epsilon_c} P_{s,\epsilon_s} f(x)$, and substitute Equation 36 into the asymptotic expansion of the target diffusion operator, as defined in Equation 21:

$$P_{t,\epsilon_t} Q_{\epsilon_c} P_{s,\epsilon_s} f(x) = f\frac{\mu}{\nu}(x) - \frac{m_2}{m_0}\left(\epsilon_s \frac{\mu}{\nu}\Delta f + \epsilon_c \Delta\left(f\frac{\mu}{\nu}\right) - f\frac{\mu}{\nu}\left(\epsilon_s\frac{\Delta\mu}{\mu} + \epsilon_c\frac{\Delta\nu}{\nu}\right)\right)$$

$$- \frac{m_2}{m_0}\epsilon_t\left(\Delta\left(f\frac{\mu}{\nu}\right) - f\frac{\mu}{\nu}\frac{\Delta\nu}{\nu}\right)(x) + O(\epsilon^2)$$

$$= f\frac{\mu}{\nu}(x) - \frac{m_2}{m_0}\left(\epsilon_s\frac{\mu}{\nu}\Delta f + (\epsilon_c + \epsilon_t)\Delta\left(f\frac{\mu}{\nu}\right)\right.$$

$$\left. - f\frac{\mu}{\nu}\left(\epsilon_s\frac{\Delta\mu}{\mu} + (\epsilon_c + \epsilon_t)\frac{\Delta\nu}{\nu}\right)\right)(x) + O(\epsilon^2), \tag{37}$$

where all terms with order $O(\epsilon^2)$ were neglected.

Assuming $\epsilon_s = \epsilon_c = \epsilon$ we get:

$$P_{t,\epsilon_t} Q_{\epsilon_c} P_{s,\epsilon_s} f(x) = f\frac{\mu}{\nu}(x) - \frac{m_2}{m_0}\epsilon\left(\frac{\mu}{\nu}\Delta f + 2\Delta\left(f\frac{\mu}{\nu}\right) - f\frac{\mu}{\nu}\left(\frac{\Delta\mu}{\mu} + 2\frac{\Delta\nu}{\nu}\right)\right)(x) + O(\epsilon^2). \tag{38}$$

Finally, by utilizing:

$$\Delta\left(f\frac{\mu}{\nu}\right) = \frac{\mu}{\nu}\Delta f + f\Delta\left(\frac{\mu}{\nu}\right) + 2\nabla f\nabla\left(\frac{\mu}{\nu}\right), \tag{39}$$

we yield the expression:

$$S_\epsilon f(x) = f\frac{\mu}{\nu}(x) - \frac{m_2}{m_0}\epsilon\left(3\frac{\mu}{\nu}\Delta f + 2f\Delta\left(\frac{\mu}{\nu}\right) - f\frac{\mu}{\nu}\left(\frac{\Delta\mu}{\mu} + 2\frac{\Delta\nu}{\nu}\right) + 4\nabla f\nabla\left(\frac{\mu}{\nu}\right)\right)(x) + O(\epsilon^2). \tag{40}$$

Lastly, for a more comprehensive analysis, we utilize the property $\frac{\nabla f}{f} = \nabla\log(f)$ to derive the expression presented in Proposition 4.1. $\square$

