# OpenReview forum: "Geometric Optimal Transport for Unsupervised Domain Adaptation"
_TMLR — Accepted by TMLR_

### Review · Reviewer_eA7v · 2025-07-20

**Summary Of Contributions:**

This paper explores geometric optimal transport (OT) for unsupervised domain adaptation (UDA).

The core idea is as follows:

- Traditional OT in UDA has focused on Euclidean space, which presents potential limitations.
- This paper,focuses on UDA under the manifold assumption.

The proposed method involves:
- Introducing source, target, and cross-domain diffusion operators.
- Defining the final operator as the product of these three diffusion operators.
- Defining the transport cost as the negative log-likelihood of this operator.

Theoretical and practical support includes:
- Theoretical analysis presented in Section 4.1.
- Experiments conducted on typical UDA benchmarks.

**Audience:**

Yes

**Claims And Evidence:**

No

**Requested Changes:**

I recommend that the authors make major revisions to the current version, particularly concerning points (2) and (3). The paper could be improved by:

- Adding clear theoretical or mathematical analysis to illustrate why and when manifold assumptions can help UDA (within the OT context).

- Differentiating experiments, algorithms, and methods in deep and non-deep OT UDA, as both require quite different assumptions and scenarios.

**Strengths And Weaknesses:**

**Audience.**
Based on these key facts, this paper addresses a long-term, meaningful problem. Its objective is to propose a better cost metric to more accurately reflect data structure. From this perspective, the data satisfies the first criterion of TMLR, meaning at least certain audiences will be interested in the paper's merits.

**Evidence**

Unfortunately, I do not believe this paper provides convincing evidence to demonstrate its technical merits. This conclusion is based on the following observations:

1. The key selling point of Geometric Optimal Transport (GOT) lies in its consideration of geometric space. However, apart from toy data, this paper does not clearly demonstrate this key advantage. I remain unclear on how geometric space is beneficial. For instance, the geometric space relies on the kernel, which is also related to Euclidean distance. This makes it much harder to understand why your choice specifically better fits real-world data. There is no support for this, which is arguably the most crucial part of the paper.
2. The theory seems somewhat unclear. Proposition 4.1 appears to lack insights into why the proposed metric is beneficial in UDA, or which data structures would particularly benefit from this approach. Based on the results, I didn't grasp this point.
3. Regarding the experimental results, I noticed that most results indicate a performance improvement. However, based on the hypothesis, certain data structures should better fit the manifold assumptions. My question is, how can real-world data, such as images, verify this?

---

> ### Author Response · Authors · 2025-08-27
>
> This is comment 1 out of 2.
>
> We thank the reviewer for the thorough review and valuable feedback.
>
> ---
>
> ### **The Gaussian kernel with Euclidean distance**
>
> We thank the reviewer for raising this concern.
>
> While the Gaussian kernel is indeed constructed from Euclidean distances, its role in our method is fundamentally different from using the squared Euclidean distance as the transportation cost.
> Under the manifold assumption, geodesic distances on a manifold are locally Euclidean.
> The Gaussian kernel, widely used to capture such manifold structures [1-3], does not enforce a purely Euclidean structure, but rather enables a representation that reflects the intrinsic geometry of the data.
>
> To clarify this distinction in the paper, we have added a note (highlighted in blue) after Equation 4.
>
> **References:**
> 1. Belkin, Mikhail, and Partha Niyogi. "Laplacian eigenmaps for dimensionality reduction and data representation." Neural computation 15, no. 6 (2003): 1373-1396.
> 2. Coifman, Ronald R., and Stéphane Lafon. "Diffusion maps." Applied and computational harmonic analysis 21, no. 1 (2006): 5-30.
> 3. Maaten, Laurens van der, and Geoffrey Hinton. "Visualizing data using t-SNE." Journal of machine learning research 9, no. Nov (2008): 2579-2605.
>
> ---
>
> ### **The manifold hypothesis**
>
> We thank the reviewer for this important question.
>
> The manifold hypothesis is a widely accepted assumption in the machine learning and computer vision communities, particularly for image data. It posits that high-dimensional observations, such as images, often lie on a lower-dimensional manifold. This assumption underpins many successful techniques, including manifold learning, dimensionality reduction, and deep representation learning [1-4].
> Several works have explored this hypothesis, both theoretically and empirically, using real-world datasets &#150; including image datasets [5-6].
> Under this assumption, methods that account for the geometric structure of the data, rather than relying solely on Euclidean distances, can better capture meaningful relationships between samples. Consequently, GOT can improve performance even when the geometry is not explicitly known, as it leverages local neighborhoods to approximate the intrinsic manifold structure.
>
> We note that in our toy example, no explicit manifold geometry is provided; the method relies solely on local neighborhood information, as in all our experiments.
> For the BCI datasets, it is known that the data lie on a Riemannian manifold, and we incorporate this information in both GOT and standard OT by using the Riemannian distance instead of the Euclidean distance.
> Even in this case, GOT achieves significantly better performance, demonstrating that exploiting local neighborhood structure provides additional benefits beyond using the correct global metric.
>
> In response to the reviewer’s comment, in addition to the short explanation already included at the beginning of Section 4, we have added a discussion of this point to the conclusions section (Section 6, highlighted in blue).
>
> **References:**
> 1. Tenenbaum, Joshua B., Vin de Silva, and John C. Langford. "A global geometric framework for nonlinear dimensionality reduction." science 290, no. 5500 (2000): 2319-2323.
> 2. Belkin, Mikhail, and Partha Niyogi. "Laplacian eigenmaps for dimensionality reduction and data representation." Neural computation 15, no. 6 (2003): 1373-1396.
> 3. Donoho, David L., and Carrie Grimes. "Hessian eigenmaps: Locally linear embedding techniques for high-dimensional data." Proceedings of the National Academy of Sciences 100, no. 10 (2003): 5591-5596.
> 4. Bengio, Yoshua, Aaron Courville, and Pascal Vincent. "Representation learning: A review and new perspectives." IEEE transactions on pattern analysis and machine intelligence 35, no. 8 (2013): 1798-1828.
> 5. Fefferman, Charles, Sanjoy Mitter, and Hariharan Narayanan. "Testing the manifold hypothesis." Journal of the American Mathematical Society 29, no. 4 (2016): 983-1049.
> 6. Pope, Phillip, Chen Zhu, Ahmed Abdelkader, Micah Goldblum, and Tom Goldstein. "The intrinsic dimension of images and its impact on learning." arXiv preprint arXiv:2104.08894 (2021).
>
> ---

---

> > ### Author Response · Authors · 2025-08-27
> >
> > This is comment 2 out of 2.
> >
> > ---
> >
> > ### **When GOT is beneficial**
> >
> > We thank the reviewer for this interesting question.
> >
> > While there is no theoretical guarantee that GOT will always outperform traditional OT costs, our empirical results consistently show that it achieves superior performance across diverse benchmarks.
> > This, combined with GOT's consideration of the geometric structure, and the manifold hypothesis &#150; which reflects that real-world data often lie on hidden low-dimensional manifolds &#150; makes GOT a strong and preferable choice for the cost in OT-based domain adaptation problems.
> >
> > Nevertheless, GOT introduces three parameters related to the Gaussian kernel scale, and poor choices can lead to degraded performance. Although several heuristics exist for setting these parameters, none can be considered universally optimal (See Appendix C.1.1, highlighted in blue).
> >
> > In summary, GOT is a favorable choice for most data structures compared to the standard Euclidean cost, but practitioners should apply it with care, particularly regarding kernel parameter selection.
> > We have added a discussion of this point to the main paper (see Section 6, highlighted in blue).
> >
> > ---
> >
> > ### **Clarification of deep vs. non-deep OT UDA experiments**
> >
> > We thank the reviewer for the comment.
> >
> > Our experiments are explicitly divided into deep OTDA experiments (Section 5.2) and non-deep non-Euclidean experiments (Section 5.3). The experimental frameworks are described in detail in the supplementary material.
> > This separation aligns with the reviewer’s request and highlights the generality of GOT, as it performs effectively across both deep and non-deep OT UDA settings.
> >
> >
> > ---

---

> > > ### Comment · Reviewer_eA7v · 2025-09-12
> > >
> > > Thank you. Parts of my questions have been addressed. But
> > >
> > > I still feel the theory cannot explain the benefit of the manifold, so if we decide to put the theory without clearly explaining the benefit of the manifold role in domain adaptation, why a theory is required in the paper? My comments on the theory is about why this theory is useful in the understanding, this can include that theory provided a tighter bound when manifold is considered in certain cases. But from current results, the role of theory is missing.

---

> > > > ### Author Response · Authors · 2025-09-15
> > > >
> > > > We thank the reviewer for the feedback.
> > > > In the revised paper, Figure 5 in Appendix B.3 provides a more intuitive illustration of the theoretical analysis, which may help clarify the role and usefulness of the theory in the paper.
> > > >
> > > > We would be happy to address any further questions the reviewer may have regarding the theory.

---

### Review · Reviewer_svjb · 2025-08-11

**Summary Of Contributions:**

This paper studies the problem of applying optimal transport to the domain adaptation problem (OTDA). The main underlying contribution is to use the diffusion geometry for this task. In particular, the authors construct the cost matrix based on a diffusion operator between the domains. This cost matrix is a product of three similarity matrices that capture intra- and inter-domain diffusion operators. The authors provide a theoretical justification for their approach, showing that the diffusion operator they propose makes sense for the DA problem. Extensive experimental results suggest that the proposed approach works well in practice.

**Audience:**

Yes

**Claims And Evidence:**

Yes

**Requested Changes:**

1. The authors missed a bunch of references on learning the cost matrix in OT. I think those should be discussed in this manuscript, and a visualization of the different learned ground matrices may be provided to illustrate the differences between them.

Here are some references:

1. Cuturi et David, "Ground Metric Learning", JMLR, 2014.
2. Dhouib et al. "Margin-aware Adversarial Domain Adaptation with Optimal Transport", ICML'20
3. Huizing et al. "Unsupervised Ground Metric Learning Using Wasserstein Singular Vectors", ICML'22
4. Jawanpuria et al. "A Riemannian Approach to Ground Metric Learning for Optimal Transport", arxiv'24.
5. Kerdoncuff et al. "Metric Learning in Optimal Transport for Domain Adaptation", IJCAI'20.

The authors can insist on defining the ground metric without learning it to better position their approach but still they need to cite the relevant work and compare whenever possible.

2. The most recent non-OT DA baseline dates back to 2020. It would be nice to include at least 1 or 2 recent baselines
3. I think that the analysis of Proposition 4.1 is not very intuitive. It would be nice to illustrate its different parts to make it clearer for the audience not familiar withg diffusion geometry.

**Strengths And Weaknesses:**

**Strengths**

1. Novel idea for constructing the cost matrix for the OTDA problem without relying on min max approach or learning it from data
2. Theoretical justification
3. Extensive experimental results with many baselines

**Weaknesses**

1. Missing references
2. More recent non-OT baselines could have been included
3. It is not clear how to interpret the theoretical results

---

> ### Author Response · Authors · 2025-08-27
>
> This is comment 1 out of 3.
>
> We thank the reviewer for the thorough review and constructive suggestions.
>
> ---
>
> ### **Additional references**
>
> We thank the reviewer for sharing these works with us.
>
> We first discuss papers [d] and [e], which we consider highly relevant to our work, and provide an initial comparison with these methods on the Office-Caltech dataset.
>
> In both [d] and [e], the authors propose to use the Mahalanobis distance for the cost matrix:
>
> $$
> C_{ij}(A) = d_A^2(x_i^s, x_j^t) =(x_i^s - x_j^t)^T A(x_i^s-x_j^t),
> $$
>
> where instead of using the covariance, they propose to learn the matrix $A$.
> In [d], a regularization term is introduced, which not only prevents trivial solutions but also enables a closed-form solution for $A$, given the transport plan $\gamma$. The problem is then solved through alternating optimization: starting with an initialization $\gamma_0$, compute $A$; update the cost matrix using $d_A$; and solve the OT problem with Sinkhorn to obtain a new $\gamma$. This process is repeated for a fixed number of iterations.
> In this approach, the learned cost $C(A)$ is optimized to minimize the OT loss $\langle C(A), \gamma \rangle$.
> Note that the Mahalanobis distance only accounts for inter-domain relationships (similar to the Euclidean cost), and the source labels are not incorporated in the optimization process.
> In [e], the authors use the decomposition $A = L^T L$, and instead of learning the Mahalanobis distance $\|L(x_i^s-x_j^t)\|_2^2$, they propose the cost
>
> $$
> C_{i,j}(L_s) = \|L_s x_i^s-L_t x_j^t\|_2^2,
> $$
>
> where $L_s$ is learned, and $L_t$ is computed via PCA, a choice justified theoretically.
> In addition to the entropy and class regularization terms typically used in OT-based domain adaptation, they introduce a regularization term on $L_s$ that leverages source labels during metric optimization. Like in [d], this cost considers only inter-domain relationships, and intra-domain structure is not incorporated.
>
> We plan to implement and evaluate these methods on the BCI dataset and will include the results in the revised paper during the two-week discussion period.
> In the meantime, due to time constraints, we provide an initial comparison on the Office-Caltech dataset, which was used in these works. The tables below report our results alongside those of [d] and [e], taken from their original papers.
> The method in [d] tuned hyperparameters using target labels, splitting the target domain into training and test sets. However, since neither the code nor the specific split is available, we compare against the optimal set of parameters, i.e., those that maximize target accuracy.
> In contrast, [e] relies solely on source labels for tuning, inspired by [1]. To ensure fairness, we implemented a cross-validation procedure based only on source labels and report results for both target-tuned and source-tuned versions.
> We also included the baselines OT and OT-reg in the table. For all methods, we followed the preprocessing of [e], applying PCA to both source and target samples. The number of PCA components (including the option of no PCA) was treated as a tunable hyperparameter.
>
> ---

---

> ### Author Response · Authors · 2025-08-27
>
> This is comment 2 out of 3.
>
> ---
>
> ### **Additional references &#150; Tables**
>
> **Table 1.** SURF features (unsupervised tuning)
>
> | Method | A$\to$C | A$\to$D | A$\to$W | C$\to$A | C$\to$D | C$\to$W | D$\to$A | D$\to$C | D$\to$W | W$\to$A | W$\to$C | W$\to$D | AVG |
> | --- | --- | --- | --- | --- | --- | --- | --- | --- | --- | --- | --- | --- | --- |
> | MLOT [e] | **42.3** | 40.8 | 41.3 | 51.5 | 52.2 | 45.9 | **37.8** | **34.4** | **87.8** | 38.0 | 33.2 | **90.8** | 49.7 |
> | OT | 35.6 | 31.2 | 34.6 | 36.1 | 44.6 | 28.8 | 31.5 | 30.6 | 79.3 | 33.8 | 30.7 | 87.9 | 42.1 |
> | OT-reg | 37.0 | **42.0** | 41.7 | 46.9 | 47.8 | 50.2 | 31.0 | 32.7 | 86.1 | 35.4 | 30.8 | 79.6 | 46.8 |
> | GOT | 38.1 | 41.4 | **43.4** | **51.7** | **52.9** | **57.6** | 35.6 | 33.1 | 87.5 | **40.7** | **34.9** | 86.0 | **50.2** |
>
> ---
>
> **Table 2.** SURF features (supervised tuning)
>
> | Method | A$\to$C | A$\to$D | A$\to$W | C$\to$A | C$\to$D | C$\to$W | D$\to$A | D$\to$C | D$\to$W | W$\to$A | W$\to$C | W$\to$D | AVG |
> | --- | --- | --- | --- | --- | --- | --- | --- | --- | --- | --- | --- | --- | --- |
> | MLOT [e] |  |  |  |  |  |  |  |  |  |  |  |  | **55.1** |
> | OT | 36.8 | 38.9 | 39.3 | 46.1 | 46.5 | 39.7 | 35.8 | 34.0 | 89.2 | 40.5 | 34.4 | 91.7 | 47.7 |
> | OT-reg | **43.3** | 42.0 | 49.8 | **56.4** | 51.0 | 55.6 | 41.4 | 36.3 | 90.2 | 41.0 | 35.4 | **93.0** | 53.0 |
> | GOT | 41.6 | **47.8** | **51.9** | 56.1 | **57.3** | **61.0** | **42.0** | **37.7** | **91.5** | **44.2** | **37.5** | 92.4 | **55.1** |
>
> ---
>
> **Table 3.** DECAF features (unsupervised tuning)
>
> | Method | A$\to$C | A$\to$D | A$\to$W | C$\to$A | C$\to$D | C$\to$W | D$\to$A | D$\to$C | D$\to$W | W$\to$A | W$\to$C | W$\to$D | AVG |
> | --- | --- | --- | --- | --- | --- | --- | --- | --- | --- | --- | --- | --- | --- |
> | MLOT [e] |  |  |  |  |  |  |  |  |  |  |  |  | 84.7 |
> | OT | 81.0 | 81.5 | 73.2 | 82.4 | **87.9** | 80.0 | 75.3 | 71.6 | **95.6** | 72.3 | 69.7 | **98.1** | 80.7 |
> | OT-reg | 82.1 | 79.0 | 74.2 | 88.1 | 87.3 | **82.7** | 72.5 | 72.5 | 92.9 | 72.5 | 68.7 | 89.2 | 80.1 |
> | GOT | **85.7** | **82.8** | **87.5** | **88.8** | **87.9** | **82.7** | **90.1** | **82.6** | 82.4 | **87.9** | **78.2** | 87.3 | **85.3** |
>
> ---
>
> **Table 4.** DECAF features (supervised tuning)
>
> | Method | A$\to$C | A$\to$D | A$\to$W | C$\to$A | C$\to$D | C$\to$W | D$\to$A | D$\to$C | D$\to$W | W$\to$A | W$\to$C | W$\to$D | AVG |
> | --- | --- | --- | --- | --- | --- | --- | --- | --- | --- | --- | --- | --- | --- |
> | Method in [d] | 83.39 | 80.0 | 79.59 | 87.79 | 79.75 | 72.57 | 87.24 | 83.14 | 95.0 | 81.41 | 78.47 | 93.16 | 83.46 |
> | OT | 84.68 | 84.71 | 78.98 | 89.67 | 87.9 | 85.08 | 88.73 | 82.9 | 97.63 | 89.04 | 79.79 | **100.0** | 87.43 |
> | OT-reg | 85.49 | 86.62 | 84.41 | 90.61 | **90.45** | 88.14 | 88.73 | 82.9 | 97.63 | 89.35 | 80.68 | **100.0** | 88.75 |
> | GOT | **87.8** | **89.17** | **91.19** | **92.9** | **90.45** | **89.49** | **92.07** | **86.29** | **98.98** | **91.75** | **81.83** | **100.0** | **90.99** |
>
> Paper [e] (shown as MLOT [e] in the table) reports results for both SURF and DECAF features; however, source&#150;target pair results are only available for SURF, tuned using source labels. Comparing the source-tuned versions, GOT achieves higher target accuracy than [e] for both SURF and DECAF (Tables 1 and 3).
> For SURF, [e] also reports the average target accuracy using the optimal parameters tuned with target labels. In this case, GOT performs comparably to [e] (Table 2).
>
> Paper [d] (shown as Method in [d] in the table) reports results only using DECAF features. As mentioned above, this method uses target labels for hyperparameter tuning; therefore, we compare it to the target-tuned version of our results, presented in Table 4.
> As can be seen in the table, GOT achieves superior performance compared to [d], as well as to the baselines OT and OT-reg. Notably, [d] also reports results for OT ($81.97$%), which is closer to the unsupervised-tuned version in our experiments ($80.72$%). Since the code for [d] is not publicly available, we cannot fully explain this discrepancy; however, we note that even with unsupervised tuning (Table 3), GOT still outperforms [d] by approximately $2$%.
>
> ---

---

> ### Author Response · Authors · 2025-08-27
>
> This is comment 3 out of 3.
>
> ---
>
> ### **Additional references &#150; continued**
>
> While [b] is not included in the tables, as it only provides results on the Amazon Reviews dataset, we consider this work relevant for comparison as well.
> In this work, the authors derive a bound on the target margin violation rate and propose a DA algorithm based on this bound, which jointly learns the transport plan $\gamma$ and a linear classifier $w$ (similarly to JDOT, discussed in Section 2 of the paper).
> Specifically, the resulting objective for optimizing $\gamma$ (given $w$) is:
>
> $$
> \Big\|\sum_{i\le m,\, j \le n} \gamma_{ij} |D_{ij} w| \Big\|_{\infty},
> $$
>
> where $D_{ij} = x_i x_i^T - x_j x_j^T$.
> In practice, they use the dual representation of the infinity norm and solve a min&#150;max problem.
> From an OT viewpoint, the induced cost function is:
>
> $$
> C_{ij} = \mathbf{q}^T |(x_i^{(s)} x_i^{(s)T} - x_j^{(t)} x_j^{(t)T}) w|,
> $$
>
> where $q\in \Delta_d$ is a learned vector and $\Delta_d$ is the probability simplex.
> In summary, this work proposes a new cost function for the OTDA problem, which considers only inter-domain relationships and does not incorporate source labels.
> We plan to include this method, along with [d] and [e], in the BCI dataset comparisons during the discussion period.
> In the meantime, we have added descriptions of all three works to Section 2 of the revised paper (highlighted in blue).
>
> As for [a] and [c], although both propose new cost functions for the OT problem, we found them incompatible for comparison to GOT in the UDA context.
> Specifically, [a] focuses on learning a distance for normalized histograms using the Wasserstein distance in a semi-supervised setting. It assumes access to coefficients  {$w_{ij}$} representing similarities between samples $i$ and $j$, and learns the cost matrix for the OT problem such that Wasserstein distances between similar histograms are small and those between dissimilar histograms are large. While this work proposes to learn the cost matrix, it relies on supervised information and assumes a single set of samples.
> Similarly, the work in [c] is also designed for a single domain. It jointly computes OT distances between samples and between features, such that the cost matrix for sample distances is given by the resulting Wasserstein distance between features, and vice versa. While this is an interesting idea, it cannot be compared to GOT in the UDA setting.
>
> **References:**
> 1. Zhong, Erheng, Wei Fan, Qiang Yang, Olivier Verscheure, and Jiangtao Ren. "Cross validation framework to choose amongst models and datasets for transfer learning." In Joint European Conference on Machine Learning and Knowledge Discovery in Databases, pp. 547-562. Berlin, Heidelberg: Springer Berlin Heidelberg, 2010.
>
> ---
>
> ### **Comparison to non-OT baselines**
>
> We thank the reviewer for this constructive suggestion.
>
> Our main objective in the experimental evaluation is to highlight the advantages of our proposed GOT cost compared to other cost functions designed for the OTDA problem. For this reason, we primarily focused on competing methods that introduce new transportation costs within the OT framework and address the DA task.
> Nonetheless, we recognize the impressive progress in DA achieved with recent SOTA architectures.
> Since the original DeepJDOT framework is not designed to compete with such non-OT methods, including them in the main body could give a misleading impression of the potential of GOT.
> To address this, in Table 9 of Appendix C.3.3 we report results obtained by integrating GOT into a stronger backbone, specifically a vision transformer, and compare them with recent SOTA methods.
>
> ---
>
> ### **Further intuition for the theoretical results**
>
> We thank the reviewer for this valuable suggestion.
>
> We fully agree that providing an illustration of the different parts of Proposition 4.1 would help convey the intuition behind the theoretical analysis.
> Due to time constraints, we are not able to provide the illustration in this response, but we are preparing such a simulation and hope to share the results during the two-week discussion period.
>
> In the meantime, we highlight the three steps of the diffusion process using the simulation already included in this part of the paper, which provides some intuition on how the diffusion operator behaves in practice (see Figure 4 in Appendix B.2).
>
> ---

---

> ### Author Response · Authors · 2025-09-11
>
> ### **Extended comparison with additional competing methods**
>
> We further evaluate the competing methods MADAOT [b], RMLOT [d] (Since the original paper did not provide a name, we refer to this method as RMLOT &#150; Riemannian Metric Learning Optimal Transport), and MLOT [e] on the MI2 BCI dataset for multi-class classification. We compare GOT against these approaches and the baselines OT and OT-reg under two parameter tuning strategies: (i) tuning based on source labels, and (ii) tuning based on target labels. In the latter case, we report the best-performing parameters according to the target-domain accuracy.
> For source-label tuning, we adopt a more robust selection procedure than the one used in MLOT, which selects the parameter set with the maximum source accuracy. Specifically, we first collect the top $5$% of parameter configurations with the highest source accuracy, or alternatively the configurations within 5 points of the maximum source accuracy (whichever results in a larger set). For each parameter, we then select the most frequent value in this subset. In case of ties, we resolve by choosing the value with the highest source accuracy. For the parameters $\epsilon_s$, $\epsilon_c$, and $\epsilon_t$, which are naturally more sensitive to the data (and to each other), we instead take the average value across the subset. This procedure is applied consistently across all competing methods.
>
> For OT, OT-reg, and GOT, we use the same algorithm as in all BCI experiments in the paper, utilizing covariance features and the Log-Euclidean metric as the cost function.
> For fairness, in MADAOT, which assumes Euclidean geometry, and in RMLOT and MLOT, which are limited to the Mahalanobis distance, we project the covariance matrices from the SPD manifold to a tangent plane (using Eq. 18 in Appendix C.4) before applying the methods.
>
> We note that we do not report results for the multi-source DA setting. While feasible in principle, integrating the competing methods into the WBT framework would be considerably more complex and computationally intensive, as it requires learning two separate metrics and tuning additional parameters.
>
> The results have been added to Section 5.3 in the main paper (Tables 4–5), accompanied by a discussion (highlighted in red), with the full results provided in Appendix C.4.2 (Tables 18–21). For completeness, the multi-class classification results obtained using a single parameter set for all experiments, tuned by target labels, remain available in Appendix C.4.2 (Tables 16-17).
>
> We thank the reviewer for suggesting the inclusion of these methods. We believe that their addition provides a more comprehensive comparison and enriches the experimental analysis of the paper.

---

> > ### Comment · Reviewer_svjb · 2025-09-11
> > **Thank you for the revision**
> >
> > I would like to thank the authors for their explanations and additional experiments. I have no more concerns, although a nice illustration for the Theorem would still be great to have.

---

> > > ### Author Response · Authors · 2025-09-15
> > >
> > > We thank the reviewer for the positive feedback and fully agree that an illustration of the theorem would be valuable.
> > > While we were unable to provide it within the official 4-week deadline, the revised paper now includes it in Figure 5 in Appendix B.3, with the discussion highlighted in red.
> > >
> > > We sincerely appreciate the reviewer’s constructive and helpful suggestions.

---

### Review · Reviewer_UqCV · 2025-08-14

**Summary Of Contributions:**

If Optimal Transport has been widely used for Domain Adaptation, the cost matrix used to compute the transport is the pairwise Euclidean distance. The authors introduce a new cost matrix called Geometric Optimal Transport (GOT), leveraging diffusion geometry and manifold learning, that allows for better transport between the two domains. The GOT cost matrix comes with theoretical insight and experiments over two modalities, image classification and  BCI Motor Imagery.

**Audience:**

Yes

**Claims And Evidence:**

Yes

**Requested Changes:**

Please go through the weaknesses to know what needs to change. This is a brief summary:

- Add a better introduction of the DeepJDOT joint loss to understand that there are two terms.
- Implement competitors for BCI experiments.
- Add a paragraph about hyperparameter selection with a sensitivity study of scale hyperparameters of GOT.
- Add computational cost comparison between different cost matrices in practice.
- Modify Figures 1 and clarify the title of Figure 2.

**Strengths And Weaknesses:**

Strengths:
- The authors propose a new method for computing a cost matrix using the manifold assumption that makes the transport between domains more robust.
- The new cost matrix went with theoretical insights about why this cost matrix is useful for domain adaptation, which is not the case in OT DA papers cited in the paper.
- This new cost is tested first on a toy example showing its efficiency, then GOT is incorporated into different baselines to show its superiority over competitors for image classification and motor imagery.

Weaknesses:
- GOT is used inside the framework of DeepJDOT, but DeepJDOT is not properly introduced in the main paper. The cost matrix used in DeepJDOT is both the Euclidean distance + a term of cross-entropy loss between the target pseudo-label and the true source label. You explain everything in the appendix, but I think it would be nice to be more transparent directly in the main paper.
- I think having a fair comparison for BCI experiments would bring more impact to your results. Right now, the authors report results from two different papers. With different software like moabb or pyriemmann, I think implementing the benchmark is doable and would bring more trust in the results.
- The scale hyperparameters are fixed to 1 for deep learning adaptation. I'm wondering if the method is sensitive to these parameters? Maybe adding a sensitive study could help practitioners apply the method easily.
- How did you choose the number of 3 nearest neighbors for the target? Is it a robust choice for different modalities?
- The majority of the methods used in the paper need hyperparameters. How are these hyperparameters tuned in practice? A paragraph should be added for transparency and reproducibility.
- The authors gave a computational cost comparison in the appendix; it would be nice to have a comparison in practice for computing the different OT distances.
- Do you think GOT should automatically be used as a cost matrix in OT-based DA? Or is it a choice that practitioners should make? Maybe adding a discussion about that could be helpful.

Small weaknesses:
- Maybe adding a new blue dot for Figure 1 could be clearer. Right now, it is propagated to three source dots and three target dots. But in your method, you can have more for the source, but only 3 for the target. Adding one blue dot can help to understand the difference.
-The title of Figure 2 should be clearer that you are plotting only the column $\gamma(., j)$ and the 10 highest values, even if it is said in the paragraph.
- There is no name for the WBT acronym (Wasserstein Barycenter Transfer ?)

---

> ### Author Response · Authors · 2025-08-27
>
> This is comment 1 out of 2.
>
> We thank the reviewer for the thorough review and insightful comments and questions.
>
> ---
>
> ### **Hyperparameter selection and sensitivity**
>
> We thank the reviewer for their detailed questions regarding the method’s hyperparameters.
>
> **The Gaussian scale:**
> The Gaussian kernel is known to be highly sensitive to its scale parameter. When the scale is too large, the kernel fails to distinguish between samples, leading to a loss of important information. Conversely, when the scale is too small, the kernel becomes overly sensitive to noise, which may result in overfitting. Moreover, the choice of scale is inherently dependent on the data scale.
> There is no single optimal methodology for selecting the Gaussian kernel parameter.
> Moreover, determining the appropriate kernel scale is an open problem that remains an active area of research across various fields, beyond OT and domain adaptation.
> A common practice is to set it as the median of the pairwise distances in the dataset, which is the approach we utilized in all our experiments. In the deep experiments, where many additional model and loss parameters were already involved, we fixed the Gaussian scale to $\epsilon=1$ and applied only the median normalization. This choice proved robust in practice, though further tuning could potentially yield even better performance.
> For the remaining experiments, in addition to the median normalization, we tuned the parameters $\epsilon_s, \epsilon_c, \epsilon_t$.
>
> A detailed discussion of this parameter is now included in Appendix C.1.1.
>
> While the effect of $\epsilon$ on the proposed cost is similar to the effect of the scale parameter in any method that uses the Gaussian kernel for similarity measurement, we will consider adding an ablation study or illustrative visualization during the discussion period, if time permits, to provide additional intuition.
>
> **The number of neighbors:**
> In general, the neighborhood of the kernel is defined by the scale parameter $\epsilon$. For the source and target operators, we enhance the neighborhoods either by incorporating label information (when available) or by choosing the neighborhood as the $K$ nearest neighbors. In all our experiments, we fixed $K=3$, and found this choice to be empirically robust. While tuning $K$ could potentially improve performance, its influence was less pronounced compared to other parameters, so we opted to keep it fixed.
>
> We agree that this point should be explicitly discussed in the paper, and we have added a discussion in Appendix C.1.2.
> Figure 5 illustrates the target accuracy in several BCI experiments as a function of $K$.
> Notably, we used the same scale parameters as in the multi-class experiments, modifying only the number of target neighbors. With full parameter tuning, it is possible that different values of $K$ would yield better results.
>
> **Hyperparameter tuning:**
> For the competing methods, we followed the hyperparameter settings reported in their respective papers, with one exception &#150; DeepJDOT.  The reported parameters for DeepJDOT led to poor results in our setting, so we re-tuned them ourselves.
> Our method, GOT, requires tuning four parameters: the diffusion operator parameters $\epsilon_s, \epsilon_c, \epsilon_t$ and the entropy regularization weight $\lambda$ (due to the use of the Sinkhorn algorithm).
> While target labels were not used during training, we did rely on them as a validation set for hyperparameter tuning, consistent with competing methods. Importantly, we optimized parameters for each dataset as a whole, rather than for each individual source-target pair. For example, the same set of parameters was used across all multi-class experiments on the BCI dataset (MI2).
>
> For transparency, we have added a discussion of this choice in the paper (see Appendix C.1.3).
>
> In addition, in reviewing the works suggested by Reviewer svjb, we identified a cross-validation strategy that relies solely on source labels for parameter tuning [1]. This approach yielded reasonable performance on the Office-Caltech dataset (though not optimal), and we plan to extend it to the BCI experiments as well.
> We hope to share these results during the two-week discussion period and include both versions in the revised paper.
>
> **References:**
> 1. Zhong, Erheng, Wei Fan, Qiang Yang, Olivier Verscheure, and Jiangtao Ren. "Cross validation framework to choose amongst models and datasets for transfer learning." In Joint European Conference on Machine Learning and Knowledge Discovery in Databases, pp. 547-562. Berlin, Heidelberg: Springer Berlin Heidelberg, 2010.
>
> ---

---

> ### Author Response · Authors · 2025-08-27
>
> This is comment 2 out of 2.
>
> ---
>
> ### **Computational time comparison**
>
> We thank the reviewer for this helpful suggestion.
>
> Following the comment, we added a practical comparison of the computational time using the two moons toy example (see the last paragraph in Appendix C.2, highlighted in blue).
> In Figure 8, we report the run-time for different numbers of samples for OT, OT-reg, OT-Laplace, and the proposed GOT, together with the number of iterations.
>
> This experiment illustrates the difference between theoretical complexity and practical performance, showing that although both GOT and OT-Laplace have a theoretical complexity of $\mathcal{O}(n^3)$, OT-Laplace is considerably slower in practice.
> This is because the main cost in GOT arises from computing the cost matrix (done once), whereas in OT-Laplace it stems from the optimization process, which requires many iterations.
> Moreover, GOT, which leverages the Sinkhorn algorithm, converges after only a few iterations, making it empirically faster even than OT-reg, despite the latter having a lower theoretical complexity of $\mathcal{O}(n^2)$.
> We believe this empirical comparison strengthens the contribution of our approach, and we greatly appreciate the reviewer’s suggestion to include it.
>
> ---
>
> ### **When to use GOT?**
>
> We thank the reviewer for this interesting question.
>
> While there is no theoretical guarantee that GOT will always outperform traditional OT costs, our empirical results consistently show that it achieves superior performance across diverse benchmarks.
> More broadly, real-world datasets are widely believed to lie on manifolds rather than being distributed in Euclidean space. GOT considers this geometric structure, which helps explain its consistent empirical advantage.
>
> Nevertheless, GOT introduces three parameters related to the Gaussian kernel scale, and poor choices can lead to degraded performance. Although several heuristics exist for setting these parameters, none can be considered universally optimal.
> To mitigate this, careful tuning of the hyperparameters is essential. When only source labels are available, as in the UDA setting, several strategies can be applied for parameter selection (as explored in our response to Reviewer svjba). While such approaches may not yield the absolute best performance, they provide a principled way to select reasonable parameters and help avoid severe drops in accuracy.
>
> In summary, we believe that GOT is a strong choice for OT-based DA on complex data, but practitioners should apply it with care, particularly regarding kernel parameter selection.
> In response to this comment, we have added a discussion of this point to the main paper (see Section 6, highlighted in blue).
>
> ---
>
> ### **Fair comparison in BCI experiments**
>
> We thank the reviewer for this valuable suggestion.
>
> We agree that, although we ensured to follow the same preprocessing as in the competing methods, a fully fair comparison would ideally involve implementing and running the baselines ourselves.
> Unfortunately, due to time constraints and the multiple tasks we are addressing during the rebuttal, we are unable to provide such results at this stage. If time permits, we will make an effort to include them during the two-week discussion period.
>
> ---
>
> ### **Suggested revisions implemented**
>
> We thank the reviewer for the helpful suggestions; the following changes have been incorporated directly into the paper.
>
> - We added a short overview of the DeepJDOT framework in the main paper, clarifying the cost used in DeepJDOT (see Section 5.2, highlighted in blue).
> - In Figure 1, we added a new blue source sample to better illustrate that source neighborhoods are defined by class, whereas target neighborhoods are defined by 3-nearest neighbors. We thank the reviewer for this suggestion.
> - We revised the caption of Figure 2 as suggested.
> - We added the acronym WBT (Wasserstein Barycenter Transport) to the main text.
>
> ---

---

> ### Author Response · Authors · 2025-09-11
>
> ### **Hyperparameter tuning &#150; unsupervised selection**
>
> As mentioned in our initial response, over the past two weeks we explored an unsupervised tuning approach that relies solely on source labels for hyperparameter selection.
> Results obtained using this approach are presented in Section 5.3 of the main paper (Tables 4-5) and in Appendix C.4.2 (Tables 18-21), with detailed discussion highlighted in red.
> Notably, these results are very similar to those obtained using a single set of parameters for the entire dataset, tuned with target labels (Tables 16-17).
>
> Due to time constraints, we could not implement the competing methods for the BCI binary classification task. However, for the multi-class experiments, all competing methods were run by us within our unified framework: MADAOT and MLOT using their publicly available code, and RMLOT implemented by us.
>
> ---
>
> ### **Gaussian scale parameters &#150; sensitivity study**
>
> To address the reviewer’s suggestion, we conducted a sensitivity analysis of the Gaussian scale parameters. We used the multi-class classification task on dataset MI2, with parameters fixed to $\lambda = 0.01$ and $(\epsilon_s, \epsilon_c, \epsilon_t)$ varied one at a time, while keeping the others fixed to $1$. This evaluation was performed after applying median normalization. We report results for both cross-subject and cross-session tasks, considering all subjects and the five valid subjects.
>
> The results (see Figure 5) indicate that $\epsilon_s$ and $\epsilon_c$ are highly sensitive: very small or very large values lead to noticeable drops in performance. In contrast, $\epsilon_t$ yields relatively stable results. This is expected, as the neighborhood is already strongly constrained by the choice of three neighbors, and increasing $\epsilon_t$ does not substantially alter it.
>
> Importantly, while setting all scales to $1$ (as we did in the deep experiments) does not always achieve peak performance, it consistently provides reasonable accuracy and thus serves as a robust, though not optimal, choice. We further note that joint tuning of all three scales may improve performance even in regions where individual parameter sweeps appear suboptimal, since the parameters interact and collectively shape the final cost.
>
> A discussion of this sensitivity study has been added to Appendix C.1.1 (highlighted in red).
>
> We thank the reviewer for the suggestion to further discuss the GOT parameters and the tuning process. In particular, we believe that this sensitivity study provides valuable insights for practitioners and adds guidance for future applications.

---

> > ### Comment · Reviewer_UqCV · 2025-09-22
> >
> > I thank the authors for the answers. Most of my concerns have been answered, and the new discussions/results added in the paper make it stronger, and it is easier to know when and how to use GOT. I still maintain that this paper has good claims and it is perfectly suitable for the TMLR audience.

---

### Decision · Action_Editor_bVeV · 2025-09-29

**Recommendation:** Accept as is

**Audience:**

Yes

**Audience Explanation:**

Domain Adaptation is a fundamental ML problem and the paper is clearly of interest to the community.

**Claims And Evidence:**

Yes

**Claims Explanation:**

The paper proposes to use the geometry of the space (the data manifold), associated to a diffusion process for Optimal Transport Domain Adaptation. The claims are reasonable and well supported by experiments. All reviewers found the paper interesting but had some comments that were taken into account in the revision of the paper.